# Geometric constraints on human brain function

James C. Pang[1,7 ✉], Kevin M. Aquino[2,3,7], Marianne Oldehinkel[4], Peter A. Robinson[2], Ben D. Fulcher[2], Michael Breakspear[5,6] & Alex Fornito[1]

The anatomy of the brain necessarily constrains its function, but precisely how remains unclear. The classical and dominant paradigm in neuroscience is that neuronal dynamics are driven by interactions between discrete, functionally specialized cell populations connected by a complex array of axonal fibres[1–3]. However, predictions from neural field theory, an established mathematical framework for modelling large-scale brain activity[4–6], suggest that the geometry of the brain may represent a more fundamental constraint on dynamics than complex interregional connectivity[7,8]. Here, we confirm these theoretical predictions by analysing human magnetic resonance imaging data acquired under spontaneous and diverse task-evoked conditions. Specifically, we show that cortical and subcortical activity can be parsimoniously understood as resulting from excitations of fundamental, resonant modes of the brain's geometry (that is, its shape) rather than from modes of complex interregional connectivity, as classically assumed. We then use these geometric modes to show that task-evoked activations across over 10,000 brain maps are not confined to focal areas, as widely believed, but instead excite brain-wide modes with wavelengths spanning over 60 mm. Finally, we confirm predictions that the close link between geometry and function is explained by a dominant role for wave-like activity, showing that wave dynamics can reproduce numerous canonical spatiotemporal properties of spontaneous and evoked recordings. Our findings challenge prevailing views and identify a previously underappreciated role of geometry in shaping function, as predicted by a unifying and physically principled model of brain-wide dynamics.

The dynamics of many natural systems are fundamentally constrained by their underlying structure. For instance, the shape of a drum influences its acoustic properties, the morphology of a river bed shapes underwater currents and the geometry of a protein determines the molecules with which it can interact[9]. The nervous system is no exception, with the rich and complex spatiotemporal dynamics of anatomically distributed neuronal populations being supported by their intricate web of axonal interconnectivity[1,10]. Several studies have shown correlations between various properties of brain connectivity and activity[11], but precisely how spatiotemporal patterns of neural dynamics are constrained by a relatively stable neuroanatomical scaffold remains unclear.

In diverse areas of physics and engineering, structural constraints on system dynamics can be understood via the system's eigenmodes, which are fundamental spatial patterns corresponding to the natural, resonant modes of the system[12]. In the linear regime, such as brain activity under normal (that is, non-seizure-like) conditions[13], eigenmodes (hereafter also referred to as modes) offer a particularly powerful and

rigorous formalism for linking brain anatomy with the physical processes that shape activity. Through this lens, spatiotemporal patterns of neuronal dynamics emerge from excitations of the brain's structural eigenmodes, much like the harmonics of a plucked violin string arise from vibrations of its own resonant modes.

Critically, just as the resonant frequencies of a violin string are determined by its length, density and tension, the eigenmodes of the brain are determined by its structural—physical, geometric and anatomical—properties. Do any of these specific structural properties make a dominant contribution to dynamics? Here we test two influential and competing theories that make different predictions about which key elements of brain structure shape function.

One classical perspective, which represents the dominant paradigm in neuroscience, has its roots in Ramon y Cajal's neuron doctrine[14], Brodmann's cytoarchitectonics[15] and over a century of work localizing functions to specific brain regions[16]. According to this view, spatiotemporal patterns of neural dynamics arise from interactions between discrete, functionally specialized cell populations connected by a topologically

[1]The Turner Institute for Brain and Mental Health, School of Psychological Sciences and Monash Biomedical Imaging, Monash University, Clayton, Victoria, Australia. [2]School of Physics, University of Sydney, Camperdown, New South Wales, Australia. [3]BrainKey Inc., San Francisco, CA, USA. [4]Donders Institute for Brain, Cognition, and Behaviour, Radboud University Medical Centre, Nijmegen, the Netherlands. [5]School of Psychological Sciences, College of Engineering, Science and the Environment, University of Newcastle, Callaghan, New South Wales, Australia. [6]School of Medicine and Public Health, College of Health, Medicine and Wellbeing, University of Newcastle, Callaghan, New South Wales, Australia. [7]These authors contributed equally: James C. Pang, Kevin M. Aquino. ✉e-mail: james.pang1@monash.edu

complex array of short- and long-range axonal connections[2,3]. In humans, these connections can be estimated at macroscopic scales by diffusion magnetic resonance imaging (dMRI) to yield a structural connectivity matrix, called a connectome[17]. This approach has been used extensively to understand brain organization and dynamics[2,17,18], and recent work has proposed that eigenmodes derived from such a discrete connectome—referred to here as connectome eigenmodes—can be used to reconstruct the spatial patterns of canonical functional networks of the human cortex mapped with functional MRI (fMRI)[19,20].

One limitation of this discrete connectome-based view is that it relies on an abstract representation of brain anatomy that does not directly account for its physical properties and spatial embedding (that is, geometry and topology). These characteristics are explicitly incorporated into a broad class of neural field theories (NFTs)[4–6] that describe mean-field neural dynamics on spatial scales above 0.5 mm (Supplementary Information 1). One physiologically constrained form of NFT has unified a diverse range of empirical phenomena[6,21] by treating cortical activity as a superposition of travelling waves propagating through a physically continuous sheet of neural tissue. In this theory, neural interactions between different cortical locations are approximated by a homogeneous spatial kernel that declines roughly exponentially with distance[22], in line with experimental evidence that the organization of the nervous systems of numerous species is universally governed by an exponential distance rule (EDR) for connectivity[10,23,24].

Given wave-like dynamics and EDR-like connectivity, a key prediction of NFT is that the intrinsic geometry of the brain physically shapes and imposes boundary conditions on emergent dynamics[7,8,25]. A remarkable corollary of this view is that, if we prioritize spatial and physical constraints on brain anatomy, we only need to consider the shape of the brain, and not its full array of topologically complex axonal interconnectivity, to understand spatially patterned activity. More formally, the theory predicts that eigenmodes derived from brain geometry—hereafter referred to as geometric eigenmodes—represent a more fundamental anatomical constraint on dynamics than the connectome[7,8,25]. This view stands in stark contrast to the classical view that complex patterns of interregional anatomical connectivity shape brain activity[26].

Here we test these competing views of the brain with the aim of identifying the principal structural constraints on human brain dynamics. In line with theoretical predictions from NFT, we show that diverse experimental fMRI data from spontaneous and task-evoked recordings in the human neocortex can be explained more parsimoniously by eigenmodes derived from cortical geometry (geometric eigenmodes) than by those obtained from measures of brain connectivity (connectome eigenmodes). We further confirm that stimulus-evoked activity is dominated by excitations of geometric eigenmodes with long spatial wavelengths, challenging classical views that such activity is localized to focal, spatially isolated clusters. To directly link these structural constraints to the physical processes driving brain dynamics, we use a generative model to show how wave dynamics unfolding on the geometry of the cortex can explain diverse features of functional brain organization. Finally, we show that the close relationship between geometry and function captured by eigenmodes extends to non-neocortical structures, indicating that this link is a universal property of brain organization.

## Geometric modes constrain cortical activity

We first examine the degree to which geometric eigenmodes can explain diverse aspects of human neocortical activity. To derive the eigenmodes, we use a mesh representation of a population-averaged template of the neocortical surface (Fig. 1a and Derivation of cortical geometric eigenmodes in Methods). We then construct the Laplace–Beltrami operator (LBO) from this surface mesh, which captures local vertex-to-vertex spatial relations and curvature, and solve the eigenvalue problem,

$$\nabla^2 \psi = \Delta \psi = -\lambda \psi, \qquad (1)$$

where $\nabla$ is the gradient operator, $\Delta$ is the LBO and $\psi = \{\psi_1(\mathbf{r}), \psi_2(\mathbf{r}), \dots\}$ is the family of geometric eigenmodes with the corresponding family of eigenvalues $\lambda = \{\lambda_1, \lambda_2, \dots\}$. The eigenvalues are ordered sequentially according to the spatial frequency or wavelength of the spatial patterns of each mode (Fig. 1a and Extended Data Fig. 1), such that $\psi_1$ is the mode with the longest wavelength. The resulting eigenmodes are orthogonal, forming a complete basis set to decompose spatiotemporal dynamics unfolding on the cortex as a weighted sum of modes with varying wavelengths (Fig. 1b and Modal decomposition of brain activity in Methods). Unless otherwise specified, we use $N = 200$ modes throughout this study.

Using this decomposition we evaluate the accuracy of geometric eigenmodes in capturing both task-evoked and spontaneous brain activity (Fig. 1c) measured in 255 healthy individuals from the Human Connectome Project[27] (HCP; HCP data in Methods and Supplementary Information 2). For task-evoked activity, we map 47 task-based contrasts drawn from seven different tasks representing distinct evoked activation patterns. We then reconstruct each individual's activation map using an increasing number of modes up to a maximum of 200 (Fig. 1d). For spontaneous, task-free (so-called resting-state) activity, we reconstruct the spatial map of activity at each time frame and then generate a region-to-region functional coupling (FC) matrix, describing correlations of activity among 180 discrete brain regions per hemisphere[28]. To allow direct comparison between task-evoked and spontaneous recordings, we apply the same regional parcellation to the task-evoked data (Cortical parcellations in Methods). Finally, we quantify reconstruction accuracy by calculating the correlation between empirical and reconstructed task-evoked activation maps and spontaneous FC matrices (Fig. 1d–f).

We observe that reconstruction accuracy increases with an increasing number of modes across all task contrasts and in the resting state, with $r \geq 0.38$ already achieved using just $N = 10$ modes (Fig. 1d). Large-scale modes are also differentially recruited across different tasks, suggesting that particular stimuli excite specific modes (Fig. 1e). Improvements in reconstruction accuracy become slow after ten modes, reaching $r \geq 0.80$ at approximately $N = 100$ modes, with only incremental increases in reconstruction accuracy beyond this point. Beause the first 100 modes have wavelengths above around 40 mm (Supplementary Table 1), and the inclusion of shorter-wavelength modes only refines reconstruction of localized patterns (arrowheads in Fig. 1e), our findings suggest that the data are predominantly comprised of spatial patterns with long spatial wavelengths (see next section for a more detailed analysis).

These results are consistent across all 47 HCP task contrasts (Supplementary Fig. 1) and parcellations of varying resolutions (Supplementary Fig. 2), but data parcellated at higher resolution require more modes to achieve high reconstruction accuracy due to the low-pass spatial filtering effect of coarser parcellations. Our results are also not affected by the use of a population-averaged cortical surface template (rather than individual-specific surfaces) to derive the geometric eigenmodes (Supplementary Figs. 3–5 and Supplementary Information 3). Together these findings indicate that cortical geometric eigenmodes form a compact representation that captures diverse aspects of task-evoked and spontaneous cortical activity. Moreover, they show that such activity is dominated by long-wavelength, large-scale eigenmodes.

We next test the hypothesis that geometric eigenmodes provide a more parsimonious and fundamental description of dynamics than eigenmodes derived from a graph-based connectome approximation. To this end we compare the reconstruction accuracy of geometric eigenmodes against three alternative connectome-derived eigenmode basis sets (see Fig. 2a for a schematic). The first basis set is derived empirically from a connectome mapped with dMRI tractography at vertex resolution and thresholded to obtain a connection density of

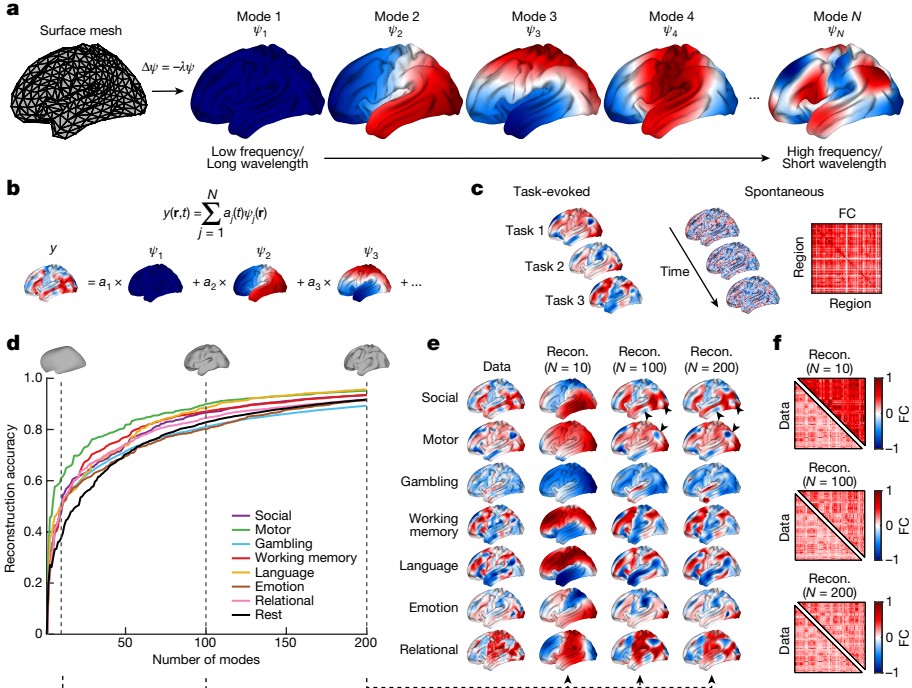

**Fig. 1 | Reconstruction of neocortical activity with geometric eigenmodes.**
**a**, Geometric eigenmodes are derived from the cortical surface mesh by solving the eigenvalue problem, $\Delta\psi = -\lambda\psi$ (equation (1)). The modes $\psi_1$, $\psi_2$, $\psi_3$, ..., $\psi_N$ are ordered from low to high spatial frequency (long to short spatial wavelengths). Negative, zero and positive values are coloured blue, white and red, respectively.
**b**, Modal decomposition of brain activity data. The example shows how a spatial map, $y(\mathbf{r}, t)$, at a given time, $t$, can be decomposed as a sum of modes, $\psi_j$, weighted by $a_j$. **c**, Left, we reconstruct task-evoked data using spatial maps of activation for a diverse range of stimulus contrasts. Right, we reconstruct spontaneous activity by decomposing the spatial map at each time frame and generating a region-to-region FC matrix. **d**, Reconstruction accuracy of seven key HCP task-contrast maps (Supplementary Information 2.1) and resting-state FC as a function of the number of modes. Insets show cortical surface reconstructions, demonstrating the spatial scales relevant to the first 10, 100 and 200 modes corresponding to spatial wavelengths of approximately 120, 40 and 30 mm, respectively. **e**, Group-averaged empirical task-activation maps and reconstructions (recon.) obtained using 10, 100 and 200 modes of the seven key HCP task contrasts. Black arrowheads indicate localized activation patterns that are more accurately reconstructed when using short-wavelength modes. **f**, Group-averaged empirical resting-state FC matrices and reconstructions using 10, 100 and 200 modes.

0.10%, as done previously[29] (Derivation of connectome eigenmodes in Methods). The second basis set is derived from a connectome constructed synthetically according to a homogeneous stochastic wiring process governed by an exponential distance-dependent connection probability to mimic simple, EDR-like connectivity (Derivation of EDR eigenmodes in Methods). Because the connection densities of empirical and EDR connectomes differed, we evaluated a third basis set derived from the empirical connectome thresholded at 1.55% to match the density of the EDR connectome. The connectome, EDR and density-matched connectome eigenmodes described above are derived from the graph Laplacian (a discrete counterpart of the LBO) of their respective connectivity matrices (Fig. 2b and Extended Data Fig. 1).

To summarize, geometric eigenmodes account for the intrinsic curvature of the cortical surface and local vertex-to-vertex relations in the surface mesh; connectome eigenmodes do not consider curvature but capture local spatial relations between mesh vertices, along with short- and long-range connections measured with dMRI; and EDR eigenmodes account for the effect of a homogeneous, stochastic, distance-dependent connection rule without fully capturing the cortical geometry (Fig. 2a). Contrasting these different basis sets thus allows us to disentangle the contributions to brain dynamics of cortical geometry from structural connectivity.

Direct comparison of the reconstruction accuracy of these different basis sets shows that geometric eigenmodes consistently show the highest reconstruction accuracy across both spontaneous (Fig. 2c) and task-evoked (Fig. 2d) data. EDR eigenmodes perform nearly as well as geometric eigenmodes whereas connectome eigenmodes are the

least accurate. This finding holds true regardless of the parcellation used (Extended Data Figs. 2 and 3), the specific connection density used to generate the connectome eigenmodes (Supplementary Figs. 6 and 7 and Supplementary Information 4) and whether we generate the connectome using a discrete regional parcellation rather than at vertex resolution (Supplementary Fig. 8 and Supplementary Information 4). We additionally find that geometric eigenmodes show stronger out-of-sample generalization than principal components of the functional data themselves (calculated via principal component analysis (PCA); Supplementary Fig. 9, Extended Data Fig. 4 and Supplementary Information 5) and better performance than Fourier spatial basis sets (Extended Data Fig. 5, Supplementary Information 6 and Comparisons with statistical basis sets in Methods).

Taken together, these results demonstrate the parsimony, robustness and generality of geometric eigenmodes as a basis set for brain function. They also support the prediction of NFT that brain activity is best represented in terms of eigenmodes derived directly from the shape of the cortex, thus emphasizing a fundamental role of geometry in constraining dynamics.

## Long wavelengths dominate cortical activity

Reconstructions of both spontaneous and task-evoked data with geometric eigenmodes show that the spatial organization of brain activity is dominated by patterns with spatial wavelengths of about 40 mm or longer (Fig. 1d–f). This result counters classical approaches to analysis of neuroimaging data, in which stimulus-evoked activations are

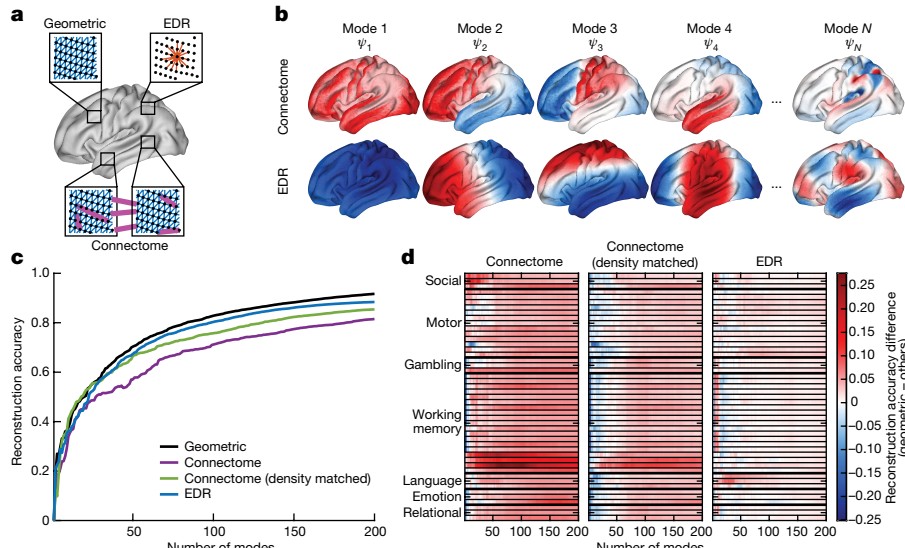

**Fig. 2 | Geometric eigenmodes benchmarked against connectome-based eigenmodes. a**, Schematic of the anatomical properties used to derive eigenmodes for cortical geometry, the connectome and the EDR connectome. Geometric eigenmodes rely on local surface mesh information such as links (blue) between neighbouring surface mesh vertices (dots) and curvature. Connectome eigenmodes rely on local links between mesh vertices (blue) and short- and long-range connections (magenta) reconstructed empirically from dMRI. EDR eigenmodes rely on connections (red) generated from a stochastic wiring process in which the probability of connection between vertices exponentially decays as a function of their distance. **b**, Example connectome and EDR eigenmodes. Negative, zero and positive values are coloured blue, white and red, respectively. Despite some similarities, the spatial patterns of the modes are distinct from those derived using cortical geometry (compare with Fig. 1a). **c**, Reconstruction accuracy of resting-state FC matrices achieved

by geometric, EDR and two variants of connectome eigenmodes: one using a connectome as defined using previous methods[29] and the other with the same connection density as the EDR connectome to allow fair comparison (for other densities see Supplementary Figs. 6 and 7). **d**, Difference in reconstruction accuracy of all 47 HCP task-contrast maps achieved by geometric eigenmodes and the other basis sets, as indicated by the text above each panel. Each row represents a different task contrast, grouped here by broad types (Supplementary Information 2.1); red indicates superior performance for geometric eigenmodes. Note that while there seems to be a performance advantage for connectome eigenmodes for reconstructions incorporating fewer than ten modes relative to geometric eigenmodes, reconstruction accuracy is generally low (average $r = 0.42$ across the different tasks) compared with that for 100 modes (average $r = 0.71$).

---

mapped by thresholding statistical maps to identify focal, isolated areas of heightened activity. This classical approach rests on the assumption that focal loci represent discrete brain regions putatively engaged by the stimulus and that subthreshold activity in other regions is of negligible interest. The surprisingly long-wavelength content of task-activation data (Fig. 1d–e) suggests that classical procedures focus only on the tips of the iceberg and obscure the underlying spatially extended and structured patterns of activity evoked by the task (see Extended Data Fig. 6 for an explanation of the reasons involved). These observations accord with the theoretical predictions of NFT and previous analyses of task-evoked electroencephalography (EEG) signals[30,31].

Here we leverage the modal decomposition described in Fig. 1b to characterize the complete spatial pattern—the entire iceberg—of task-evoked activation. To this end we analyse the spatial power spectrum obtained using a geometric mode decomposition of group-averaged unthresholded activation maps from the 47 task contrasts in HCP[27,32] (Modal power spectra of task-evoked activation maps in Methods). As an independent replication, we also analyse 10,000 unthresholded activation maps from 1,178 independent experiments available in the NeuroVault repository[33], thus providing a comprehensive picture of the diversity of stimulus-evoked activation patterns mapped in the human brain.

Despite the wide range of stimuli, paradigms and data-processing approaches used to acquire these activation maps, we observe that a large fraction of power in the maps is concentrated in the first 50 modes, corresponding to spatial wavelengths greater than around 60 mm (Fig. 3a; similar results are found separately for each of the key HCP task-contrast maps; Extended Data Fig. 7). Using surrogate data, we confirm that these findings cannot be explained by the spatial smoothing

induced by typical fMRI processing pipelines, which can filter out short-wavelength spatial patterns of activity (Extended Data Fig. 8 and Supplementary Information 7). We further observe that incremental, sequential removal of long-wavelength modes has a much greater impact on reconstruction accuracy than removal of short-wavelength modes (Fig. 3b and Contributions of long- and short-wavelength modes in Methods). For instance, across the seven key HCP task contrasts, removal of the top 25% long-wavelength modes (modes 1–50) yields a drop in reconstruction accuracy of around 40–60% whereas removal of the top 25% short-wavelength modes (modes 151–200) yields a drop of only around 2–4% (Fig. 3b, insets). These results indicate that, on temporal and spatial scales accessible with fMRI, evoked cortical activity comprises large-scale, nearly brain-wide spatial patterns, challenging classical views that such activity should be described in terms of discrete, isolated and anatomically localized activation clusters.

## Wave dynamics bridge geometry and function

Geometric eigenmodes of the cortex are obtained by solving the eigenvalue problem of the LBO, which is also known as the Helmholtz equation (equation (1)). In physically continuous systems, the solutions of the Helmholtz equation correspond to the spatial projections of the solutions of a more general wave equation, such that the resulting eigenmodes inherently represent the vibrational patterns, or standing waves, of the system's dynamics[34]. This equivalence implies that the superior efficacy of geometric eigenmodes in the reconstruction of diverse patterns of brain activity results from a fundamental role of wave dynamics in shaping these patterns, as predicted by NFT. This prediction has been confirmed through models of EEG recordings[21,35],

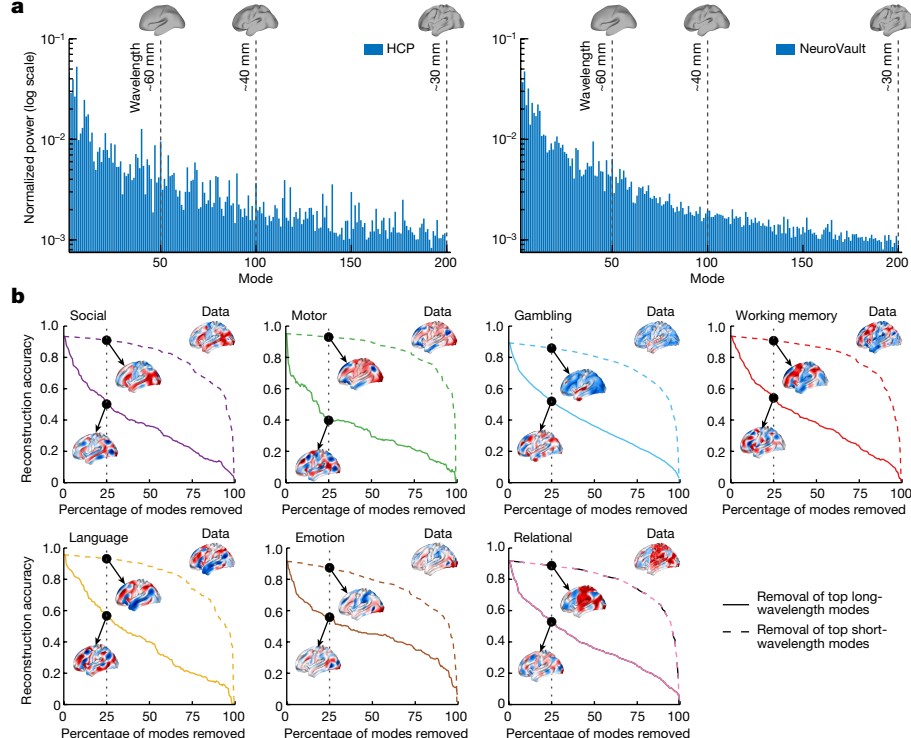

**Fig. 3 | Task-evoked activity excites long-wavelength modes. a**, Normalized mean power spectra of the 47 HCP task-contrast maps (left) and 10,000 contrast maps from the NeuroVault database (right). Insets show cortical surface reconstructions demonstrating spatial scales relevant to the first 50, 100 and 200 modes corresponding to spatial wavelengths of approximately 60, 40 and 30 mm, respectively. Contrast-specific spectra for the seven key HCP task contrasts are presented in Extended Data Fig. 7. **b**, Reconstruction accuracy of the seven key HCP task-contrast maps as a function of the percentage of modes (among 200) removed in the reconstruction process. Solid and dashed lines correspond to removal of the top long- and short-wavelength modes, respectively. Insets show group-averaged empirical activation maps (data) and their reconstructions after removal of 25% of modes. Negative, zero and positive values are coloured blue, white and red, respectively.

but waves across the whole brain have only recently been observed in fMRI signals[36,37] and thus far lack a theoretical explanation. Here we use NFT and geometric eigenmodes to show that wave dynamics can provide a unifying account of diverse empirical and physiological phenomena observed at scales accessible with fMRI.

We model neural activity using an isotropic damped NFT wave equation without regeneration[6] (Fig. 4a and NFT wave model in Methods). Under this model, activity propagates between points on the neocortex through their white-matter connectivity with a strength that decays approximately exponentially with distance (Supplementary Fig. 10 and Supplementary Information 1 and 8). To simulate resting-state neural activity we use a white noise input to mimic unstructured stochastic fluctuations[21] (Modelling resting-state dynamics in Methods). We compare the performance of this simple wave model with a biophysically based neural mass model (balanced excitation–inhibition (BEI) model) that has been used extensively to understand resting-state fMRI signals[38] (Fig. 4a and Neural mass model in Methods). The neural mass model is closely aligned with the classical, connectome-centric view of brain function, representing dynamics as the result of interactions between neuronal populations in discrete anatomical regions, coupled according to an empirically measured connectome.

We first compare the efficacy of the two models in capturing distinct and commonly studied properties of spontaneous, task-free FC: namely, static pairwise FC (edge FC), static node-level average FC (node FC) and time-resolved dynamic properties of FC (FCD) (Modelling resting-state dynamics in Methods). Across all FC-based benchmark measures, the wave model shows comparable or superior performance in reconstruction of empirical data relative to the neural mass model (Fig. 4b). The wave model also captures time-lagged properties[36,37,39]

of empirical resting-state activity more accurately than the mass model (Extended Data Fig. 9 and Measuring time-lagged properties of resting-state dynamics in Methods). This strong performance of the wave model is remarkable given its relative simplicity: the wave model only requires the geometry of the cortex (that is, the surface mesh) as input and includes one fixed parameter and one free parameter ($r_s$) for fitting to data (Extended Data Fig. 10) whereas the neural mass model requires a dMRI-derived interregional anatomical connectivity matrix and comprises 15 fixed parameters and four free parameters (Supplementary Information 9). These considerations indicate that wave dynamics offer a more accurate and parsimonious mechanistic account of macroscale, spontaneous cortical dynamics captured by fMRI.

We next consider stimulus-evoked cortical activity in the wave model. We analyse cortical responses to sensory stimulation of primary visual cortex (V1), because it elicits a well-defined hierarchy of regional cortical responses[40,41] (Modelling stimulus-evoked dynamics in Methods). A 1 ms pulse input to V1 yields a propagating wave of activity that rapidly splits along the dorsal and ventral visual processing streams (Fig. 4c (arrows) and Supplementary Video 1), consistent with the mainstream understanding of hierarchical visual processing[42]. Remarkably, this result indicates that geometric constraints on travelling waves of evoked activity are sufficient for the segregation of the dorsal and ventral processing streams, which have traditionally been thought to be driven primarily by complex patterns of layer-specific connectivity[40,42,43]. Furthermore, the temporal profile of evoked responses across the visual system follows a well-defined timescale hierarchy, with higher-order association areas showing peak responses that are delayed and prolonged compared with lower-order visual areas (Fig. 4d). These findings thus indicate that this hierarchical ordering,

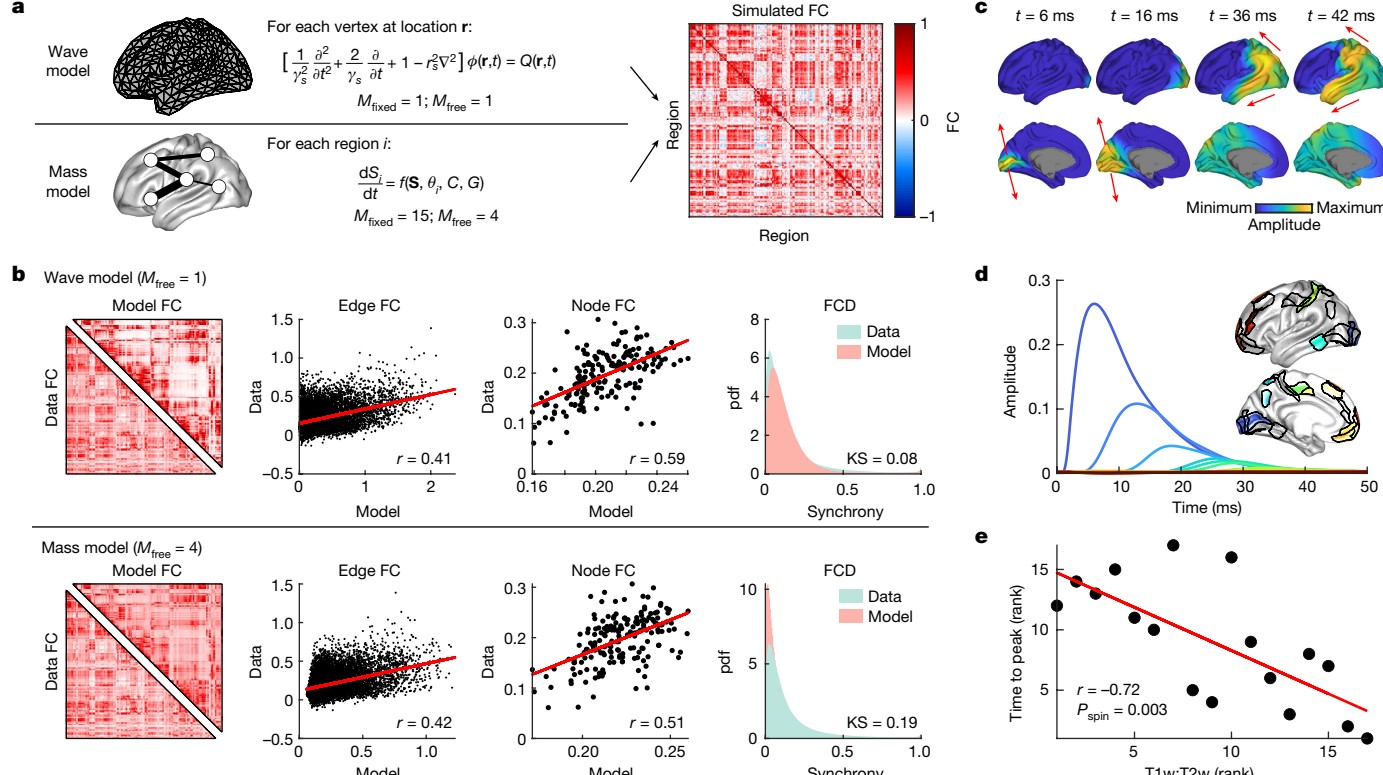

**Fig. 4 | Wave dynamics shape patterns of spontaneous and stimulus-evoked activity. a**, For the wave model, the activity $\phi(\mathbf{r}, t)$ at location $\mathbf{r}$ and time $t$ is governed by a wave equation with damping rate $\gamma_s$, spatial length scale $r_s$ and input $Q$. For the neural mass model the activity $S_i(t)$ of region $i$ is described by the function $f$, which depends on the activity of other regions $S$, local population parameters $\theta_i$ and connectome $C$ scaled by the global coupling parameter $G$. The model dynamics are used to calculate a simulated FC matrix (Methods). $M_{fixed}$ and $M_{free}$ correspond to the number of fixed and free parameters of each model, respectively. **b**, Comparison of data and model simulations based on various metrics which, from left to right, are: FC matrix (for visual purposes), edge FC, node FC and FCD. For edge FC and node FC, red lines represent linear fits with Pearson correlation coefficient $r$; for FCD, the probability density function (pdf) of the similarity of interregional synchrony in data and model dynamics is compared using the Kolmogorov–Smirnov (KS) statistic. **c**, Wave propagation of activity after 1 ms stimulation of V1 for $t = 1$–2 ms. Arrows indicate the direction of propagation (Supplementary Video 1). **d**, Activity profile of 17 regions in the visual cortical hierarchy. Insets show spatial locations of regions on the cortical surface coloured according to their activation profiles. **e**, Relationship of ranked activity profile time to peak and ranked T1w:T2w value of the regions in **d**. Red line represents a linear fit of the ranked variables, with Spearman correlation coefficient $r$ and one-sided $P_{spin}$, estimated from 10,000 permutations.

previously identified in experimental and modelling studies[41,44,45], emerges naturally from waves of excitation propagating through the cortical medium. Critically, this hierarchical temporal ordering of areal responses strongly correlates with an independent anatomical measure of the cortical processing hierarchy based on non-invasive estimates of myeloarchitecture (T1-weighted (T1w) and T2-weighted (T2w) ratio)[46,47]. This correlation is particularly strong within the visual processing hierarchy ($r = -0.72$, one-sided spin-test $P$ value ($P_{spin}$) = 0.003; Fig. 4e) but is also present when considering all cortical areas ($r = -0.44$, $P_{spin} = 0.037$; Supplementary Fig. 11). Together, our modelling results show how simple wave dynamics unfolding on the geometry of the cortex provide a unifying generative mechanism for capturing complex properties of spatiotemporal brain activity.

## Geometry constrains subcortical activity

Our analyses thus far have focused on the strong coupling of geometry and dynamics in the neocortex. We next investigate this coupling in non-neocortical areas, focusing on the thalamus, striatum and hippocampus, because these structures have geometries easily captured using MRI data and their functional organization has been extensively studied[48].

We first generalize our eigenmode analysis to three-dimensional (3D) volumes (Estimating the geometric eigenmodes of non-neocortical structures in Methods), yielding geometric eigenmodes that extend spatially through the three spatial dimensions of each structure. Next, to fully capture the macroscale functional organization of these non-neocortical regions, we apply a widely used manifold learning procedure to voxel-wise FC data to obtain the key functional gradients in each structure[49] (Mapping the functional organization of non-neocortical structures in Methods). These functional gradients describe the principal axes of spatial organization dictated by similarities in FC, thus representing the dominant modes of variation in functional organization, ordered according to the percentage of variance in FC similarity that they explain.

The spatial profiles of the first three functional gradients of the thalamus, striatum and hippocampus (accounting for 24, 50 and 47% of the variance in FC similarity, respectively) show a near-perfect match to the first three geometric eigenmodes (Fig. 5a–c; spatial correlations $r \geq 0.93$). This tight correspondence generalizes out to the first 20 modes and first 20 gradients of each structure (with the first 20 gradients respectively accounting for 49, 70 and 68% of total variance in FC similarity), with all absolute spatial correlations $|r| > 0.5$, except for the 20th gradient and 20th mode in the striatum and hippocampus (Fig. 5d–f). This strong relationship is striking given that the functional gradients are generated via a complex processing pipeline applied to fMRI-derived FC measures whereas the eigenmodes are derived simply from each structure's geometry, independent of the

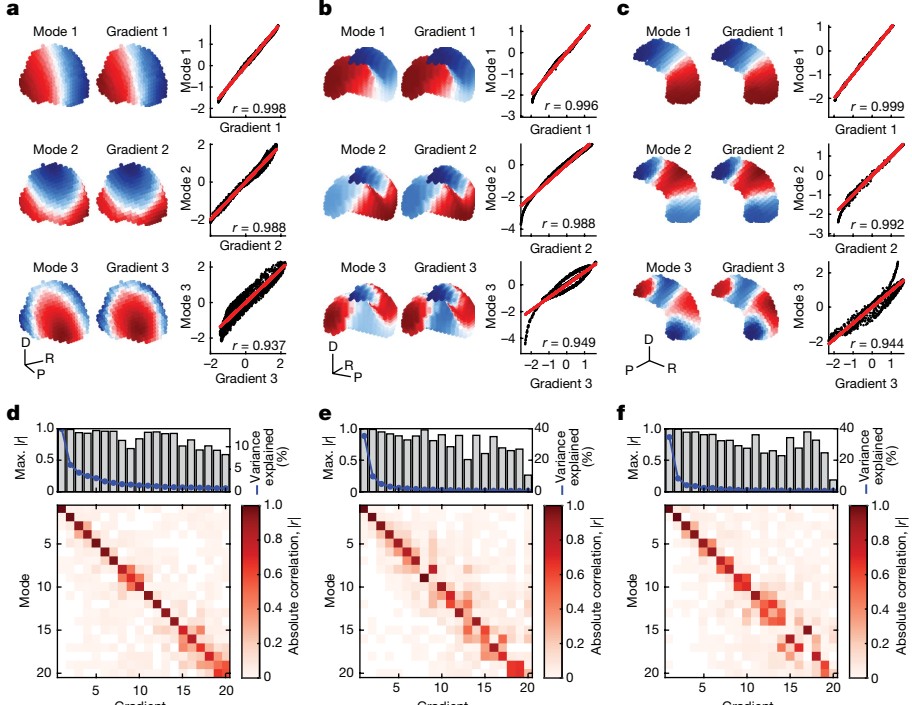

**Fig. 5 | Geometry shapes non-neocortical function. a–c**, First three geometric eigenmodes and FC-based functional gradients in the thalamus (**a**), striatum (**b**) and hippocampus (**c**). The modes and gradients are shown in 3D coordinate space with negative, zero and positive values coloured blue, white and red, respectively. D, dorsal; P, posterior; R, rightward. Scatter plots show the relationship between modes and gradients, with red lines representing linear fits with Pearson correlation coefficient *r*. **d–f**, Bottom, absolute correlation (|*r*|) of the first 20 geometric eigenmodes and functional gradients in the thalamus (**d**), striatum (**e**) and hippocampus (**f**). Top, highest |*r*| value obtained for each functional gradient (grey bars), taking into account order flips in geometric eigenmodes, along with the percentage of variance explained by each functional gradient (blue lines). Max., maximum.

functional data. These findings suggest that the functional organization of non-neocortical structures derives directly from their geometric eigenmodes.

## Discussion

The dynamics of many physical systems are constrained by their geometry and can be understood as excitations of a relatively small number of structural modes[12]. Here we show that structural eigenmodes derived solely from the brain's geometry provide a more compact, accurate and parsimonious representation of its macroscale activity than alternative connectome-based models. This mode-based view of the brain further shows that both spontaneous and evoked brain activity captured by fMRI are dominated by large-scale eigenmodes with relatively long wavelengths, whose dynamics are derived from a biophysically motivated wave equation. These findings challenge the classical neuroscientific paradigm in which topologically complex patterns of interregional connectivity between discrete, specialized neuronal populations are viewed as a critical anatomical foundation for dynamics. Instead, our results indicate that a physically grounded approach that treats the brain as a continuous, spatially embedded system offers a unifying framework for understanding structural constraints on diverse aspects of macroscopic neuronal function.

The extensive comparisons of geometric eigenmodes with other anatomical (connectome and EDR eigenmodes) and statistical (PCA and Fourier) basis sets show that the superior performance of geometric eigenmodes in capturing macroscale neocortical activity is not trivially driven by generic mathematical properties of basis set expansions. Rather, this result indicates that geometry represents a fundamental anatomical constraint on dynamics. Additionally, the strong performance of the EDR eigenmodes derived from a synthetic network suggests that a homogeneous, distance-dependent connectivity with near-exponential form represents another important anatomical constraint on activity. EDR-like connectivity is mathematically embedded within the Helmholtz equation in equation (1)[6,7] (Supplementary Information 8), so the role of such connectivity is implicitly captured by the geometric eigenmodes.

The comparatively poor performance of connectome eigenmodes indicates that topologically complex connections that exist beyond a simple EDR afford minimal further benefit in obtaining eigenmodes that can accurately explain spatiotemporal patterns of cortical activity as measured with fMRI. Our findings thus counter traditional views that emphasize intricate patterns of anatomical connections as the primary driver of coordinated dynamics[26,50–52]. Indeed, recent estimates indicate that long-range cortical connections are rare[53]—they may therefore represent a relatively minor perturbation of the dominant effect imposed by EDR-like connectivity. Nonetheless the topological centrality, metabolic cost and tight genetic control of such connections[54,55] suggest that they provide important functional and evolutionary advantages beyond wave-like dynamics[56] (Supplementary Information 11). The limited resolution and sensitivity to preprocessing pipelines of dMRI and fMRI data[57,58] complicate attempts to fully uncover the functional role of these connections. High-quality animal tract-tracing and electrophysiological data may be helpful in this regard.

The close coupling between geometry and dynamics is apparent in neocortical and non-neocortical structures alike, suggesting that the functional organization of regions outside the neocortex is also dominated by distance-dependent anatomical connectivity and wave dynamics, as found in recent experiments[36,59,60]. These observations indicate that geometric eigenmodes offer a simpler, more parsimonious and mechanistically informative account of putative gradients of functional organization in non-neocortical structures than the complex manifold learning procedures currently used in the literature[49]. This is because such procedures are phenomenological, providing statistical

descriptions of dominant sources of variances in the data, whereas the study of structural eigenmodes derives from a generative process.

Geometric mode decomposition offers unique insights into the spatial properties of brain activation maps. Classical brain-mapping analyses focus on responses in isolated clusters of spatial locations that exceed a statistical threshold[61]. By contrast, our approach aligns with rigorously established results from physics and engineering in which perturbations of spatially continuous systems elicit system-wide responses, just as the musical notes of a violin string result from vibrations across its entire length rather than from the behaviour of a restricted segment[62]. Accordingly, the use of geometric eigenmodes indicates that, across more than 10,000 diverse maps from task-based fMRI studies, task engagement is associated predominantly with the excitation of modes with wavelengths of roughly 60 mm and longer. This result coincides with similar observations of long-wavelength excitations in empirical EEG and evoked response potential data[30,31] and suggests that classical analyses reliant on thresholding of pointwise statistical maps obscure the spatially extended and complex patterns of activity actually evoked by a task.

Our modelling results offer insight into the physical processes that underlie the close link observed between geometry and function. In particular, the relative simplicity and superior performance of the wave model in capturing diverse aspects of spontaneous fMRI dynamics indicate that the model provides a more parsimonious account than a complex neural mass model, which treats the brain as a graph of discrete anatomical regions (nodes) coupled via the connectome (edges). This finding is consistent with experimental observations of wave dynamics in both human and animal fMRI data[63,64]. Future work could explore whether the introduction of spatial heterogeneities[65] or complex structured input[66] into the wave model further improves its accuracy in explaining diverse empirical phenomena.

Application of the wave model to mimic visual stimulation shows that waves propagating from the stimulation site segregate along the classical dorsal and ventral visual pathways, and that regional responses to the perturbation conform to a well-described hierarchy of timescales that range from rapidly responding unimodal areas to slower transmodal regions[41,44,45]. These canonical properties of hierarchical visual processing have been extensively studied for decades and are conventionally thought to be driven by complex patterns of layer-specific interregional connectivity[40,42,43], but our analysis shows that waves travelling through the cortical geometry are sufficient for the emergence of segregated, hierarchical processing streams. Thus, while our findings cannot rule out a role for complex interregional connectivity they do indicate that such connectivity is not necessary for the emergence of these macroscale dynamics.

The superior performance of geometric eigenmodes offers an immediate practical benefit because the modes can be estimated using only a mesh representation of the structure of interest, which can easily be derived using well-established, automated processing pipelines for T1w anatomical images[67]. By contrast, connectome eigenmodes require a graph-based model of macroscopic interregional connectivity generated via complex data-processing pipelines applied to both T1w and dMRI images[58,68]; the definition of graph nodes, which is a topic of contention[69]; and the application of a thresholding procedure to remove putatively spurious connections, which our own analysis shows can affect the findings (Supplementary Figs. 6 and 7). The fact that such choices are not required to obtain the geometric eigenmodes means that they can be applied robustly and flexibly across different experimental contexts in both humans and other species[70,71], opening new avenues of research. For example, one can investigate how geometric eigenmodes vary through neurodevelopment or are disrupted in clinical disorders. Indeed, the close link we identify between geometry and function implies that interspecies differences in spatiotemporal dynamics may largely be driven by differences in brain shape. The characterization of how variations in brain geometry, both within and between species, shape brain function will be essential for understanding physical and anatomical constraints on neuronal activity.

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

# Methods

## Derivation of cortical geometric eigenmodes

If brain structure can be approximated as being constant in time, the resulting spatial and temporal dynamics can be treated separately via eigenmode decomposition[1,7], similar to the treatment of other physical systems[12]. In particular, the spatial aspect satisfies the Laplacian eigenvalue problem, which is also known as the Helmholtz equation, defined in equation (1).

For the cerebral cortex, which we consider as a two-dimensional (2D) model embedded within 3D Euclidean space, the LBO in equation (1) captures intrinsic geometry, which includes the curvature of the cortical surface[72] and is defined generally as[73,74],

$$\Delta := \frac{1}{W} \sum_{i,j} \frac{\partial}{\partial x_i}\left( g^{ij} W \frac{\partial}{\partial x_j} \right), \tag{2}$$

where $x_i, x_j$ are the local coordinates, $g^{ij}$ is the inverse of the inner product metric tensor $g_{ij} := \langle \frac{\partial}{\partial x_i}, \frac{\partial}{\partial x_j} \rangle$, $W := \sqrt{\det(G)}$, det denotes the determinant and $G := (g_{ij})$.

We employed the LaPy python library[72,75] installed in the MASSIVE high-performance computing facility[76] to derive the geometric eigenmodes of the human cortex. Specifically, we used a triangular surface mesh representation of the midthickness human cortical surface, comprising 32,492 vertices in each hemisphere, obtained from a downsampled, left–right symmetric version of the FreeSurfer's fsaverage population-averaged template[77] (https://github.com/ThomasYeoLab/CBIG/tree/master/data/templates/surface/fs_LR_32k). This template is independent of the data sample used in all our analyses, thus obviating any concerns about circularity.

Note that the continuous LBO operates on the underlying Riemannian manifold of the surface and not directly on mesh vertices. LaPy uses the cubic finite element method on the surface mesh to achieve numerically tractable solutions of equation (1) on an interpolated smooth manifold. This distinguishes it from the discrete graph Laplacian[78], which does not encode spatial relations between points. All our analyses were focused on unihemispheric eigenmodes, but our approach can easily be extended to the whole brain because bihemispheric eigenmodes can be represented as symmetric or antisymmetric combinations of the eigenmodes derived from each hemisphere[7]; symmetric combinations correspond to mirror symmetry across the sagittal midplane and asymmetric combinations correspond to cases in which the hemispheres have the same spatial structure but with flipped signs.

The eigenvalue solutions of equation (1) are ordered sequentially according to the spatial frequency or wavelength of the spatial patterns of each eigenmode—that is, $0 \leq \lambda_1 \leq \lambda_2 \leq \ldots$. Note that the first eigenvalue, $\lambda_1$, is approximately equal to zero (wavelength much greater than brain size) and the corresponding eigenmode, $\psi_1$, is a constant function with no nodal lines (zero sets of the function). Throughout our study, we used the first 200 modes (including the constant mode, $\psi_1$) in our analyses given the diminishing improvements in reconstruction accuracy observed when using an increasing number of modes (Fig. 1d).

Each eigenmode comprises spatial patterns with a specific spatial wavelength. Following ref. 7, we approximate the eigenmode wavelengths using an idealized spherical case because it is topologically comparable to the human cortex. By solving equation (1) on a sphere, degenerate solutions exist such that certain eigenmodes have the same eigenvalue and spatial wavelength—this is analogous to spherical harmonics in quantum physics. In fact, because the eigenmodes will approach the spherical harmonics in the limit of vanishing cortical folding[7], the former can be grouped together into an eigengroup with spatial wavelength[71],

$$\text{wavelength} = \frac{2\pi R_s}{\sqrt{l(l+1)}}, \tag{3}$$

where $R_s$ is the radius of the sphere (for the fsaverage population-averaged template used in this study, $R_s \approx 67.0$ mm) and $l$ is the eigengroup number (the angular momentum quantum number in atomic physics). The wavelengths of the first 15 eigengroups and the eigenmodes included in the eigengroup are shown in Supplementary Table 1.

## Modal decomposition of brain activity

We used the geometric eigenmodes to decompose spatiotemporal fMRI data, measured at spatial location $\mathbf{r}$ and time $t$, for each individual, as a weighted sum of modes,

$$y(\mathbf{r}, t) = \sum_{j=1}^{N} a_j(t) \psi_j(\mathbf{r}), \tag{4}$$

where $a_j$ is the amplitude of mode $j$ in explaining the data, $\psi_j$ is the $j$th mode and $N$ is the number of modes used; we used $N = 200$ for our analyses. For spatiotemporal data—that is, recordings of spontaneous dynamics from task-free fMRI—each time frame of the data was substituted into equation (4), resulting in a time-dependent amplitude $a_j(t)$ for each mode $\psi_j$. For purely spatial data—that is, task-evoked activation maps—amplitudes are independent of time such that $a_j(t) \rightarrow a_j$. In both cases, the amplitudes can be obtained by integrating over the cortical surface,

$$a_j(t) = \int y(\mathbf{r}, t) \psi_j(\mathbf{r}) d\mathbf{r}, \tag{5}$$

which can be derived from equation (4) using the orthogonal property of the eigenmodes[62,79]. If there are insufficient measurements to evaluate the integral, the amplitudes can also be estimated via a statistical general linear model.

After obtaining the amplitudes, equation (4) was used to calculate the reconstructed data. We quantified the accuracy of this reconstruction by calculating the correlation between empirical and reconstructed data. For the spatiotemporal task-free data, we first parcellated the empirical and reconstructed data by taking the average of the data within discrete parcels/regions as per standard practice in the field[80], and then constructed a matrix of interregional FC by calculating the Pearson correlation coefficient of pairs of parcel time series. For task-evoked data, we applied the same parcellation on the activation maps to allow direct comparison. Finally, reconstruction accuracy for task-free data was calculated by taking the correlation of the upper triangular elements of the empirical and reconstructed FCs. For task-evoked data, reconstruction accuracy was calculated from the spatial correlation of parcellated empirical and reconstructed maps. We then took the average reconstruction accuracy across all participants.

## HCP data

We used preprocessed fMRI data from HCP[27]. We did not perform any additional preprocessing steps, such as global signal removal, because the first eigenmode (considered as the global, constant mode) already explicitly captures global deviations in the data, allowing the other modes to capture functionally relevant non-global activity. We analysed data from 255 unrelated healthy individuals (aged 22–35 years, 132 females and 123 males), which is the largest HCP sample excluding twins or siblings, and with all participants having completed task-evoked and task-free resting-state data. All participants were volunteers and provided informed consent. The open-access data were acquired by the WU–Minn HCP consortium with local overseeing ethics committee approval and were shared with the authors according to HCP's data use terms. All our procedures were carried out in accordance with protocols set by these data use terms. For a detailed account of the image acquisition protocol, preprocessing pipelines and ethics oversight, see refs. 27,32.

Within the HCP dataset we analysed task-evoked fMRI measured in seven task domains (Supplementary Table 2) and task-free

resting-state fMRI, which were already preprocessed by HCP. Both datasets were mapped onto the fsLR-32k CIFTI space with 32,492 vertices in each hemisphere. Further details on data and preprocessing can be found in Supplementary Information 2.1 and 2.2. We also analysed individual connectomes derived from dMRI data, as provided in ref. 81. The connectomes represent a high-resolution weighted matrix of size 32,492 × 32,492 for each hemisphere. Further details on data and connectome construction can be found in Supplementary Information 2.3.

## Cortical parcellations

The main results of the study analysed task-evoked activation maps and task-free FC data parcellated into discrete regions. We presented results using the HCP–MMP1 parcellation with 180 regions per hemisphere (we term this the Glasser360 parcellation), which reflects sharp areal boundaries based on the combination of cortical architecture, function, connectivity and topography[28]. The parcellation was on the fsLR-32k space as provided by HCP. To test the robustness of our results (Supplementary Fig. 2 and Extended Data Figs. 2 and 3) we also performed our analysis using parcellations provided by Schaefer et al.[82] on the fsLR-32k space at varying resolution (100, 200, 400, 600, 800 and 1,000 parcels across both hemispheres; we refer to these as, respectively, Schaefer100, Schaefer200, Schaefer400, Schaefer600, Schaefer800 and Schaefer1000 parcellations).

## Derivation of connectome eigenmodes

Connectome eigenmodes were derived according to previous methods[29] to enable comparison with previous findings. Note that, in previous studies[19,20,29,83], connectome eigenmodes have been referred to as connectome harmonics but we use the term eigenmodes here because the term harmonics implies integer frequency ratios, which is not necessarily guaranteed for brain-derived modes.

We obtained high-resolution maps of connectivity measured with dMRI tractography as described in ref. 81, in which the connectivity of each of the 32,492 vertices in the cortical surface mesh was estimated by tracing streamlines from each point until they terminated at some other point. Connection weights between vertices (considered as nodes) were estimated as the number of interconnecting streamlines without the need for normalization[84]. Tractography was performed on individuals from HCP (see Supplementary Information 2.3 for further details on the data and tractography method). From the tractography data we combined the individual weighted connectivity matrices of size 32,492 × 32,492 to generate a group-averaged connectome, $W_{connectome}$, with weights representing the average number of streamlines. We then generated a binary adjacency matrix, $A_{local}$, that captures the discrete representation of local spatial relations between points in the cortical surface mesh model constructed by connecting two vertices that are direct neighbours in the mesh. These links are intended to capture local, very-short-range connectivity that cannot be resolved by traditional dMRI tractography[29].

Following ref. 29, the group-averaged weighted connectome, $W_{connectome}$, was thresholded to remove the smallest weights such that the number of connections was fourfold greater than $A_{local}$. The resulting thresholded matrix was binarized to obtain the group adjacency matrix, $A_{connectome}$. Finally we generated a merged adjacency matrix $A_C = A_{local} || A_{connectome}$ (with matrix of size 32,492 × 32,492; || is the logical OR operator), which captures both local vertex-to-vertex connectivity and complex short- and long-range connections as measured empirically. Note that the connection density of the adjacency matrix, $A_C$, that resulted from the thresholding process described above was 0.10%.

The connectome eigenmodes were obtained by solving the eigenvalue problem,

$$L'\psi = -\lambda\psi, \tag{6}$$

where $L'$ is the normalized graph Laplacian, a discrete counterpart of the LBO. The normalized graph Laplacian is related to the unnormalized graph Laplacian, $L$, as $L' = D^{-1/2}LD^{-1/2}$, with $L$ defined according to previous work[34],

$$L = \frac{1}{2}[(D - A_C) + (D - A_C)^T], \tag{7}$$

where $D$ is the diagonal degree matrix and superscript $T$ denotes matrix transpose. As with geometric eigenmodes, the eigenvalue solutions of equation (6) form the sequence $0 \leq \lambda_1 \leq \lambda_2 \leq \dots$. Note also that the use of a high-resolution, vertex-level connectome results in connectome eigenmodes spanning a space with dimensions (number of modes) equal to the number of vertices, allowing fair comparison with geometric eigenmodes.

## Derivation of EDR eigenmodes

Exponential distance rule eigenmodes were derived also by solving equation (6). However, in this case the non-normalized graph Laplacian, $L$, was defined using a synthetically constructed 32,492 × 32,492 adjacency matrix, $A_E$, that follows a stochastic EDR. To construct $A_E$ we used the group-averaged, unthresholded weighted connectome, $W_{connectome}$, from the previous section. We then fitted the variation of weights as a function of Euclidean distance, $d$, between vertices in the cortical surface by an exponential function of the form $e^{-\alpha d}$, where $\alpha$ is a scale exponent parameter. The fitting was performed in MATLAB 2019b using a nonlinear least-squares method, resulting in an optimal empirical parameter value of $\alpha_{empirical} = 0.12$, consistent with previous estimates based on the connection probability versus distance function[85]. We then generated a random, binary adjacency matrix following a stochastic EDR wiring process in which two vertices were connected with probability $e^{-\alpha d}$ with $\alpha = \alpha_{empirical} = 0.12$.

The adjacency matrix, $A_E$, generated by instances of the EDR model had a connection density of 1.55%, which was substantially higher than the 0.10% density of $A_C$. We therefore constructed another version of connectome eigenmodes that thresholded the group-averaged empirical connectome to achieve a final $A_C$ with a density that matched $A_E$ to allow fair comparison (we termed this density-matched connectome eigenmodes). Supplementary Figs. 6 and 7 present a more thorough evaluation of the performance of connectome eigenmodes as a function of network connection density.

## Comparisons with statistical basis sets

The geometric, connectome and EDR eigenmode basis sets were all derived from a generative model that accounts for how brain function emerges from anatomy. This approach contrasts with the statistical basis sets commonly used in the literature, which can efficiently summarize the data but offer no insights into the underlying generative process. We evaluated the performance of geometric eigenmodes with respect to two statistical basis sets, one derived from PCA of the functional data themselves and the other based on a Fourier spatial basis set. Further details of these analyses are provided in Supplementary Information 5 and 6, respectively.

## Modal power spectra of task-evoked activation maps

To investigate the spectral content of task-evoked activation maps we calculated their modal power spectra using the modal decomposition in equation (4) and taking the absolute square of the amplitudes, $a$. This is analogous to calculation of the temporal power spectral density from Fourier analysis. We then normalized the power in mode $j$ with respect to the total power in all modes, such that

$$P_j = \frac{|a_j|^2}{\sum_{j=1}^{N}|a_j|^2}. \tag{8}$$

We calculated the modal power spectra of two sets of task-evoked data. The first set comprises unthresholded activation maps from the HCP task-evoked data (Supplementary Information 2.1). Before perfoming spectral analysis we took the group average of the activation maps across 255 individuals and then analysed the power spectrum of the group-averaged activation map of each task contrast. The results in Fig. 3a and Extended Data Fig. 8a show the mean power spectrum across the 47 HCP task-contrast maps. Contrast-specific power spectra for the seven key HCP task contrasts are shown in Extended Data Fig. 7.

The second set comprises 10,000 activation maps from 1,178 independent experiments from the NeuroVault repository (https://neurovault.org/)[33]. We used the python module Nilearn[86] to retrieve activation maps from NeuroVault that were unthresholded and with a modality tag of fMRI-BOLD. We projected the activation maps from volume onto the fsLR-32k CIFTI space to match the HCP data using Nilearn (via the function nilearn.surface.vol_to_surf). We then analysed the power spectrum of each activation map. The results in Fig. 3a and Extended Data Fig. 8a show the mean power spectrum across the 10,000 NeuroVault maps.

We then compared the power spectra of the empirical maps with those of surrogate random maps with varying levels of smoothing to further investigate the relevance of long-wavelength modes. In particular, we generated 10,000 random maps in volume space, which we smoothed at kernel sizes with full-width at half-maximum ranging from 0 to 50 mm, and projected onto the fsLR-32k CIFTI space. We then calculated the mean square logarithmic error (MSLE) of the power spectra between the empirical and surrogate maps. Further details of this analysis and the MSLE measurements are provided in Supplementary Information 7.

## Contributions of long- and short-wavelength modes

To understand the contributions of long- and short-wavelength geometric eigenmodes in the reconstruction of task-evoked activation maps, we sequentially removed modes before performing the reconstruction process. Specifically, we started by reconstructing the seven key HCP contrast maps using 200 geometric eigenmodes and calculated the reconstruction accuracy (that is, correlation between empirical and reconstructed maps), serving as our baseline. We then performed an incremental, sequential removal of modes starting from long-wavelength modes (that is, removal of mode 1, modes 1–2, modes 1–3, modes 1–4, …, modes 1–200) and calculated reconstruction accuracy at every increment. We repeated the same procedure but starting from short-wavelength modes (that is, removal of mode 200, modes 199–200, modes 198–200, modes 197–200, …, modes 1–200).

## NFT wave model

As stated above, an implication of the superior performance of geometric eigenmodes is that neuronal activity is dominated by wave dynamics as predicted by NFT. To investigate whether wave dynamics can explain complex spatiotemporal patterns of neuronal activity, we implemented a simple NFT wave model in which dynamics are described by an isotropic damped wave equation without regeneration[5,6,87],

$$\left[\frac{1}{\gamma_s^2}\frac{\partial^2}{\partial t^2} + \frac{2}{\gamma_s}\frac{\partial}{\partial t} + 1 - r_s^2\nabla^2\right]\phi(\mathbf{r}, t) = Q(\mathbf{r}, t), \quad (9)$$

where $\phi(\mathbf{r}, t)$ is the neural activity at location $\mathbf{r}$ and time $t$, $Q$ is an external input, $\gamma_s$ is the damping rate and $r_s$ is the spatial length scale of the wave propagation (conceptually related to the $\alpha$-length scale of the stochastic EDR wiring process in the derivation of EDR eigenmodes). This form tells us that an impulse input will produce an activity that dissipates at a rate of $\gamma_s$ and propagates at a velocity of $\gamma_s r_s$. Here we treated $\gamma_s$ as a fixed parameter with the value of 116 s$^{-1}$ taken from electrophysiological estimates[21] and $r_s$ as a free parameter. We applied the

model on the cortical midthickness surface mesh with 32,492 vertices per hemisphere to solve the activity at each vertex. Note that the propagation of activity between points is governed by their white-matter connectivity, with strength that decays approximately exponentially with distance (Supplementary Fig. 10). This distance dependence is more apparent when equation (9) is converted into its equivalent integral form (see Supplementary Information 8 for details).

## Neural mass model

We compared the performance of our simple NFT wave model with a biophysical large-scale neural mass model in which mesoscopic dynamics emerge from the interactions of neural populations (that is, neural masses) coupled via an empirical anatomical connectivity[18]. In a typical neural mass model, each brain region $i$ has its own mean-field population dynamics and its temporal activity, $S_i$, is defined by the general equation,

$$\frac{\mathrm{d}S_i}{\mathrm{d}t} = f(\mathbf{S}, \theta_i, C, G), \quad (10)$$

where $f$ is a function describing the evolution of the region's activity. The function depends on the activity of other regions $\mathbf{S}$, local population parameters $\theta_i$, anatomical connectivity between regions $C$ (a parcellated version of the vertex-resolution connectome, $W_{connectome}$, used to derive the connectome eigenmodes) and global coupling parameter $G$ that scales the connectivity between regions.

There are several whole-brain neural mass models available in the literature that we can use[88], from simple phase oscillator models (for example, the Kuramoto model[89]) to more complex biophysical population models (for example, the Wilson–Cowan model[90]). All these models follow the form of equation (10), especially their reliance on an anatomical interregional connectivity matrix, $C$. Here we focus on one widely used neural mass model, the BEI model[38,65,91,92]. The BEI model uses a mean-field approach to approximate local population dynamics in each brain region, which are coupled via an anatomical connectivity matrix derived from dMRI. Each brain region $i$ comprises interacting populations of excitatory ($E$) and inhibitory ($I$) neurons governed by the following nonlinear stochastic differential equations,

$$\frac{\mathrm{d}S_i^{(E)}}{\mathrm{d}t} = -\frac{S_i^{(E)}}{\tau_E} + (1 - S_i^{(E)})\gamma r_i^{(E)} + \sigma v_i(t), \quad (11)$$

$$\frac{\mathrm{d}S_i^{(I)}}{\mathrm{d}t} = -\frac{S_i^{(I)}}{\tau_I} + r_i^{(I)} + \sigma v_i(t), \quad (12)$$

$$r_i^{(E)} = H^{(E)}(I_i^{(E)}) = \frac{a_E I_i^{(E)} - b_E}{1 - \exp[-d_E(a_E I_i^{(E)} - b_E)]}, \quad (13)$$

$$r_i^{(I)} = H^{(I)}(I_i^{(I)}) = \frac{a_I I_i^{(I)} - b_I}{1 - \exp[-d_I(a_I I_i^{(I)} - b_I)]}, \quad (14)$$

$$I_i^{(E)}(t) = I^{ext} + W_E I_0 + w_{EE} S_i^{(E)}(t) + GJ\sum_j C_{ij} S_j^{(E)}(t) - w_{IE} S_i^{(I)}(t), \quad (15)$$

$$I_i^{(I)}(t) = W_I I_0 + w_{EI} S_i^{(E)}(t) - S_i^{(I)}(t), \quad (16)$$

where $S_i^{(E,I)}$, $r_i^{(E,I)}$ and $I_i^{(E,I)}$ represent the synaptic gating variable, firing rate and total input current, respectively, for $E$ and $I$ populations. The parameters $\tau_{E,I}$ are the time constants, $\gamma$ is a kinetic rate constant and $v_i(t)$ is a time-varying random Gaussian input with standard deviation $\sigma$. The functions $H^{(E,I)}$ are sigmoidal neuronal response functions transforming total input currents $I_i^{(E,I)}$ into firing rates $r_i^{(E,I)}$, which are parametrized by the gain factors $a_{E,I}$, threshold currents $b_{E,I}$ and

curvature parameters $d_{E,I}$. In equations (15) and (16), $I_i^{\text{ext}}$ is the external input, $I_0$ is the local input current scaled by $W_E$ and $W_I$ for the excitatory and inhibitory populations, respectively, $w_{EE}$ is the excitatory–excitatory strength, $w_{EI}$ is the excitatory–inhibitory strength, $w_{IE}$ is the inhibitory–excitatory strength, $G$ is the global coupling parameter, $J$ is the effective $N$-methyl-D-aspartate (NMDA) conductance and $C_{ij}$ is the structural connectivity strength between regions $i$ and $j$ estimated from dMRI.

Overall, the BEI model has 15 fixed parameters and four free parameters, as detailed in Supplementary Table 3. The values of the fixed parameters were taken from ref. 91. To allow direct comparison we also used the discretized connectome data provided by ref. 91 to define the interregional structural connectivity matrix, $C$. These connectome data were derived from minimally preprocessed dMRI data of 334 unrelated HCP subjects and constructed via the probabilistic tractography tool of FMRIB Software Library (FSL)[93]. We refer readers to ref. 91 for further details of data processing and connectome construction. Our results did not change when the parcellated version of our connectome data was used. We note that numerical solutions of NFT equations (such as in equation (9)) also spatially discretize the cortex, but into a very fine array of points, before integrating whatever temporal dynamics have been chosen at each point with connectivity that is correct in the continuum limit. As such, neural mass models are approximations to the more general NFT approach[94,95]. We also reiterate that local dynamics do not affect the spatial eigenfunctions of the NFT wave model.

## Haemodynamic model

To simulate fMRI data we transformed the neural activity generated by the NFT wave and BEI neural mass models to a blood oxygen-level dependent (BOLD) signal using the well-established Balloon–Windkessel haemodynamic model[96]. Note that, although this model is a simple approximation to more detailed models of the physiological haemodynamic processes underlying the BOLD signal[97–99], we use this approximation here to allow direct comparison with the vast majority of modelling studies in the literature[65,91,92,100,101]. The BOLD–fMRI signal in each vertex or brain region $i$ is governed by the following differential equations,

$$\frac{dz_i}{dt} = N_i(t) - \kappa z_i(t) - \gamma[f_i(t) - 1], \tag{17}$$

$$\frac{df_i}{dt} = z_i(t), \tag{18}$$

$$\frac{dv_i}{dt} = \frac{1}{\tau}[f_i(t) - v_i^{1/\alpha}(t)], \tag{19}$$

$$\frac{dq_i}{dt} = \frac{1}{\tau}\left\{\frac{f_i(t)}{\rho}[1 - (1-\rho)^{1/f_i(t)}] - v_i^{1/\alpha - 1}(t)\right\}, \tag{20}$$

$$\frac{dy_i}{dt} = V_0\left\{k_1[1 - q_i(t)] + k_2\left[1 - \frac{q_i(t)}{v_i(t)}\right] + k_3[1 - v_i(t)]\right\}, \tag{21}$$

where $z_i, f_i, v_i, q_i$ and $y_i$ are the vasodilatory signal, blood inflow, blood volume, deoxyhaemoglobin content and BOLD signal variables, respectively. The variable $N_i$ represents the neural activity generated by the wave and BEI neural mass models. For the wave model, $N(t)$ is $\phi(t)$, and for the BEI neural mass model, $N_i(t)$ is $S_i^{(E)}(t)$. The model parameters and their values taken from previous work[96,102] were as follows: $\kappa = 0.65\text{ s}^{-1}$ is the signal decay rate, $\gamma = 0.41\text{ s}^{-1}$ is the flow-dependent elimination rate, $\tau = 0.98\text{ s}$ is the haemodynamic transit time, $\alpha = 0.32$ is the Grubb's exponent, $\rho = 0.34$ is the resting oxygen extraction fraction, $V_0 = 0.02$ is the resting blood volume fraction and $k_1 = 3.72$, $k_2 = 0.53$ and $k_3 = 0.53$ are 3T fMRI parameters.

## Modelling resting-state dynamics

We used the NFT wave and BEI models, together with the haemodynamic model, to estimate various spontaneous FC properties. For the wave model we solved equation (9) with a white noise input to mimic the absence of any structured stimulus, following previous studies[21,103]. We combined the resulting solution with the haemodynamic model in equations (17–21) to simulate the BOLD–fMRI signal. The simulated BOLD signal was downsampled to a sampling interval of 0.72 s with 1,200 time frames to match the resting-state HCP data described in Supplementary Information 2.2. Finally we parcellated the simulated BOLD signal using the HCP–MMP1 parcellation and calculated the correlation matrix representing the model FC.

For the BEI model we can use the method described above to calculate the model FC starting from the numerical solutions of equation (11) for each brain region. However, it has been shown that one can linearize the equations of the BEI and haemodynamic models to obtain an analytic approximation of the model FC[91]. Due to the large number of model parameters in the BEI model we used this analytic approximation in this study because it allows more comprehensive and computationally efficient model fitting. See ref. 91 for further details.

We fitted the model FCs to empirical FCs (also parcellated using the HCP–MMP1 parcellation) of the same HCP participants used in all our analyses by fitting each model's free parameters. We divided the participant sample into training and test sets, each with 125 individuals, to enable out-of-sample evaluation of model performance and to avoid overfitting. In particular, we fitted the model parameters on the training set data and used the fitted model parameters to predict the test set data. Model fitting and performance evaluation were based on three widely used FC-related metrics[65,92]: edge FC, node FC and FCD.

The edge FC metric was calculated by taking the Pearson correlation of the upper triangular elements (that is, the strength of FC edges) of the $z$-transformed model and empirical FCs. We only took the upper triangular elements because the FC values are symmetric with respect to the diagonal. Higher correlations represent a better fit between model and data.

The node FC metric was calculated by taking the Pearson correlation of the average FC strength of each brain region in the model and empirical FCs. The average FC strength of a brain region $i$ was defined as $\frac{1}{n}\sum_{j=1}^{n}\text{FC}_{ij}$, where $n$ is the total number of brain regions. Once again, higher correlations indicate better model fits.

The FCD metric captures the spatiotemporal statistics of resting-state activity and was calculated as follows[92]. The time series at each region $i$ was filtered between 0.04 and 0.07 Hz using a second-order Butterworth filter; this band was based on ref. 104 and its inclusion was motivated by its functional relevance to the brain[105,106]. The time series was then Hilbert ($H$) transformed to calculate the quantity $y_i(t) = x_i(t) + jH_i(t)$ where $j$ is the imaginary number. The instantaneous complex argument $\theta_i(t) = \tan^{-1}[H_i(t)/x_i(t)]$ was then computed. The level of synchrony between regions $i$ and $j$ at time $t$, $\Delta(i,j,t)$, was calculated as

$$\Delta(i,j,t) = \cos[\theta_i(t) - \theta_j(t)]. \tag{22}$$

Note that we do not term $\theta_i(t)$ a phase, as is sometimes done in the literature, because our signals are broadband whereas this interpretation is applicable only to signals that are nearly monochromatic. We computed this quantity for comparison with previous work[65,92], then we calculated the similarity of global synchrony, $\varphi_{uv}$, between two time instances, $\tau_u$ and $\tau_v$, with $\varphi_{uv}$ defined as

$$\varphi_{uv} = \frac{1}{d_u d_v}\sum_{i>j}\Delta(i,j,\tau_u)\Delta(i,j,\tau_v), \tag{23}$$

where

$$d_x = \sqrt{\sum_{i>j} [\Delta(i,j,\tau_x)]^2}. \tag{24}$$

Here, $\varphi_{uv}$ is the FCD, which is a symmetric time × time matrix interpreted as the similarity of global synchrony between times $\tau_u$ and $\tau_v$. We then compared the distributions of the upper triangular elements of the model and empirical FCD estimates, concatenated across all individuals or model realizations as per ref. 92, to better capture the general fluctuations of the data. The FCD distributions of the model and the data were compared using the KS statistic, with lower KS statistic indicating better model fit.

The wave model has one free parameter—the spatial length scale of wave propagation, $r_s$—that was optimized to fit the empirical fMRI data. Specifically, we constructed a vector of 20 values of $r_s$ evenly spaced between 10 and 100 mm. For each $r_s$ value the FC metrics described above were calculated. The optimization landscapes are shown in Extended Data Fig. 10. We took the value of $r_s$ that minimized the FCD KS statistic because this metric has been found to be the most stringent benchmark for model–data comparisons among the three metrics discussed above[92]. This procedure resulted in an optimized value of $r_s = 28.9$ mm, which is smaller than those obtained previously in analyses of EEG data[21], possibly due to either the haemodynamic processes limiting neural activity propagation to neighbouring regions or our focus here on cortico-cortical dynamics[97,98].

The BEI neural mass model has 15 fixed parameters and four free parameters—that is, $w_{EE}$, $w_{EI}$, $w_{IE}$ and $G$—that were optimized to fit the data, following ref. 91. This procedure resulted in optimized parameters $w_{EE} = 9.80$, $w_{EI} = 1.48$, $w_{IE} = 7.13$ and $G = 6.87$. Further details of the optimization process can be found in Supplementary Information 9.

### Measurement of time-lagged properties of resting-state dynamics

In addition to the FC-based metrics used for model fitting and evaluation of model performance, we investigated whether the wave and neural mass models could also capture the temporal properties of propagated activity. In particular we analysed the lag structure (or lag threads) of resting-state BOLD–fMRI time courses, as proposed by refs. 39,107. We briefly discuss the algorithm for calculation of the lag structure of both empirical and simulated fMRI data in Supplementary Information 10 and refer readers to previous articles[39,107] for further details. Note that there are several other ways of characterizing the spatiotemporal properties of resting-state activity[37,108,109], but the method currently chosen is sufficient for our purposes.

### Modelling stimulus-evoked dynamics

We also evaluated the degree to which the wave model could capture classical properties of evoked neural responses. Specifically we used the optimized wave model (with $r_s = 28.9$ mm) to investigate neural dynamics in response to a stimulus applied to the left V1. In particular, the stimulus $Q(\mathbf{r},t)$ was a 1 ms pulse ($t = 1$–2 ms) with magnitude 20 s$^{-1}$ (the results are robust to changes in amplitude) restricted to vertices in the cortical surface that fall within the V1 region as defined by the HCP–MMP1 parcellation. We performed the simulation over a 100 ms time period with 0.1 ms resolution. Finally we parcellated the activity $\phi(\mathbf{r}, t)$ into 180 regions using the HCP–MMP1 parcellation.

We compared the activity profile (amplitude versus time) of each brain region, focusing on the time for the activity to reach peak amplitude (that is, time to peak). Specifically we investigated whether the temporal precedence of activity follows the human visual cortical hierarchy from visual to frontal cortices, which includes the following 17 brain regions: V1, V4, 7m, 7Am, TE1p, 7AL, 24dd, 2, 24dv, 8BM, 10r, 10v, 8BL, 10pp, 10d, 9-46v and 9-46d. These regions closely resemble areas previously identified in the macaque neocortex visual hierarchy using tract-tracing data and nonlinear network modelling[41].

We also compared the regional estimates of peak response times to T1w:T2w values, which is a non-invasive measure sensitive to intracortical myelin content[46] and a good proxy for cortical hierarchy rank[110]. The myelin map in fsLR-32k space was obtained from the HCP dataset[27,28] and then parcellated using the HCP–MMP1 parcellation. We quantified the relationship between the regional values of time to peak and myelin content via Spearman rank correlation. The statistical significance of the correlation was assessed by comparison with a null distribution of 10,000 correlation values obtained using a spatially constrained spin-test approach[111,112]. This approach calculates the correlation between one map and random spatially rotated versions of the other map, thereby preserving the spatial relationship of the parcels in the map. The resulting $P$ value, $P_{\text{spin}}$, is the fraction of null correlation values greater than the empirical correlation value. Finally we repeated the above statistical test including all brain regions (that is, not restricted to visual hierarchy brain regions) to ensure that our findings were not driven by our particular selection of regions of interest (ROIs). This is a more conservative test because not all brain regions show strong evoked responses to visual stimulation.

### Estimation of the geometric eigenmodes of non-neocortical structures

We extended our eigenmode analysis to regions outside the neocortex, focusing in particular on the subcortex (thalamus and striatum) and archicortex (hippocampus). Unlike the cortical ribbon, which can be modelled as a 2D sheet, these structures are solid 3D objects. We therefore calculated the geometric eigenmodes using a tetrahedral mesh rather than the surface-based triangular mesh to account for the full 3D geometry of the non-neocortical structures[72], as outlined below.

We first used the probabilistic Harvard–Oxford subcortical atlas (https://fsl.fmrib.ox.ac.uk/fsl/fslwiki/Atlases) to generate a volumetric binary mask in each hemisphere for the thalamus, striatum and hippocampus. Voxels with a probability of 25% or more of belonging to these structures were included in the masks. We used FreeSurfer's mri_mc function, which implements a marching-cubes algorithm, to construct a 2D surface by tessellating the volumetric masks, followed by the Gmsh software (https://gmsh.info/) to convert the 2D surface into a 3D tetrahedral mesh. We then used the LaPy python library to solve equation (1) on the tetrahedral mesh of each non-neocortical structure to obtain the eigenmodes and their corresponding eigenvalues. Finally we projected the eigenmodes in tetrahedral space back into the natural volumetric space via interpolation. Hence, the resulting eigenmodes spatially vary through the 3D voxels comprising each non-neocortical structure's volume. For this part of the study we discarded the first constant mode and used the next 20 modes in our analyses.

### Mapping the functional organization of non-neocortical structures

The signal-to-noise ratio of fMRI is generally weaker in non-neocortical structures than in the neocortex[113]. Moreover, fine-grained task activations are often hard to resolve in these smaller structures due to the limited resolution of fMRI and the spatial smoothing induced by common fMRI processing methods. Thus, to efficiently map the functional organization of the thalamus, striatum and hippocampus we used connectopic mapping[49] of the resting-state fMRI signals in each structure to obtain their dominant functional modes (often called gradients, although the patterns capture spatial variations in pointwise similarities of FC profiles). This technique, and related procedures, have been extensively used in past work to study the functional organization of these structures[114–116].

We applied connectopic mapping to the volumetric voxel-wise resting-state fMRI data of the HCP individuals following the procedure described in ref. 49. Specifically, for each individual and non-neocortical ROI (that is, thalamus, striatum and hippocampus) we constructed an ROI time-series data matrix, $A$, of size $T \times N$, where $T$ is the number of

# Article

time frames and $N$ is the number of voxels in the ROI. Similarly, for each individual we constructed the grey matter time-series data matrix, $B$, of size $T \times M$, where $M$ is the number of grey matter voxels outside the ROI. Because $M$ is generally large ($10^3$ or above), we reduced the dimensionality of $B$ using singular value decomposition to construct the matrix $\tilde{B}$ of size $T \times (T-1)$. The connectivity fingerprint of every voxel within the ROI was then calculated using the Pearson correlation of $A$ and $\tilde{B}$ to obtain the matrix, $C$, of size $N \times (T-1)$. Similarity in the connectivity fingerprints between each pair of ROI voxels was calculated using the $\eta^2$ coefficient[117], resulting in the matrix, $S$, of size $N \times N$. The $\eta^2$ coefficient represents the fraction of variance in one connectivity profile accounted for by the variance in another. We then took the average $S$ across all individuals. A nonlinear manifold learning procedure using the Laplacian Eigenmaps algorithm[118] was applied to the $S$ matrix of each ROI to calculate its eigenvectors and eigenvalues. The eigenvectors represent the functional patterns, termed functional gradients, of the ROI, ordered according to the variance in FC similarity they explain. We only analysed the first 20 non-constant gradients to enable direct comparisons with the geometric eigenmodes. Typical applications of FC-based gradient analysis rarely consider more than the first five gradients[49,119].

We analysed the correspondence between the geometric eigenmodes and functional gradients in each non-neocortical structure by taking their absolute spatial correlations, given that the signs of the modes and gradients are arbitrary. Moreover, the ordering of the geometric eigenmodes within the same eigengroup can change (Supplementary Table 1; for example, the first eigengroup can have order flips among modes 2–4) and hence one-to-one correspondence between the indices of geometric eigenmodes and functional gradients is not guaranteed. We therefore examined all possible pairwise correlations between the modes and gradients and evaluated correspondence with respect to the mode that maximally correlated with each gradient. The maximal order differences observed between geometric modes and functional gradients were one, eight and five out of 20 for the thalamus, striatum and hippocampus, respectively.

## Reporting summary

Further information on research design is available in the Nature Portfolio Reporting Summary linked to this article.

## Data availability

Raw and preprocessed HCP data can be accessed at https://db.humanconnectome.org/. NeuroVault data can be accessed at https://neurovault.org/. Source data to replicate the results of the study are openly available at https://github.com/NSBLab/BrainEigenmodes and https://osf.io/xczmp/.

## Code availability

Computer codes used to calculate the eigenmodes, analyse results and reproduce the figures of the study are openly available at https://github.com/NSBLab/BrainEigenmodes.

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

**Acknowledgements** We thank S. Mansour L for providing access to high-resolution connectome data and K. Haak for assistance with connectopic mapping. HCP data were provided by the HCP, Wu–Minn Consortium (principal Investigators D. Van Essen and K. Ugurbil; no. 1U54MH091657) funded by the 16 NIH Institutes and centres that support the NIH Blueprint for Neuroscience Research, and by the McDonnell Center for Systems Neuroscience at Washington University. This work was supported by the MASSIVE HPC facility (www.massive.org.au), Sylvia and Charles Viertel Foundation grant no. 2017042 to A.F., National Health and Medical Research Council grant nos. 1197431 and 1146292 to A.F., Australian Research Council grant nos. FL220100184 and DP200103509 to A.F., National Health and Medical Research Council grant no. 2008612 to M.B., Australian Research Council Laureate Fellowship no. FL140100025 to P.A.R. and Australian Research Council Center of Excellence no. CE140100007 to P.A.R.

**Author contributions** K.M.A. and A.F. conceptualized the study. J.C.P., K.M.A., A.F. and M.O. designed the methodology. J.C.P., K.M.A. and A.F. performed the investigation and administered the project. J.C.P. and K.M.A. developed visualizations. A.F. acquired funding. A.F., P.A.R., B.D.F. and M.B. supervised the project. J.C.P. and A.F. wrote the original draft. All authors reviewed and edited the final manuscript.

**Competing interests** K.M.A. is a scientific advisor and shareholder in BrainKey Inc., a medical image analysis software company. The other authors declare no competing interests.

**Additional information**
**Correspondence and requests for materials** should be addressed to James C. Pang.

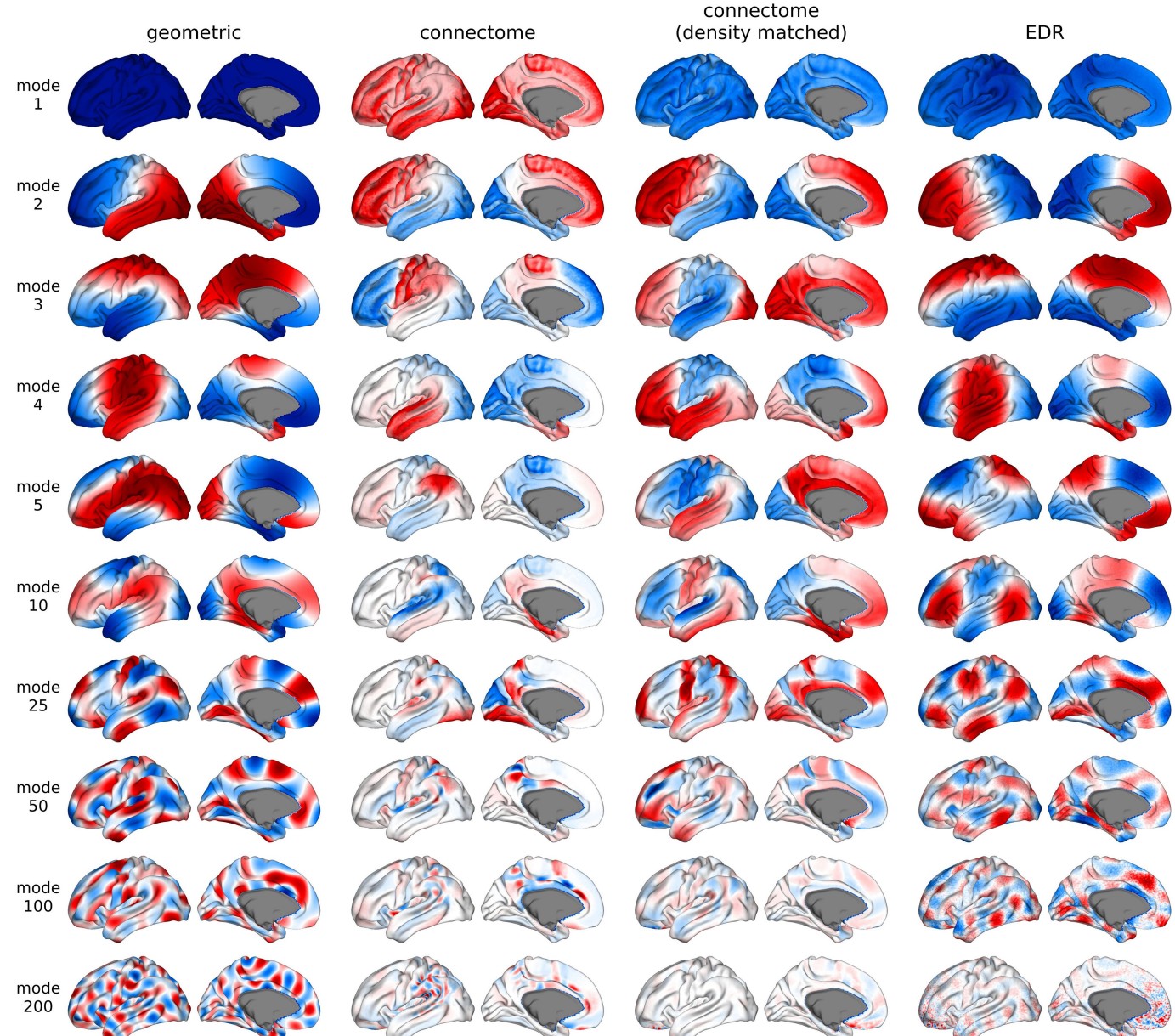

**Extended Data Fig. 1 | Eigenmode basis sets.** The basis sets from left to right are geometric eigenmodes, connectome eigenmodes, connectome eigenmodes using a connectivity matrix matching the density used by the exponential distance rule (EDR) eigenmodes, and EDR eigenmodes. Negative–zero–positive values are coloured as blue–white–red.

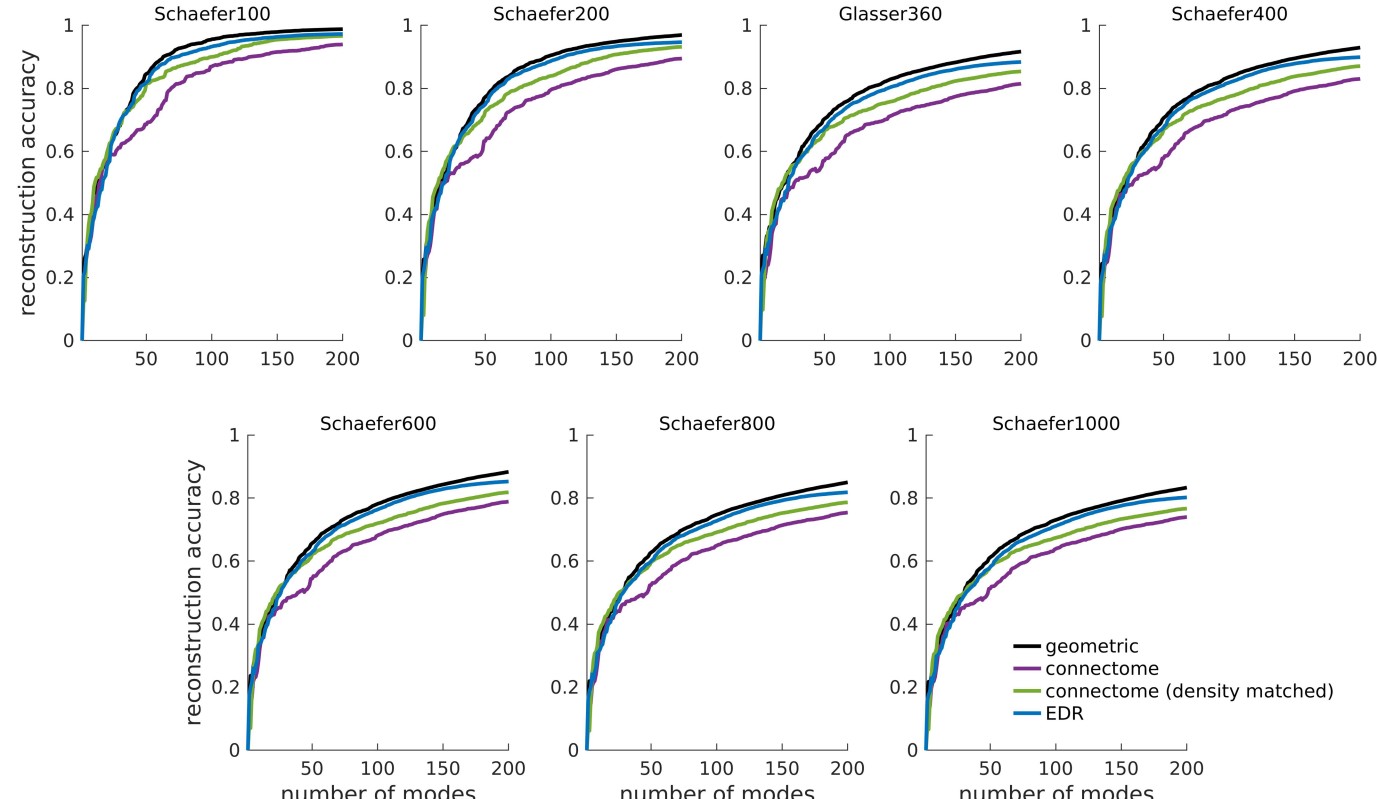

**Extended Data Fig. 2 | Reconstruction accuracy of resting-state FC achieved by different basis sets for different parcellation resolution.** The basis sets are geometric eigenmodes, connectome eigenmodes, connectome eigenmodes using a connectivity matrix with density matched to that used by the EDR eigenmodes, and EDR eigenmodes. Schaefer100, Schaefer200, Glasser360, Schaefer400, Schaefer600, Schaefer800, and Schaefer1000 has 100, 200, 360, 400, 600, 800, and 1000 parcels, respectively, across both hemispheres.

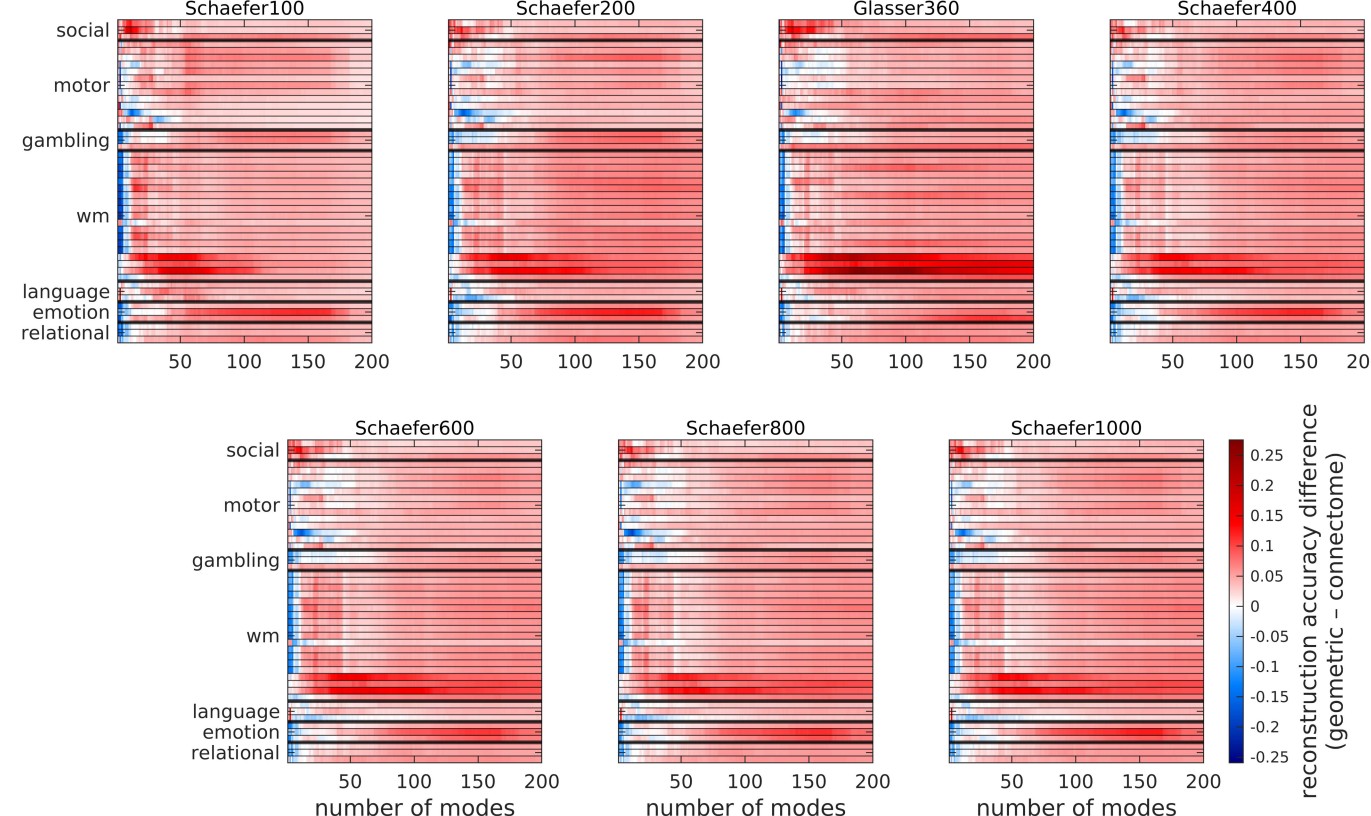

**Extended Data Fig. 3 | Difference in reconstruction accuracy of all 47 HCP task-contrast maps achieved by geometric eigenmodes and connectome eigenmodes for different parcellation resolution.** Each row represents a different task contrast, which have been grouped here by broad types (Supplementary Information 2.1). wm = working memory. Red indicates superior performance for geometric eigenmodes. Schaefer100, Schaefer200, Glasser360, Schaefer400, Schaefer600, Schaefer800, and Schaefer1000 has 100, 200, 360, 400, 600, 800, and 1000 parcels, respectively, across both hemispheres.

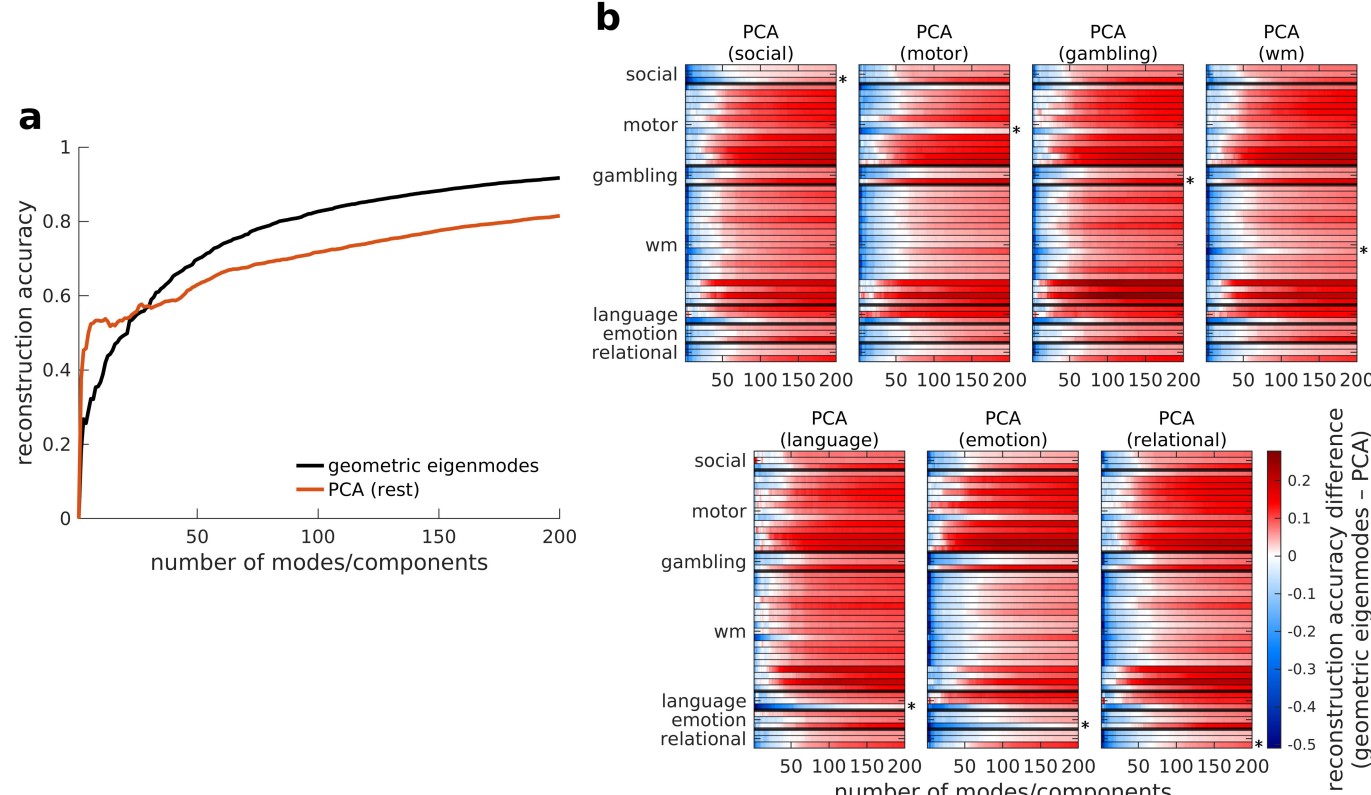

**Extended Data Fig. 4 | Reconstruction accuracy achieved by geometric eigenmodes and PCA.** (**a**) Reconstruction accuracy of resting-state FC. (**b**) Comparison of the reconstruction accuracy of all 47 HCP task-contrast maps, which have been grouped here by broad types (Supplementary Information 2.1). wm = working memory. Each row represents a different task contrast. Red indicates superior performance for geometric eigenmodes. The asterisk denotes the contrast (that is, the seven key HCP task contrasts) within the relevant task used to train the PCA.

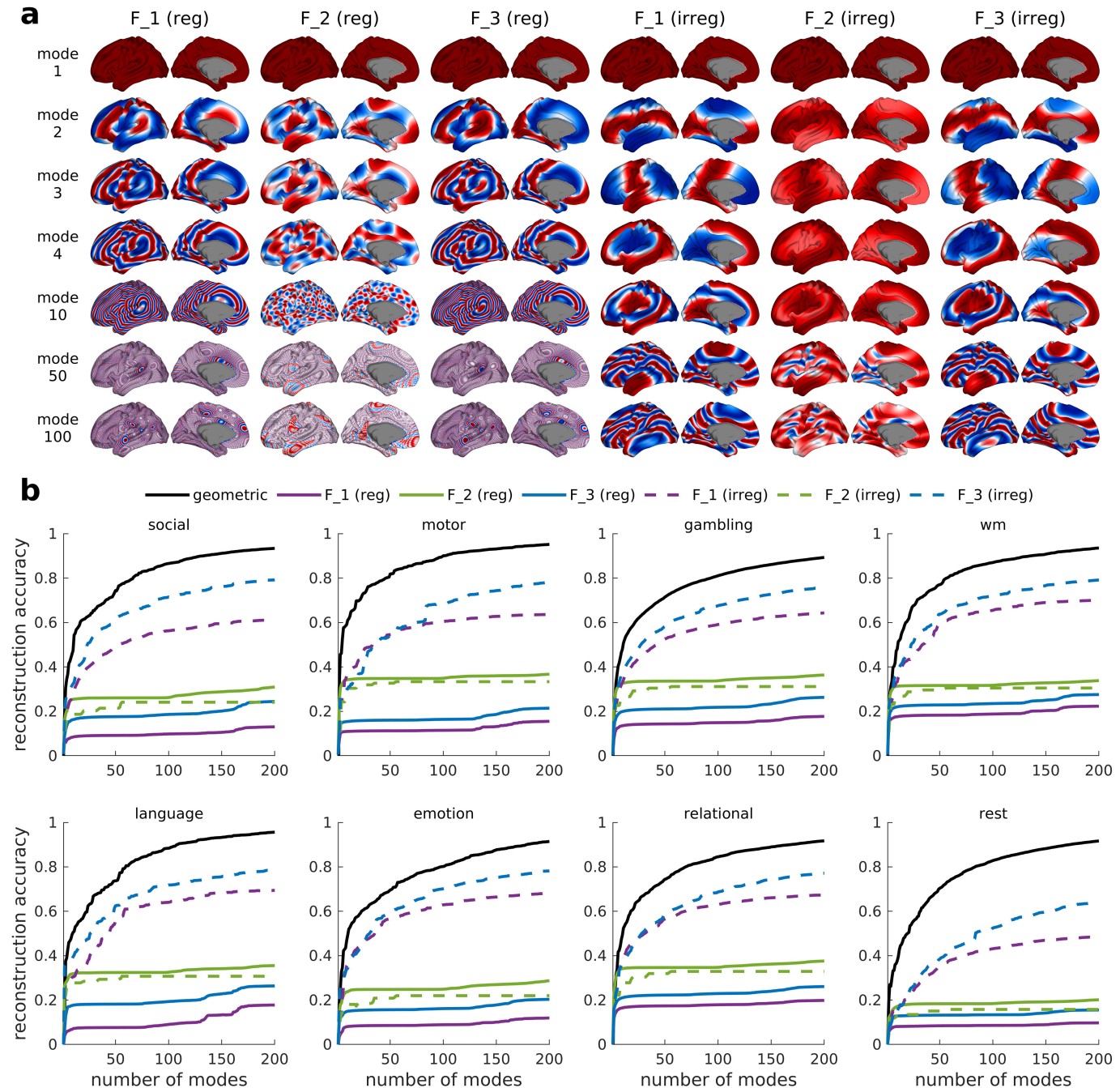

**Extended Data Fig. 5 | Comparison of geometric eigenmodes and Fourier basis sets. a**, Spatial maps of modes 1, 2, 3, 4, 10, 50 and 100 of six different Fourier basis sets with unit coefficients. The terms reg and irreg mean that the spatial wavelengths of the modes in the x-, y-, and z-directions are spaced in regular and irregular increments, respectively. See Supplementary Information 6 for details. **b**, Reconstruction accuracy of seven key HCP task-contrast maps and resting-state FC. See Supplementary Information 2.1 for details about the contrast maps. wm = working memory.

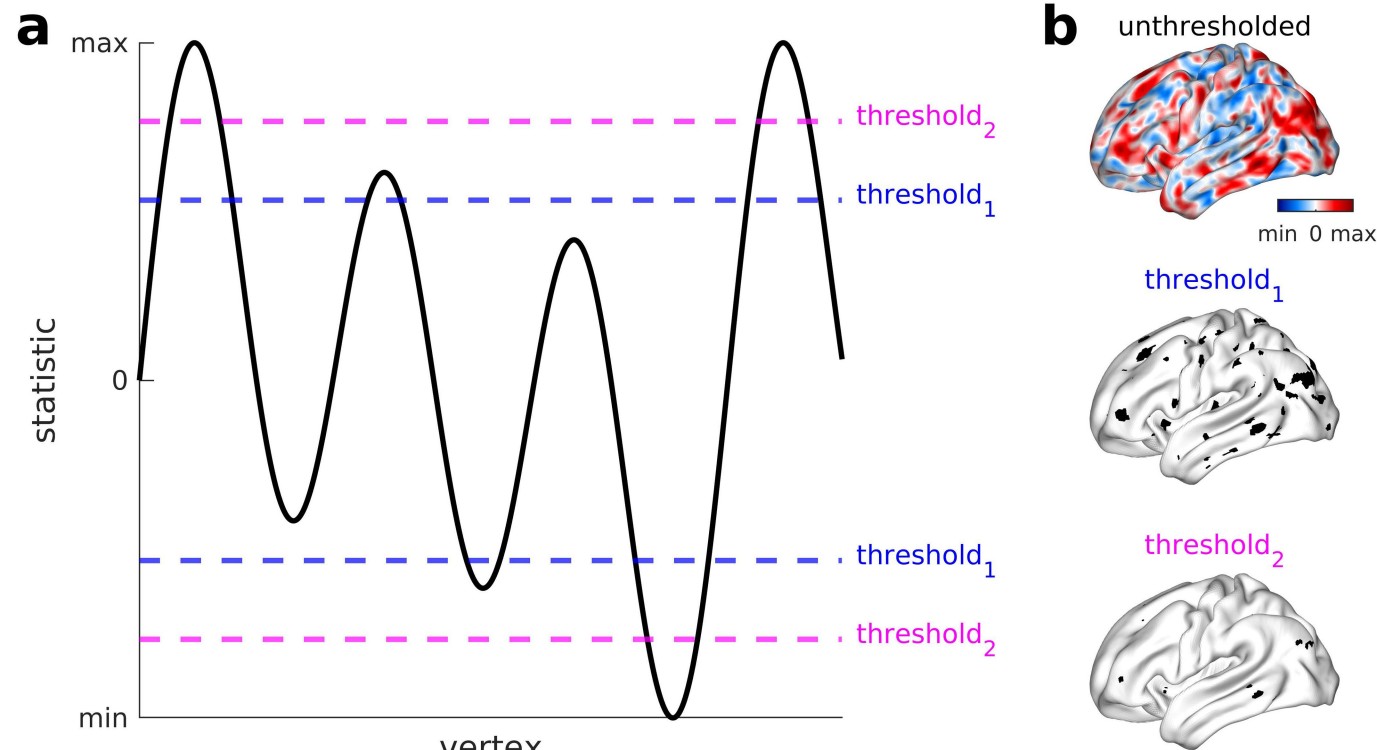

**Extended Data Fig. 6 | Classical neuroimaging approach of thresholding statistical maps. a**, Simple one-dimensional example of how different thresholds only capture focal clusters of activations and ignore the underlying structured pattern of activations. **b**, Spatially embedded demonstration of the concept depicted in **a** using unthresholded and binarized thresholded maps.

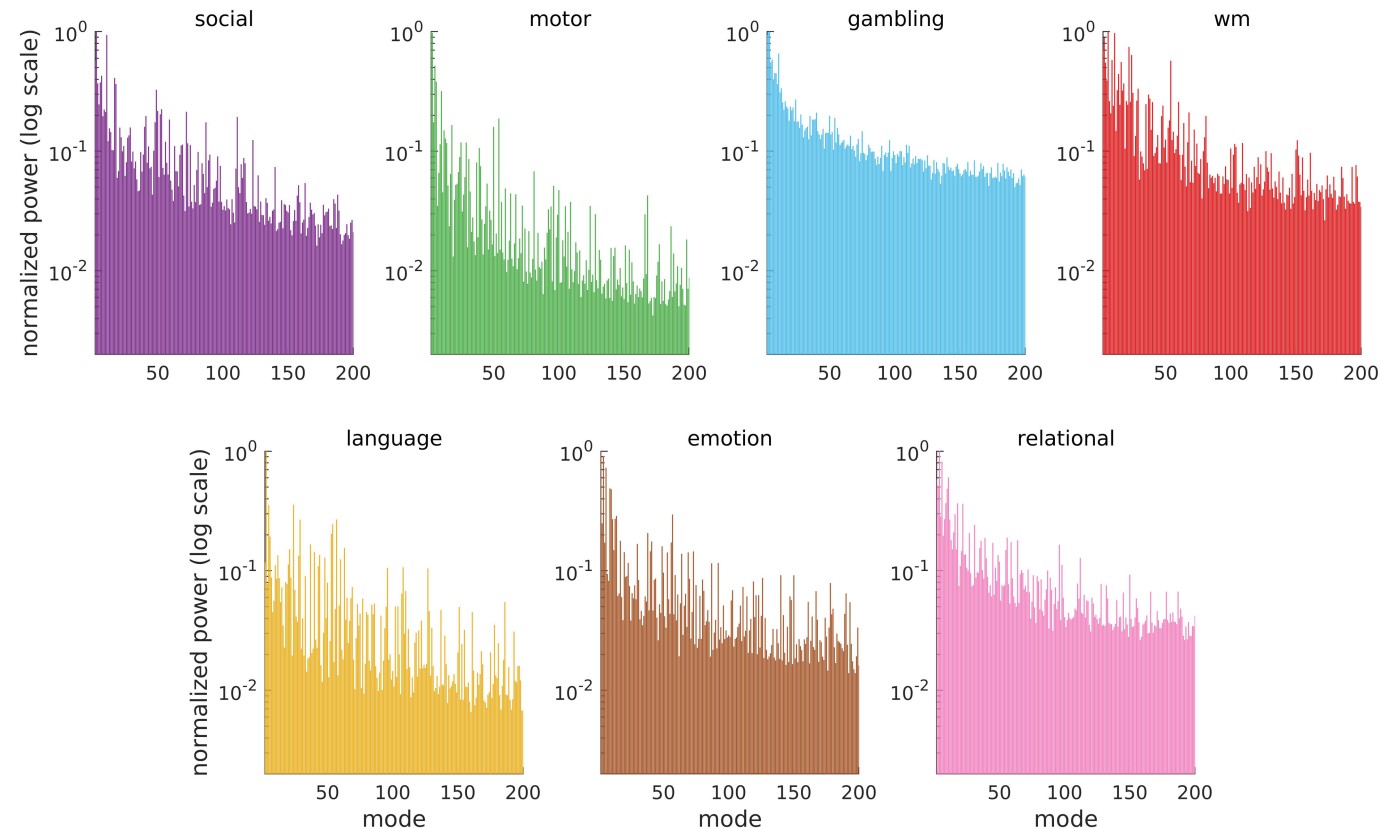

**Extended Data Fig. 7 | Normalized power spectrum of each of the seven key HCP task-contrast maps.** See Supplementary Information 2.1 for details about the contrast maps. wm = working memory.

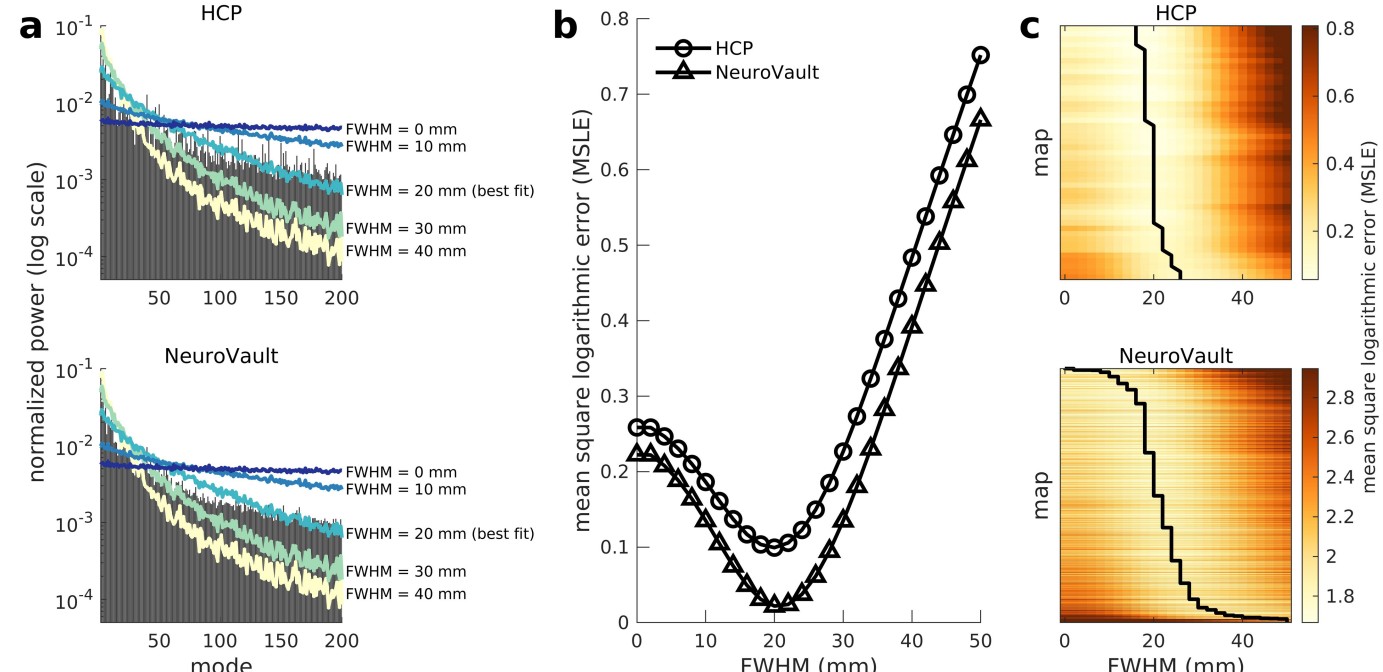

**Extended Data Fig. 8 | Power spectra of empirical task-activation maps and surrogate maps. a**, Normalized mean power spectra of 47 HCP task-contrast maps (top) and 10,000 contrast maps from the NeuroVault database (bottom). The coloured lines correspond to power spectra of surrogate data following the application of spatial smoothing filters with varying full-width at half-maximum (FWHM). **b**, Average mean square logarithmic error (MSLE) as a function of FWHM between normalized mean power spectra of HCP and NeuroVault contrast maps and smoothed surrogate data. **c**, MSLE separately obtained between the power spectra of each of the 47 HCP and 10,000 NeuroVault contrast maps and the smoothed surrogate data. Each row represents a different task-contrast map. The lines correspond to the FWHM where MSLE is minimum for each map.

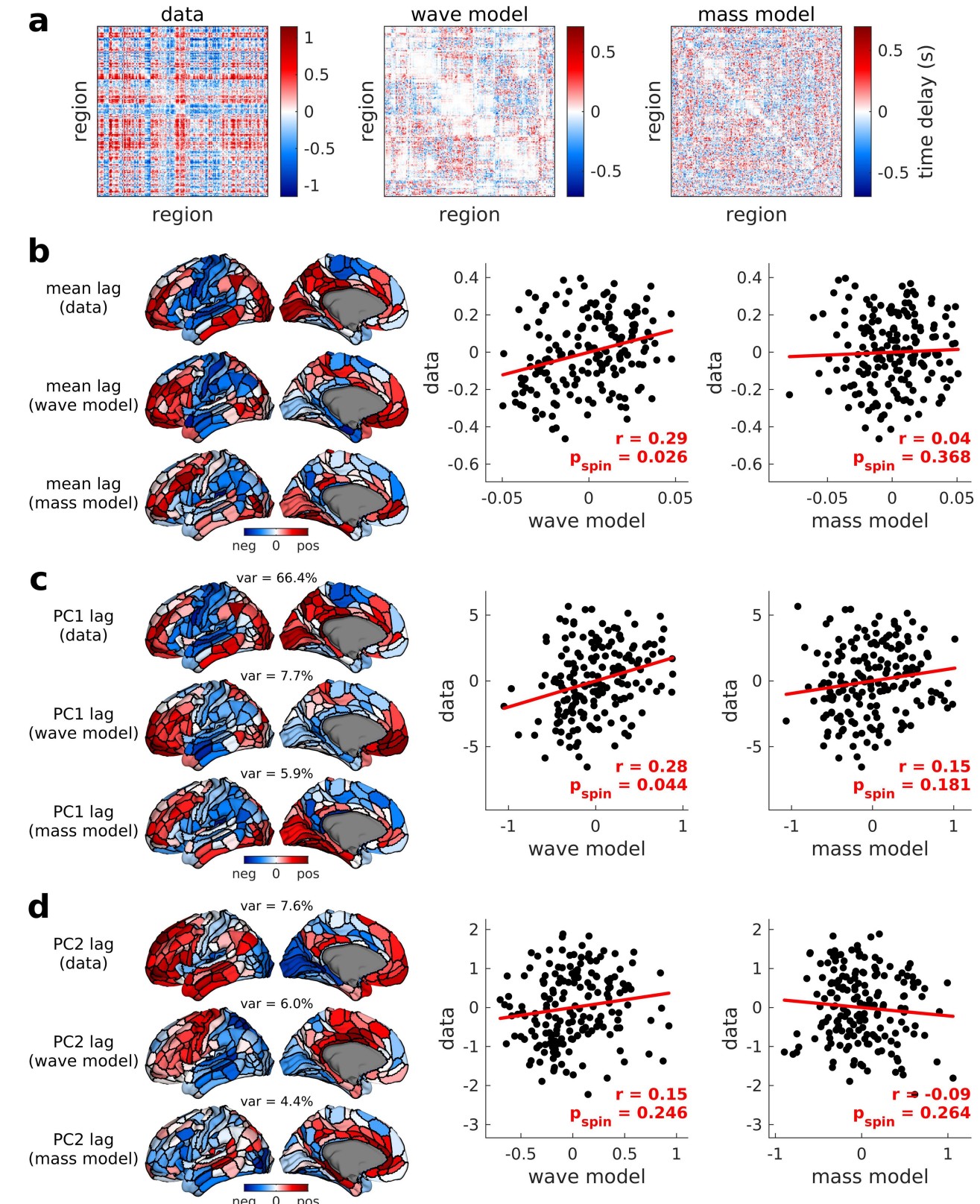

**Extended Data Fig. 9 | Comparison of the wave and neural mass models in capturing time-lagged properties of fMRI data. a**, Time-delay matrices from empirical data, simulated data using the wave model and simulated data using the neural mass model of the left hemisphere. Negative–zero–positive values are coloured as blue–white–red. **b**, Mean lags from the matrices in **a** (mean of each column) projected on the cortical surface. Negative–zero–positive values are coloured as blue–white–red. The scatter plots show the relationship of

mean lags from empirical data and simulated data from the two models for 180 brain regions. The red line represents a linear fit with Pearson correlation coefficient *r* and one-sided spin-test *p*-value, $p_{spin}$, estimated from 10,000 permutations. **c**, Similar to **b** but on the first principal component (PC1) of the matrices in **a**. The number above the surfaces (var) corresponds to the variance explained by the PC. **d**, Similar to **c** but on the second PC (PC2) of the matrices in **a**.

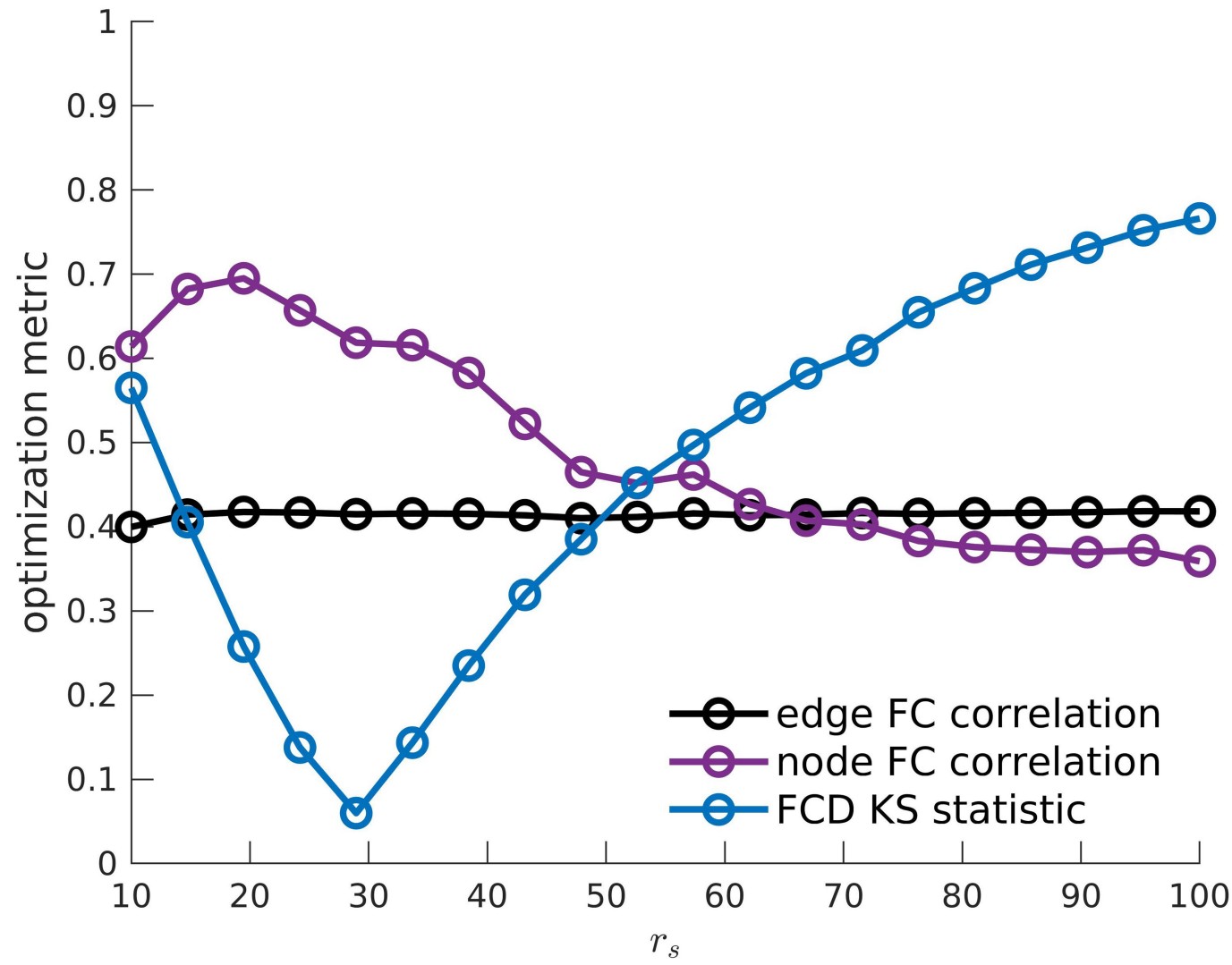

**Extended Data Fig. 10 | Optimization of the wave model.** The model is trained on 125 HCP individuals to find the optimal value of the parameter $r_s$ (in mm). Optimization performance compares data and model FC based on the following metrics: edge FC correlation, node FC correlation and FCD KS statistic. Higher edge FC correlation, higher node FC correlation and lower FCD KS statistic correspond to better model fit. We take $r_s$ = 28.9 mm as the optimal parameter as it leads to the minimum FCD KS statistic value.

# Reporting Summary

## Statistics

For all statistical analyses, confirm that the following items are present in the figure legend, table legend, main text, or Methods section.

| n/a | Confirmed | |
|---|---|---|
| ☐ | ☒ | The exact sample size ($n$) for each experimental group/condition, given as a discrete number and unit of measurement |
| ☐ | ☒ | A statement on whether measurements were taken from distinct samples or whether the same sample was measured repeatedly |
| ☐ | ☒ | The statistical test(s) used AND whether they are one- or two-sided *Only common tests should be described solely by name; describe more complex techniques in the Methods section.* |
| ☐ | ☒ | A description of all covariates tested |
| ☐ | ☒ | A description of any assumptions or corrections, such as tests of normality and adjustment for multiple comparisons |
| ☐ | ☒ | A full description of the statistical parameters including central tendency (e.g. means) or other basic estimates (e.g. regression coefficient) AND variation (e.g. standard deviation) or associated estimates of uncertainty (e.g. confidence intervals) |
| ☐ | ☒ | For null hypothesis testing, the test statistic (e.g. $F$, $t$, $r$) with confidence intervals, effect sizes, degrees of freedom and $P$ value noted *Give P values as exact values whenever suitable.* |
| ☒ | ☐ | For Bayesian analysis, information on the choice of priors and Markov chain Monte Carlo settings |
| ☒ | ☐ | For hierarchical and complex designs, identification of the appropriate level for tests and full reporting of outcomes |
| ☐ | ☒ | Estimates of effect sizes (e.g. Cohen's $d$, Pearson's $r$), indicating how they were calculated |

*Our web collection on statistics for biologists contains articles on many of the points above.*

## Software and code

Policy information about availability of computer code

| Data collection | Analyses were undertaken on human neuroimaging data from the open-access dataset of the Human Connectome Project (HCP) at https://db.humanconnectome.org/. See Van Essen et al. (2013, NeuroImage) and Barch et al. (2013, NeuroImage) for further details. |
|---|---|
| Data analysis | Custom-written computer codes (MATLAB R2019b; Python 3.7) to calculate the eigenmodes, analyse results, perform other computational analyses, and reproduce the figures of the study are openly available at the GitHub repository https://github.com/NSBLab/BrainEigenmodes.<br><br>Additionally, the following open-source software packages were used in the study:<br>- FreeSurfer (v7.1.1)<br>- FSL (v6.0.4)<br>- gmsh (v3.0.3)<br>- connectome workbench (v1.5.0)<br>- HCP minimal preprocessing pipeline (Glasser et al., 2013, NeuroImage)<br>- HCP diffusion preprocessing pipeline (v3.19.0)<br>- Python libraries Nilearn, LaPy |

For manuscripts utilizing custom algorithms or software that are central to the research but not yet described in published literature, software must be made available to editors and reviewers. We strongly encourage code deposition in a community repository (e.g. GitHub). See the Nature Portfolio guidelines for submitting code & software for further information.

## Data

Policy information about availability of data

All manuscripts must include a data availability statement. This statement should provide the following information, where applicable:

- Accession codes, unique identifiers, or web links for publicly available datasets
- A description of any restrictions on data availability
- For clinical datasets or third party data, please ensure that the statement adheres to our policy

Raw and preprocessed MRI data were taken from the open-access Human Connectome Project (HCP) dataset at https://db.humanconnectome.org/. See Van Essen et al. (2013, NeuroImage) and Barch et al. (2013, NeuroImage) for further details. The dataset was acquired by the WU-Minn HCP consortium, with access provided to anyone agreeing to their data use terms.

10,000 human task-activation maps were taken from the open-access repository of NeuroVault (http://neurovault.org/).

All source data to generate the results of the study are openly available at Github repository https://github.com/NSBLab/BrainEigenmodes and the open science framework repository https://osf.io/xczmp/.

# Field-specific reporting

Please select the one below that is the best fit for your research. If you are not sure, read the appropriate sections before making your selection.

☒ Life sciences    ☐ Behavioural & social sciences    ☐ Ecological, evolutionary & environmental sciences

For a reference copy of the document with all sections, see nature.com/documents/nr-reporting-summary-flat.pdf

# Life sciences study design

All studies must disclose on these points even when the disclosure is negative.

| Sample size | No new data were collected. We used data from 255 participants from the HCP dataset, which is the largest available unrelated sample with complete functional MRI data. |
|---|---|
| Data exclusions | Out of 1113 available human participants from the HCP dataset, we excluded participants that are related either as twins or siblings and do not have a complete set of task-evoked and resting-state functional MRI data. |
| Replication | N/A |
| Randomization | Participants were not allocated to experimental groups. Hence, randomization into groups was not applicable. |
| Blinding | Participants were not allocated to experimental groups and the study did not involve hypothesis testing. Hence, blinding was not applicable. |

# Reporting for specific materials, systems and methods

We require information from authors about some types of materials, experimental systems and methods used in many studies. Here, indicate whether each material, system or method listed is relevant to your study. If you are not sure if a list item applies to your research, read the appropriate section before selecting a response.

### Materials & experimental systems

| n/a | Involved in the study |
|---|---|
| ☒ | ☐ Antibodies |
| ☒ | ☐ Eukaryotic cell lines |
| ☒ | ☐ Palaeontology and archaeology |
| ☒ | ☐ Animals and other organisms |
| ☐ | ☒ Human research participants |
| ☒ | ☐ Clinical data |
| ☒ | ☐ Dual use research of concern |

### Methods

| n/a | Involved in the study |
|---|---|
| ☒ | ☐ ChIP-seq |
| ☒ | ☐ Flow cytometry |
| ☐ | ☒ MRI-based neuroimaging |

## Human research participants

Policy information about studies involving human research participants

| Population characteristics | All data were taken from the open-access HCP dataset. The participants comprised 255 healthy young adults (ages 22-35; 132 females). |
|---|---|

| Recruitment | See van Essen et al. (2013, NeuroImage) for recruitment details. |
| Ethics oversight | The open-access HCP dataset were acquired by the WU-Minn HCP consortium with local oversighting ethics committee approval and shared with the authors according to HCP's data use terms. All our procedures were carried out in accordance with protocols set by these data use terms. See van Essen et al. (2013; NeuroImage) for details of oversighting ethics committee. |

Note that full information on the approval of the study protocol must also be provided in the manuscript.

# Magnetic resonance imaging

## Experimental design

| Design type | No new data were collected. See van Essen et al. (2013, NeuroImage) and Barch et al. (2013, NeuroImage) for details. |
| Design specifications | No new data were collected. See van Essen et al. (2013, NeuroImage) and Barch et al. (2013, NeuroImage) for details. |
| Behavioral performance measures | No new data were collected. See van Essen et al. (2013, NeuroImage) and Barch et al. (2013, NeuroImage) for details. |

## Acquisition

| Imaging type(s) | functional MRI and diffusion MRI |
| Field strength | 3T |
| Sequence & imaging parameters | For functional MRI data (van Essen et al., 2013, NeuroImage; Barch et al., 2013, NeuroImage): gradient-echo EPI, FOV of 208x180 mm^2, matrix size of 104x90, isotropic voxel size of 2 mm, TR of 1720 ms, TE of 33.1 ms, flip angle of 52 deg.<br><br>For diffusion MRI data (van Essen et al., 2013, NeuroImage; Barch et al., 2013, NeuroImage): spin-echo EPI, FOV of 210x210 mm^2, matrix size of 140x140, isotropic voxel size of 1.25 mm, TR of 5520 ms, TE of 89.5 ms. |
| Area of acquisition | Whole-brain scan |

Diffusion MRI ☒ Used ☐ Not used

Parameters | 90 diffusion directions, b-weightings of 1000, 2000, 3000 s/mm^2, 174 slices

## Preprocessing

| Preprocessing software | Already preprocessed MRI data were used based on the HCP minimal preprocessing pipeline and HCP diffusion preprocessing pipeline (Glasser et al., 2013, NeuroImage). Preprocessing involved a combination of tools from FSL, FreeSurfer, and Connectome Workbench. |
| Normalization | Already normalized MRI data were used based on the HCP minimal preprocessing pipeline (Glasser et al., 2013, NeuroImage). Normalization included spatially alignment to the MNI standard space using FSL FNIRT and then data were projected onto the cortical surface. |
| Normalization template | All data were projected onto the fsLR cortical template surface with 32,492 vertices per hemisphere. |
| Noise and artifact removal | Already preprocessed MRI data were used based on the HCP minimal preprocessing pipeline and HCP diffusion preprocessing pipeline (Glasser et al., 2013, NeuroImage). Preprocessing of resting-state fMRI data included ICA-FIX. |
| Volume censoring | No volume censoring was performed. |

## Statistical modeling & inference

| Model type and settings | Pearson correlation was used in Figs. 4b, 5a, 5b, 5c, 5d, 5e, 5f and Extended Data Figs. 9b, 9c, 9d.<br><br>Spearman rank correlation was used in Fig. 4e and Supplementary Fig. 11.<br><br>Kolmogorov-Smirnov test was used in Fig. 4b. |
| Effect(s) tested | For Pearson correlation:<br>Association between edge and node FC properties of model (simulated) and empirical data (Fig. 4b).<br>Association between the first three functional gradients and geometric eigenmodes (Figs. 5a, 5b, 5c).<br>Association between all pairs of functional gradients and geometric eigenmodes (Figs. 5d, 5e, 5f).<br>Association between lag properties of model (simulated) and empirical resting-state time series data (Extended Data Figs. 9b, 9c, 9d).<br><br>For Spearman rank correlation:<br>Association between T1w:T2w and response time to peak across regions of interest (Fig. 4e).<br>Association between T1w:T2w and response time to peak across the whole brain (Supplementary Fig. 11). |

For Kolmogorov-Smirnov test:
Comparison of the distribution of FCD statistics of model (simulated) and empirical data (Fig. 4b).

Specify type of analysis: ☐ Whole brain ☐ ROI-based ☒ Both

Anatomical location(s)    The following atlas-based parcellations were used:
HCP-MMP1 atlas (Glasser et al., 2016, Nature) and Schaefer atlas (Schaefer et al., 2018, Cerebral Cortex).

Statistic type for inference    No inferences were made.
(See Eklund et al. 2016)

Correction    No corrections were performed.

## Models & analysis

| n/a | Involved in the study |
|-----|-----|
| ☐ ☒ | Functional and/or effective connectivity |
| ☒ ☐ | Graph analysis |
| ☒ ☐ | Multivariate modeling or predictive analysis |

Functional and/or effective connectivity    Pearson correlation.

