## [Peer Review File · Nature]

Manuscript Title: Geometric constraints on human brain function

Reviewer Comments & Author Rebuttals

Reviewer Reports on the Initial Version:

Referee #1 (Remarks to the Author):

This is a well-written and very interesting paper aimed at identifying the fundamental structural constraints on human brain dynamics.

The overarching premise of this work is that data from spontaneous and task-evoked recordings in the human neocortex can be more parsimoniously explained by eigenmodes derived from cortical geometry rather than those obtained from connectivity.

The authors derive a set of geometric eigenmodes from a triangular mesh tessellation to which a Laplace-Beltrami operator is applied from which a basis set of geometric eigenmodes are obtained. They then show that this basis set is well suited to capturing both task-evoked and spontaneous activity. By capturing, it is meant, that the basis set can represent a wide range of activation maps and spontaneous activity (after generating a connectivity matrix from mode-reconstructed image frames over time).

The authors demonstrate that cortical geometric eigenmodes form a compact basis set to recapitulate both task-evoked and spontaneous activity and that such activities are dominated by long-wavelength large-scale eigenmodes.

Overall, this is a potentially impactful paper that may contain a very interesting finding, but there are a few points that should be clarified and/or places where alternative explanations should be considered.

1. The primary conclusion of this work is that the geometric basis set the authors have derived can recapitulate evoked and spontaneous maps and this is interpreted as meaning that geometry derives function. This may well be the case, but a simpler explanation is that almost any basis set (with enough terms) can describe these maps (consider a Fourier basis set for example) and being able to recapitulate evoked and spontaneous maps with different basis sets doesn't imply that brain activity is governed by a particular basis set. If this is the case, then the conclusions might need to be tempered or at least alternative interpretations discussed.

2. Similarly, the fact that much of the evoked- and spontaneous- maps can be derived with a limited number of terms and that adding higher order terms only has minimal impact on the accuracy of the maps is also consistent with how a Fourier basis set behaves (and what the power spectra of an MRI image looks like). The lowest order terms (low frequency information in the Fourier basis), captures most of the image – the higher order terms (high frequency components in FT) capture sharp details and edge information. The zeroth component of the FT (center of k-space) reflects the integration

(power) of the entire image and is by far the largest component. Collecting a bigger image matrix means adding higher order Fourier terms which provides more and more image detail but most of the image can be captured with only a small set of terms.

3. That much power is concentrated in the first 50 modes is interpreted as meaning the brain communicates at wavelengths greater than $\sim 60\text{mm}$, and this is further interpreted as evidence that we should change how we think about brain activity. This finding would also be present with a Fourier basis set. The low frequency data varies slowly over large spatial regions – high frequency Fourier components introduce details to the image and sharp edges. (This is easily demonstrated by taking the 2D FT of an MR image. Most of the power in k-space will be at zero (the center of k-space is the zeroth order of the Fourier basis set – just like the first geometric mode). If a Gaussian filter is applied to reduce power at high frequencies (or just truncate the k-space data to a 32×32 matrix) and a 2DFT is used to convert this data back into image space a blurred image will result (that's using the low order Fourier basis to reconstruct an image – like using the first 60 geometric modes). As the filtering is reduced (say a wider and wider Gaussian filter) the image will improve but by smaller and smaller amounts as more details are added (just like what is shown in Fig 3). If the opposite filter is applied to the k-space data to remove only the low frequencies and Fourier transformed back into image space only the high frequency information (sharp edges in the image) will be visible, and the intensity of the image will be very low. The geometric mode basis set is behaving in exactly this manner).

Given the above comments I believe it would be useful to compare the geometric basis set with a Fourier basis set. After all, the brain data the authors are using is obtained from a Fourier basis set (2DFT (fmri data) or 3DFT (T1 MPRAGE), of k-space data obtained in MRI, with the number of terms dependent upon how far out in k_x and k_y the acquisition sampling reached). Much of the behavior demonstrated in Figures 1-3 will be recapitulated with the Fourier basis set (where modes are replaced by Fourier terms – i.e. how far out you go in k-space). The filtering steps described above would also yield the finding that sequential removal of long-wavelength modes (low frequency k-space lines in the MR data) has a much greater impact on reconstruction accuracy than removal of short-wavelength modes (high frequency k-space data).

4. It may well be that there is something special about this geometric basis set – after all a basis set derived from the 3D brain itself should perhaps be more efficient at reconstructing brain images than the basis sets to which the authors compared their method. One could also hypothesize however, that since the 3D brain is derived from a 3D Fourier transform the Fourier basis will be exact).

5. A comment about basis set efficiency. While MR images are reconstructed using a Fourier basis set, the FT basis set wasn't derived specifically to reconstruct a brain image, and thus there could be more efficient basis sets such as the geometric basis set the authors derive.

I don't mean to be glib, but want to provide a clear example. A geometric basis derived from a picture of a tractor, applied to reconstruct an image of a tractor in this manor may also be more efficient than a more generalized basis set such as the FT – that doesn't mean however that tractor design is governed by geometry – and therefore I urge caution in interpretation of these results. Deriving a basis set from a tractor to reconstruct an image of a tractor is somewhat analogous to

double-dipping in model building (deriving the basis set) and testing (reconstructing a tractor from the basis set).

6. Given the above considerations, there is concern about over-interpretation of the findings when performance of the geometric modes is used to suggest that this can disentangle the contributions to brain dynamics of cortical geometry from structural connectivity, given that similar findings may be shown with a Fourier basis set.

If the authors do compare these results to the Fourier transform, and if they can show that it is in fact the geometry (not the mathematics of how power is distributed in basis sets) driving the activity patterns observed, then this will be a very high impact paper.

Referee #2 (Remarks to the Author):

In this manuscript, Pang et al. compare two competing models for the organization of intrinsic activity in the brain. The first is a proximity-based model where activity propagates along the cortical sheet as a traveling wave, with amplitude decreasing as a function of distance. The second approach models activity as propagating between areas along white matter tracts, which may connect physically-distant areas. The second model underlies common interpretation of functional connectivity and other analyses based on fMRI data; the first model has traditionally been underappreciated as a contributor to large-scale patterns of brain activity. The authors directly compare the explanatory power of these models in existing fMRI data, and describe three main results:

1. Geometrically-based eigenmodes explain activation patterns and functional connectivity better than connectomic-based eigenmodes.

The authors show that geometrically-based eigenmodes can better reproduce functional connectivity from resting state fMRI and unthresholded activation maps from a variety of tasks. Neural field theory and connectome-based harmonics (using the term loosely) are well-established, but this paper is the first attempt, to my knowledge, to directly compare the explanatory power of the two models. However, the differences are not large, especially when the connectome has a higher density. The most plausible explanation is that both processes contribute to the organization of brain activity. Certainly it is known that long-distance white matter pathways exist to facilitate information transfer between physically distant areas, but at the same time, connections within grey matter also allow communication between adjacent areas. The combination of these processes may account for the lack of correspondence between eigenmodes and FCGs in the cortex, as long distance connections may be necessary to account for some of the structure that is observed. In support of this idea, many brain network models based on the structural connectome capture the sensory vs. association structure that lies at either end of the primary FCG.

2. Traveling waves explain dynamics.

The authors show that traveling waves can explain average and dynamic functional connectivity at a level similar to that obtained with connectome-based models. I found this section to be weaker than the rest of the paper, because no traveling waves were actually examined. As the authors state, recent work by Raut et al. and Bolt et al. describes the presence of traveling waves in fMRI data, and it is unclear whether traveling waves based on the geometric eigenmodes can recapitulate their

features. The authors provide a toy example based on the timing of activity in different areas, but since the timing differences are in milliseconds, it is difficult to relate it directly to fMRI, which is the focus of the rest of the paper. The comparisons to average activity maps, average functional connectivity, and even dynamic functional connectivity collapse the relationships across time. I would be very interested to see if the eigenmode model can capture different dynamic aspects of the data. Different brain network models based on the structural connectome are mostly successful in reproducing time-averaged metrics, but differ in their ability to recreate dynamic features^{1,2}. The reports of ‘waves’ of activity in the brain are not actually new—they have been seen for years and in multiple modalities^{3–6}, as have other time-lagged phenomena⁷. These features can provide a more rigorous test for models of the brain’s intrinsic organization than time-averaged features like functional connectivity.

3. Eigenmodes explain FCGs in subcortical structures.

One of the strengths of the study is that the authors examined areas outside of the cortex, and found that geometric constraints could explain the functional connectivity gradients previously reported for these areas. These results are interesting and might be considered in the context of prior work that reported that large, repeated spatiotemporal patterns (in many ways comparable to the traveling waves in this manuscript) sweep across these FCGs⁸. The existence of a common mechanism across multiple brain areas is an elegant aspect of the geometrically-based model.

One of the aspects of this work that requires clarification is whether global signal regression was performed for any of the studies. The global signal itself is a spatiotemporal pattern^{9,10} that is linked to neuromodulatory input and arousal^{11,12}. If global signal regression is not performed, then changes in vigilance or arousal at the onset of a task might be included in the resulting activation maps. The global signal is not necessarily the only large-scale signal related to neuromodulation or arousal, and so even if GSR was performed, the long wave aspect of activity captured in the unthresholded maps could reflect these types of changes. We know that certain areas of the brain respond strongly and specifically to particular stimuli (e.g., whisker barrel in rodents) so it may be that the classical approach of thresholding and the proposed examination of whole brain maps will turn out to provide complementary views of the brain. On the other hand, activation of nearly the entire brain has been observed during task performance¹³, and it may be that the unthresholded maps are capturing sub-threshold task-related activity rather than other ongoing processes.

Overall, this was a well-conducted, carefully-designed study. The authors’ careful examination of choice of parcellation and threshold, along with the use of different datasets, shows their attention to rigor. While I feel that the emphasis on the geometric eigenmode model’s (slightly) better performance in predicting activity patterns and functional connectivity is a bit overstated, it is a welcome counterbalance to the innumerable studies that simply assume a connectome-based model during their analysis and interpretation.

A few minor comments for the authors’ consideration:

Given the dependence on geometry and the curvature of the cortex, I would expect the eigenmode approach to rely heavily on individual geometry. It would be interesting to see if differences in individual eigenmodes explain differences in activation or functional networks across subjects.

The motivation for the very narrow frequency band for FCD (0.04-0.07 Hz) was unclear.

There is no F label in Figure 1.

Regards,
Shella Keilholz

1. Cabral J, Kringelbach ML, Deco G. Functional connectivity dynamically evolves on multiple time-scales over a static structural connectome: Models and mechanisms. *NeuroImage*. 2017;160:84-96. doi:10.1016/j.neuroimage.2017.03.045
2. Kashyap A, Keilholz S. Dynamic properties of simulated brain network models and empirical resting-state data. *Netw Neurosci*. 2019;3(2):405-426. doi:10.1162/netn_a_00070
3. Majeed W, Magnuson M, Hasenkamp W, et al. Spatiotemporal dynamics of low frequency BOLD fluctuations in rats and humans. *NeuroImage*. 2011;54(2):1140-1150. doi:10.1016/j.neuroimage.2010.08.030
4. Matsui T, Murakami T, Ohki K. Transient neuronal coactivations embedded in globally propagating waves underlie resting-state functional connectivity. *Proc Natl Acad Sci U S A*. 2016;113(23):1521299113-. doi:10.1073/pnas.1521299113
5. Chan AW, Mohajerani MH, LeDue JM, Wang YT, Murphy TH. Mesoscale infraslow spontaneous membrane potential fluctuations recapitulate high-frequency activity cortical motifs. *Nat Commun*. 2015;6:7738. doi:10.1038/ncomms8738
6. Tong Y, Hocke LM, Fan X, Janes AC, Frederick B deB. Can apparent resting state connectivity arise from systemic fluctuations? *Front Hum Neurosci*. 2015;9. doi:10.3389/fnhum.2015.00285
7. Mitra A, Snyder AZ, Blazey T, Raichle ME. Lag threads organize the brain's intrinsic activity. *Proc Natl Acad Sci U S A*. Published online March 30, 2015:1503960112-. doi:10.1073/pnas.1503960112
8. Yousefi B, Keilholz S. Propagating patterns of intrinsic activity along macroscale gradients coordinate functional connections across the whole brain. *NeuroImage*. 2021;231:117827. doi:10.1016/j.neuroimage.2021.117827
9. Bolt T, Nomi JS, Bzdok D, et al. A parsimonious description of global functional brain organization in three spatiotemporal patterns. *Nat Neurosci*. 2022;25(8):1093-1103. doi:10.1038/s41593-022-01118-1
10. Scholvinck ML, Maier A, Ye FQ, et al. Neural basis of global resting-state fMRI activity. *Proc Natl Acad Sci U A*. 2010;107(22):10238-10243. doi:10.1073/pnas.0913110107
11. Turchi J, Chang C, Ye FQ, et al. The Basal Forebrain Regulates Global Resting-State fMRI Fluctuations. *Neuron*. 2018;97(4):940-952.e4. doi:10.1016/j.neuron.2018.01.032
12. Wong CW, Olafsson V, Tal O, Liu TT. The amplitude of the resting-state fMRI global signal is related to EEG vigilance measures. *NeuroImage*. 2013;83:983-990. doi:10.1016/j.neuroimage.2013.07.057
13. Gonzalez-Castillo J, Saad ZS, Handwerker DA, Inati SJ, Brenowitz N, Bandettini PA. Whole-brain, time-locked activation with simple tasks revealed using massive averaging and model-free analysis. *Proc Natl Acad Sci U A*. 2012;109(14):5487-5492. doi:10.1073/pnas.1121049109

Referee #3 (Remarks to the Author):

1. Summary of the key results

This is a truly fascinating paper, bringing ideas highly influential in the natural sciences (mainly physics) to the domain of systems neuroscience. The authors adopted a neural field perspective on large scale brain dynamics, with the geometry of the cortical sheet imposing boundary conditions that determine a basis set of normal modes. From these normal modes, different aspects of brain activity (e.g. spontaneous, task-evoked) can be derived, crucially with better accuracy than when adopting a similar decomposition based on large-scale connectivity. Perhaps the most surprising key result is that a neural field assuming local connectivity provides a better description of several brain phenomena than a similar description based on the structural connectome.

The specific key results of this manuscript are the following:

- A discrete set of normal modes can be obtained when adopting brain geometry as a boundary condition of a suitable partial differential equation
- This description supersedes a similar one based on the (long range) connectome
- Brain activations correspond to low frequency modes, which implies a global perturbation as opposed to a very localized perturbation
- Traveling waves can be used to describe some of the aforementioned phenomena
- Structures outside the cortex are functionally characterized by the geometric normal modes

2. Originality and significance

This approach is certainly novel. The closest work is that of connectome eigenmodes, but besides some mathematical superficial similarities (e.g. the use of eigenmode decomposition), the interpretation of the findings and their scope is very different.

3. Data & methodology

I have some concerns and commentaries about the methods, and also about the interpretation of the results. I can summarize these in the following key points:

- The exponential decay rule is not valid at the large scale. Locally, both DTI and fMRI FC show an exponential decay with distance from a given seed. However, we also know that this decay is bound to stop after a certain distance, and that connectivity must then start to increase. This is due to the strong homotopic (interhemispheric) connectivity of the brain: even if a region is weakly connected to its neighboring regions within a certain hemisphere, it is most likely strongly connected with its counterpart at the other brain hemisphere. To which extent does this represent a limitation for the proposed methodology, and how can this potential limitation be overcome?

- All reasonable functions can be written as a series of the eigenmodes with appropriate coefficients. This is known since the times of Fourier, when he obtained such decomposition in terms of sines and/or cosines. More complicated geometries will change the precise mathematical form of the eigenmodes, but the result will still be valid. In this sense, the results could be deemed trivial by some (personally I think that even if a result is mathematically trivial, it could be highly relevant within a certain neurobiological context, as in this particular case). However, I feel that because of this, the comparison between the functional and connectome eigenmodes is not entirely fair. Any

function with the cortex as its domain can be written as a suitable expansion of the corresponding functional eigenmodes (by construction), but the connectome eigenmodes do not span the same space (they were obtained in a different problem with different boundary conditions) and hence it is not surprising (at least in my opinion) that they are less accurate in this context compared to the functional eigenmodes.

- I am also wondering about the utility of this approach beyond determining the locality of certain patterns of spontaneous or evoked brain activity. Different tasks and/or brain states result in clearly different patterns of brain activation without changes to brain geometry, therefore the same set of basis functions should be useful to describe them. Based on my understanding of this article, the relevant information is encoded in the coefficients of this expansion, which in turn signal the relative "energy" (wavelengths) needed for this expansion to be accurate. This translates to considerations about locality of the patterns (i.e. is only the first coefficients are needed, this implies low spatial frequencies). While certainly interesting, I also find that this information could be obtained without resorting to such complicated methods, simply focusing on the spatial smoothness of the patterns. Regarding this point, the fMRI volumetric data can be subject to a standard 3D Fourier decomposition, and I wonder whether the proposed method gives information that is not available from this representation.

4. Appropriate use of statistics and treatment of uncertainties

I do not have any concerns regarding the use of statistics in this manuscript.

5. Conclusions: robustness, validity, reliability

The conclusions are generally valid and robust, except when they depend on some of the concerns I raised above. For instance, the authors dismiss the utility of long-range connectomics to describe whole-brain activity patterns in contrast to their own approach because of the comparison between functional and connectome eigenmodes. Yet as I said, I do not think this is an entirely fair comparison, since the functional eigenmodes are guaranteed to allow the accurate reconstruction of any function on the cortical mesh, by construction, while this is not true of the connectome decomposition.

6. Suggested improvements

I suggest the following new analysis/changes to improve the manuscript:

- Explore the consequences of including long range homotopic connections in the model, or provide an argument (most likely based on considerations about symmetry?) showing that this is not a concern for this approach.
- Check whether the proposed method is capable of providing more information than a simple 3D Fourier transform of the, e.g., activation patterns, without any consideration to boundary conditions and/or the geometry of the cortical sheet.

- Tone down the comparison between functional and connectome eigenmodes, or provide a counter-argument to show that my concern of a biased comparison is not valid.

7. References

No comments in this regard.

8. Clarity and context: lucidity of abstract/summary, appropriateness of abstract, introduction and conclusions

The manuscript is clearly written and will be of interest to an ample and general audience. Figures are beautiful and informative.

9. Final comment

I think this is a thought-provoking and original article, and that it is suitable for publication in Nature. My concerns and comments should not be interpreted as criticism to prevent the publication of this manuscript, but as a natural part of a reviewing process aimed to improve the published work, the robustness of its results, and the scope of its conclusions. I will happily endorse the publication of a revised version of this manuscript after these concerns are adequately addressed.

Referee #4 (Remarks to the Author):

This is a very well-written manuscript focusing on the study of brain physiological organization principles, in particular investigating the role of pure geometry -as opposed to neural bundle architecture- in shaping dynamics of brain activity. To do so, brain data from Magnetic Resonance Imaging (MRI) taken from existent online databases of healthy subjects are employed. The key findings reported by the authors are twofold: 1) Brain activity could be equally well or better explained by modes based on brain geometry, than by complex white matter architecture of neural connections; 2) Mechanisms driving brain activity propagation follow simple wave-like dynamics, rather than complex neural mass models that were employed so far.

The exploration proposed is novel and the topic approached - mechanisms of brain organization- is in principle of high relevance for a broad audience.

The overall manuscript quality appears excellent: clearly written and very well presented (high-quality graphical elements efficiently summarizing results and analyses); the analyses performed are sound; statistical analysis is appropriate; previous work was correctly referenced.

The study encompasses a very large body of analyses, which are per se very interesting for the field, to show the relevance of geometric constrains in brain functioning. However, I have some concerns about the very strong claims the authors report in favor of a geometry-centric view that completely discard the role of white matter architecture in brain neural dynamics. I think therefore that these claims should be reduced, in light of the following remarks. I also think this would not affect the

validity, nor the interest of the experiments performed.

Since we know that neural transmission happens through white matter intricate connections, here the interest should lie more in investigating which *kind* of information present in the fMRI signal can be justified by brain geometry, and which one cannot. This would go beyond the sole comparison of reconstruction accuracies (*amount* of information), which appear anyway fairly similar in the case of geometric and connectome eigenmodes (and depending a lot on how the connectome is constructed). In Fig. 2C and D the authors compare the reconstruction accuracy between connectome and geometric eigenmodes, claiming it higher for the geometric case. However, they do not discuss how initial modes from structural connectomes -the ones with more energy- appear to perform better than geometric ones (e.g. up to the first 15/20). This goes well with the hypothesis that SC modes would, as expected, capture large-scale distributed networks, “function-wise” important, better than geometric ones present with the same energy in the functional data.

Further exploration of this matter would help in interpreting which portion of functional activity measured via fMRI can be explained well by geometry and which one cannot, since more dependent on the white matter architecture.

In Fig. 4C the authors show the propagation of activity with the wave model after stimulation of the visual cortex, bringing it as an evidence to claim that stimulus-evoked cortical activity would be solely driven by geometry and not by “complex patterns of connectivity”. Since we know from previous literature that white matter anatomy in the visual cortex indeed accounts for directionality and propagation of neural activation, it would seem more fair to say that geometry is enough to explain (a big portion of) what we record at the level of fMRI signals, rather than it is enough to justify thoroughly neural activation processes in the brain. In this regard, it would be interesting to see how the wave model performs in explaining more complex high-level cognitive processes involving widely distributed networks, which supposedly would require long-range neural connections.

The authors limit the exploration at $N=200$ eigenmodes throughout the study, but this is very few w.r.t the full decomposition. Low frequency eigenmodes are known to explain most of the energy in the signal, but it would be interesting to see what happens also at higher frequency modes, if they have a role in explaining individual features of functional activity, as it is known to happen for connectome modes, or not. In other words, the reconstruction with $N=200$ modes reaches around 80-90% accuracy depending on the task, but what is the remaining part of the signal, not explainable with geometric low frequency modes, therefore not directly inferable from geometry?

This would also help disentangling the part of information that is common to the population, possibly driven by the task and comprising low-frequency modes, and the one that is more peculiar to different individuals.

About the FC reconstruction with the two models (Fig. 4), authors claim the wave model performs better than the neural mass model, but a clear benefit cannot be appreciated nor when looking visually at the matrix, nor when comparing correlation values (edge-wise correlation of 0.4). In addition to the metrics proposed to evaluate the models, it would be interesting to see if the FC main blocks are preserved in these reconstructions.

In Figure S7, authors assess the effect of thresholding the connectome at different densities, showing that this has an impact on the reconstruction accuracy measure. First, we can indeed expect methodological choices in the construction of the connectomes to affect the results, so the effect of these variables – such as the tractography algorithms used and the type of measure taken as connectivity metric, in this case the number of streamlines not normalized for regional volumes – should be considered. Second, the density analysis is not completely clear to me: fig. S7 shows that lower density connectomes lead to better reconstruction accuracy. How do the authors explain then that between the two connectomes considered in the main analysis (Fig. 2), that should only differ in density, the one at higher density (1.55% vs 0.1%) performs better?

Response to Reviewer 1

Comment 1.0: *This is a well-written and very interesting paper aimed at identifying the fundamental structural constraints on human brain dynamics.*

The overarching premise of this work is that data from spontaneous and task-evoked recordings in the human neocortex can be more parsimoniously explained by eigenmodes derived from cortical geometry rather than those obtained from connectivity.

The authors derive a set of geometric eigenmodes from a triangular mesh tessellation to which a Laplace-Beltrami operator is applied from which a basis set of geometric eigenmodes are obtained. They then show that this basis set is well suited to capturing both task-evoked and spontaneous activity. By capturing, it is meant, that the basis set can represent a wide range of activation maps and spontaneous activity (after generating a connectivity matrix from mode-reconstructed image frames over time).

The authors demonstrate that cortical geometric eigenmodes form a compact basis set to recapitulate both task-evoked and spontaneous activity and that such activities are dominated by long-wavelength large-scale eigenmodes.

Overall, this is a potentially impactful paper that may contain a very interesting finding, but there are a few points that should be clarified and/or places where alternative explanations should be considered.

Response 1.0: We thank the Reviewer for their feedback and appreciation of our work. We will address the points raised by the Reviewer in the below responses.

Comment 1.1: *The primary conclusion of this work is that the geometric basis set the authors have derived can recapitulate evoked and spontaneous maps and this is interpreted as meaning that geometry derives function. This may well be the case, but a simpler explanation is that almost any basis set (with enough terms) can describe these maps (consider a Fourier basis set for example) and being able to recapitulate evoked and spontaneous maps with different basis sets doesn't imply that brain activity is governed by a particular basis set. If this is the case, then the conclusions might need to be tempered or at least alternative interpretations discussed.*

Response 1.1: The Reviewer raises an important point, which we now address below in three ways. First, we present the requested comparison with several Fourier basis sets, which shows the superiority of geometric eigenmodes. Second, we outline mathematical reasons why Fourier basis sets are poorly suited for the present context and why geometric eigenmodes are more appropriate. Third, we clarify that our goal is not to derive an optimal basis set but to compare different anatomically constrained basis sets to understand how brain structure drives dynamics.

(i) Geometric eigenmodes outperform Fourier basis sets

To address the Reviewer's comment, we compared the performance of the geometric eigenmodes with several Fourier basis sets in decomposing empirical spatiotemporal fMRI data on the cortex. Specifically, we constructed six different real-valued forms of the spatial Fourier basis set using sines and/or cosines in 3D Euclidean space combined in various combinations with either regularly or irregularly spaced spatial frequencies (see new Supplementary Material-S9 or the revisions in the text shown below for details). The resulting spatial modes are shown in the new Fig. S14A, reproduced below. Alternative implementations are

possible, but we chose these six because they span the range of simple and reasonable choices one can make, they produce real-valued spatial modes, and they can be compared with the geometric eigenmodes in a straightforward way. We then used these Fourier basis sets to reconstruct task-activation maps and resting-state data (similar to Fig. 1D) and compared their reconstruction accuracies with the geometric eigenmodes. The new Fig. S14B shows that geometric eigenmodes substantially outperform the Fourier basis sets in reconstructing both task-evoked and resting-state fMRI data. This new analysis strengthens our conclusions that geometry drives patterns of macroscale activity and that the accurate representation provided by its eigenmodes is not a trivial result that can be generated by any basis set expansion.

We have added the following text to Supplementary Material-S9 and Results and Discussion sections to discuss the above results:

Supplementary Material Lines 370–418: To further confirm that the performance of geometric eigenmodes is not trivially driven by how the mathematics of any basis set expansion, we compared the geometric eigenmodes to six simple spatial Fourier basis sets based on combinations of sines and/or cosines. The Fourier basis sets were constructed using the following real-valued functions:

$$F_1 := \cos\left(\frac{2\pi(j_x - 1)x}{L_x} + \frac{2\pi(j_y - 1)y}{L_y} + \frac{2\pi(j_z - 1)z}{L_z}\right), \quad (S8)$$

$$F_2 := C_1 \cos\left(\frac{2\pi(j_x - 1)x}{L_x}\right) + C_2 \cos\left(\frac{2\pi(j_y - 1)y}{L_y}\right) + C_3 \cos\left(\frac{2\pi(j_z - 1)z}{L_z}\right), \quad (S9)$$

$$F_3 := C_1 \cos\left(\frac{2\pi(j_x - 1)x}{L_x} + \frac{2\pi(j_y - 1)y}{L_y} + \frac{2\pi(j_z - 1)z}{L_z}\right) + C_2 \sin\left(\frac{2\pi(j_x - 1)x}{L_x} + \frac{2\pi(j_y - 1)y}{L_y} + \frac{2\pi(j_z - 1)z}{L_z}\right), \quad (S10)$$

where j_x, j_y, j_z are integer constants, x, y, z are the spatial positions of each point on the spherical surface mesh representation of the cortex, L_x, L_y, L_z are the periods in each direction, and C_1, C_2, C_3 are fitting constants. For each function, we constructed a regular and an irregular version, resulting in six basis sets. The regular version corresponds to when $j_x = j_y = j_z = j$ such that the spatial wavelengths of mode j in the x -, y -, and z -directions are regularly spaced and increase by $\frac{2\pi}{L_x}, \frac{2\pi}{L_y}, \frac{2\pi}{L_z}$ as the mode number increases, which is a standard implementation in Fourier analysis. The irregular version corresponds to when j_x, j_y, j_z are integer combinations and not necessarily equal; hence, the spatial wavelengths in the x -, y -, and z -directions are irregularly spaced. We implemented this version because it affords the Fourier basis sets greater freedom and thus the best possible chance of performing well. To associate a (j_x, j_y, j_z) combination to a single mode j , we arranged the combinations in order of increasing $j_x + j_y + j_z$. For example, the (j_x, j_y, j_z) combination for the first ten modes follows the set $\{(1,1,1), (1,1,2), (1,2,1), (2,1,1), (1,2,2), (2,1,2), (2,2,1), (1,1,3), (1,3,1), (3,1,1)\}$. These two versions allow us to explore how the spatial wavelengths can affect the decompositions of the Fourier basis sets, given that the spatial wavelengths of the geometric eigenmodes are irregularly spaced (see Table S1).

Next, we defined the periods to be $L_i := [\max(i) - \min(i)]$ for $i = x, y, z$, to have a heuristic that respects the shape of the brain. In addition, the -1 in $(j_i - 1)$ ensures that the first mode is constant to make it comparable to the first geometric mode. The forms in Eqs. (S9) and (S10) have more degrees of freedom, with the fitting constants, C_1, C_2, C_3 , estimated separately, allowing us to weight the cosine and/or sine

functions differently. Therefore, mode j now has multiple amplitudes to be estimated during the decomposition in Eq. (S4) instead of just one. These added degrees of freedom result in an increased model complexity relative to geometric eigenmodes; more specifically, the Fourier basis sets formed by Eqs. (S9) and (S10) involve estimating 2 and 3 coefficients per mode, respectively, whereas geometric eigenmodes require fitting just one coefficient per mode.

Although other forms and complicated choices can be made to construct a Fourier basis set, the above choices are well motivated by their simplicity, yielding unique, real-valued spatial modes, and cover the key implementation choices one could make. Figure S14A shows the spatial profiles of the resulting modes with unit coefficients, which we used to reconstruct task-activation maps and resting-state data (similar to Fig. 1D). Figure S14B shows that geometric eigenmodes significantly outperform the Fourier basis sets in reconstructing both task-evoked and resting-state data, further emphasizing that the accurate representation provided by geometric eigenmodes is not a trivial result of any basis set expansion. Moreover, this conclusion remains regardless of how the spatial wavelengths of the modes in the x-, y-, and z-directions are defined. To reiterate, even though the Fourier basis sets based on Eqs. (S9) and (S10) have more degrees of freedom to fit the data when compared to geometric eigenmodes, geometric eigenmodes still show superior performance, underscoring their parsimony in accounting for brain dynamics.

Main Text Lines 239–242: We additionally find that geometric eigenmodes show stronger out-of-sample generalization than principal components of the functional data itself (calculated via principal component analysis (PCA); Supplementary Material-S8 and Figs. S12 and S13) and better performance than Fourier spatial basis sets (Supplementary Material-S9 and Fig. S14).

Main Text Lines 445–449: The extensive comparisons of geometric eigenmodes with other anatomical (connectome and EDR eigenmodes) and mathematical (PCA and Fourier) basis sets show that their superior performance in capturing macroscale neocortical activity is not trivially driven by generic mathematical properties of basis set expansions. Rather, this result indicates that geometry represents a fundamental anatomical constraint on dynamics.

New Fig. S14:

Fig. S14. Comparison of geometric eigenmodes and Fourier basis sets. (A) Spatial maps of modes 1, 2, 3, 4, 10, 50, and 100 of six different Fourier basis sets with unit coefficients. The terms reg and irreg mean that the spatial wavelengths of the modes in the x-, y-, and z-directions are spaced in regular and irregular increments, respectively. See Supplementary Material-S9 for details. **(B)** Reconstruction accuracy of 7 key HCP task-contrast maps and resting-state FC. See Section S4.2 and Table S2 for details about the contrast maps. wm = working memory.

(ii) Fourier basis sets are mathematically inappropriate in the present context

The poor performance of the Fourier basis sets shown above speaks to some of the fundamental motivations for using geometric eigenmodes as a basis set of brain activity. The construction of a basis set relies on the appropriate treatment of the boundary conditions of the system under investigation. Fourier basis sets work well for objects with regular shapes (e.g., rectangle) and for functions that can be defined on such shapes (e.g., 2D images). As evident in the new Fig. S14, Fourier decompositions are poorly suited to non-regularly shaped objects, such as the cortical surface, which are actually defined on a Riemannian manifold. Moreover, the cortical surface has intrinsic boundary conditions (e.g., boundary conditions for the medial wall that cannot be written in simple functions), which cannot be satisfied by Fourier basis sets (Do Carmo, 1976). For such objects, our approach using the Laplace-Beltrami operator (LBO) for deriving the eigenmodes is more appropriate, based on well-established theories in the fields of differential geometry and physics (Chavel, 1984; Klingenberg, 1995; Reuter, 2006).

We have added the following text to Supplementary Material-S9 to discuss the above points:

Supplementary Material Lines 420–433: We present the above analysis of Fourier basis sets for completeness and to demonstrate that such functions cannot accommodate the boundary conditions of non-regularly shaped objects, such as the cortical surface (e.g., Dirichlet or Neumann conditions for the medial wall)⁶¹. This is because Fourier basis sets can only be properly constructed for objects with regular shapes (e.g., rectangle) and are more suitable to be used for analyzing functions defined on such shapes (e.g., 2D images). Hence, for the cortical surface defined on a Riemannian manifold, using a Fourier basis set is generally not well motivated and using the eigenmodes of the Laplace-Beltrami operator is more appropriate^{62,63}. In fact, mathematically speaking, LBO eigenmode analysis on the Riemannian manifold is considered a generalization of Fourier analysis⁶⁴. The same applies to discrete networks, such as a connectome; for this reason, the eigendecomposition of the graph Laplacian (as done in the work) is commonly used in the field of spectral graph theory as an alternative to a classical Fourier transform⁶⁵. Thus, whilst it is possible to decompose spatial maps of brain activity using Fourier basis sets, they are highly inefficient and poorly suited to the current problem. We therefore do not advocate their use.

(iii) Our goal is to understand anatomical constraints on function, not to identify an optimal basis set

Finally, we clarify that the aim of our work is not to find an optimal basis set for decomposing brain maps. Our goal is specifically to investigate physiologically and anatomically grounded basis sets to gain insights into structural constraints on brain function. This is why we focused on comparing the two most physiologically sensible and anatomically grounded basis sets that have been proposed thus far in the literature—geometric eigenmodes and connectome eigenmodes. Contrasting the relative performance of these two basis sets offers insights into which anatomical properties are more fundamental in constraining brain function. Fourier basis sets are derived from abstract mathematical functions that are not directly related to brain anatomy; thus, they cannot be used to understand structural constraints on brain function. A statistically optimal basis set (in the linear regime) is provided by PCA, but this is purely phenomenological, characterizing the functional data itself without offering insights into the relationship between structure and function. In Supplementary Material-S8 and Figs. S9 and S10 of the original manuscript (currently Supplementary Material-S8 and Figs. S12 and S13 of the revised manuscript), we presented a comparison with PCA and showed that geometric eigenmodes provide greater out-of-sample generalizability.

We have added the following text to Supplementary Material-S9 to clarify the above points:

Supplementary Material Lines 433–437: We also note that Fourier basis sets offer no insights into the generative processes underlying brain activity. Our primary focus is to compare physiologically principled and anatomically constrained basis sets (i.e., geometric and connectome eigenmodes) to uncover the critical constraints on brain dynamics, and not to identify statistically optimal basis sets.

Comment 1.2: *Similarly, the fact that much of the evoked- and spontaneous- maps can be derived with a limited number of terms and that adding higher order terms only has minimal impact on the accuracy of the maps is also consistent with how a Fourier basis set behaves (and what the power spectra of an MRI image looks like). The lowest order terms (low frequency information in the Fourier basis), captures most of the image – the higher order terms (high frequency components in FT) capture sharp details and edge information. The zeroth component of the FT (center of k-space) reflects the integration (power) of the entire image and is by far the largest component. Collecting a bigger image matrix means adding higher order Fourier terms which provides more and more image detail but most of the image can be captured with only a small set of terms.*

That much power is concentrated in the first 50 modes is interpreted as meaning the brain communicates at wavelengths greater than ~60mm, and this is further interpreted as evidence that we should change how we think about brain activity. This finding would also be present with a Fourier basis set. The low frequency data varies slowly over large spatial regions – high frequency Fourier components introduce details to the image and sharp edges. (This is easily demonstrated by taking the 2D FT of an MR image. Most of the power in k-space will be at zero (the center of k-space is the zeroth order of the Fourier basis set – just like the first geometric mode). If a Gaussian filter is applied to reduce power at high frequencies (or just truncate the k-space data to a 32x32 matrix) and a 2DFT is used to convert this data back into image space a blurred image will result (that's using the low order Fourier basis to reconstruct an image – like using the first 60 geometric modes). As the filtering is reduced (say a wider and wider Gaussian filter) the image will improve but by smaller and smaller amounts as more details are added (just like what is shown in Fig 3). If the opposite filter is applied to the k-space data to remove only the low frequencies and Fourier transformed back into image space only the high frequency information (sharp edges in the image) will be visible, and the intensity of the image will be very low. The geometric mode basis set is behaving in exactly this manner).

Given the above comments I believe it would be useful to compare the geometric basis set with a Fourier basis set. After all, the brain data the authors are using is obtained from a Fourier basis set (2DFT (fmri data) or 3DFT (T1 MPAGE), of k-space data obtained in MRI, with the number of terms dependent upon how far out in kx and ky the acquisition sampling reached). Much of the behavior demonstrated in Figures 1-3 will be recapitulated with the Fourier basis set (where modes are replaced by Fourier terms – i.e. how far out you go in k-space). The filtering steps described above would also yield the finding that sequential removal of long-wavelength modes (low frequency k-space lines in the MR data) has a much greater impact on reconstruction accuracy than removal of short-wavelength modes (high frequency k-space data).

It may well be that there is something special about this geometric basis set – after all a basis set derived from the 3D brain itself should perhaps be more efficient at reconstructing brain images than the basis sets to which the authors compared their method. One could also hypothesize however, that since the 3D brain is derived from a 3D Fourier transform the Fourier basis will be exact).

Response 1.2: We agree with these qualitative parallels between decomposition of data using geometric eigenmodes and using a Fourier basis set. However, as detailed in Response 1.1, the former is the more appropriate choice for functions (e.g., task activation) expressed along the cortical surface due to the non-regular shape and boundary conditions of the cortical Riemannian manifold, neither of which can be properly accommodated by a Fourier basis set.

We also clarify an important distinction between the example raised by the Reviewer and our present work. The Reviewer is referring to a Fourier decomposition of the entire image volume (in either k-space or physical space). The result of this decomposition will be sensitive to signal variations spanning white matter, grey matter, cerebrospinal fluid, non-brain tissue, the space surrounding the head, and the edges between these different physical objects. The geometric and connectome eigenmodes are defined solely with respect to the anatomical domain of interest—either the continuous cortical sheet (geometric eigenmodes) or a discrete graph model of inter-regional connectivity (connectome eigenmodes). Since we are interested in characterizing maps of brain function, such constraints are critical, since other properties (e.g., the boundary between the head and background space) are not relevant.

We also agree that a Fourier (or any other frequency-dependent) basis set would identify more power in low frequencies. This is because the data itself has more power in low spatial frequencies and is not a trivial mathematical property of our basis set expansion. Our analysis in Figs. S17A and S17B shows that when the

same spectral analysis is performed on surrogate data (zero-centered white noise without smoothing; i.e., FWHM = 0), the resulting modal power spectrum is flat and does not capture the high power in long-wavelength modes seen in the empirical task-activation maps. This result highlights that our findings are not a trivial property of the decomposition itself but are real characteristics of the data, thus supporting the idea that brain activity comprises large-scale spatial patterns.

Comment 1.3: *A comment about basis set efficiency. While MR images are reconstructed using a Fourier basis set, the FT basis set wasn't derived specifically to reconstruct a brain image, and thus there could be more efficient basis sets such as the geometric basis set the authors derive.*

I don't mean to be glib, but want to provide a clear example. A geometric basis derived from a picture of a tractor, applied to reconstruct an image of a tractor in this manner may also be more efficient than a more generalized basis set such as the FT – that doesn't mean however that tractor design is governed by geometry – and therefore I urge caution in interpretation of these results. Deriving a basis set from a tractor to reconstruct an image of a tractor is somewhat analogous to double-dipping in model building (deriving the basis set) and testing (reconstructing a tractor from the basis set).

Response 1.3: Regarding basis set efficiency, we reiterate that our goal here is not to find the most efficient basis set. Instead, we use the basis sets to test the relative importance of geometry or complex inter-regional connectivity in shaping brain activity. Thus, our focus is on comparing the efficacy of the most physiologically plausible and anatomically grounded basis sets that have been proposed thus far in the literature (i.e., geometric and connectome eigenmodes) to understand how anatomy constrains function. We have included text in the revised manuscript to clarify this aim, as outlined in Response 1.1.

We would also like to clarify some key points about the tractor analogy. As phrased in the Reviewer's comment, the analogy is logically equivalent to using anatomical eigenmodes of the brain to reconstruct the anatomy of the surface, or using eigenmodes of a brain activation map to reconstruct the activation map itself. This is not what we do in our analysis. Instead, we use the *anatomical* eigenmodes to reconstruct *spatial patterns of brain activity*. Comparing the relative performance of different types of anatomical eigenmodes, as we do, offers insights into the most fundamental structural constraints on dynamics.

Critically, the concern about double-dipping does not apply here as the geometric eigenmodes were derived from a T1-based template surface (for the cortex) and volume (for the subcortex), which are completely independent of the data sample from which the activity decompositions and reconstructions were performed. Furthermore, only one set of geometric eigenmodes was calculated to explain all functional datasets. The generality and parsimony of this basis set in explaining a hugely diverse empirical dataset (task-evoked and resting-state) is a key novelty of our work. These points are discussed in Supplementary Material-S2.

We have added the following text to Supplementary Material-S2 to briefly clarify the above last point:

Supplementary Material Lines 61–63: This template is independent of the data sample used in all our analyses, thus obviating any concerns about circularity.

Comment 1.4: *Given the above considerations, there is concern about over-interpretation of the findings when performance of the geometric modes is used to suggest that this can disentangle the contributions to*

brain dynamics of cortical geometry from structural connectivity, given that similar findings may be shown with a Fourier basis set.

If the authors do compare these results to the Fourier transform, and if they can show that it is in fact the geometry (not the mathematics of how power is distributed in basis sets) driving the activity patterns observed, then this will be a very high impact paper.

Response 1.4: We trust that the new analyses incorporated into the paper together with the accompanying theoretical considerations provide strong support to address the concerns of the Reviewer.

Response to Reviewer 2

Comment 2.0: *In this manuscript, Pang et al. compare two competing models for the organization of intrinsic activity in the brain. The first is a proximity-based model where activity propagates along the cortical sheet as a traveling wave, with amplitude decreasing as a function of distance. The second approach models activity as propagating between areas along white matter tracts, which may connect physically-distant areas. The second model underlies common interpretation of functional connectivity and other analyses based on fMRI data; the first model has traditionally been under-appreciated as a contributor to large-scale patterns of brain activity. The authors directly compare the explanatory power of these models in existing fMRI data, and describe three main results...*

Response 2.0: We first wish to clarify that the distinction between the two models highlighted by the Reviewer with respect to the substrate through which activity propagates (cortical sheet vs white-matter tracts) is not completely accurate and take this opportunity to address any confusion. In neural field theory (NFT), from which the simple wave model is derived, neuronal activity of points on the cortex also propagates along white-matter tracts, with a connectivity that falls off approximately exponentially with distance, as is observed experimentally (Nunez, 1974) (see Introduction and Supplementary Material-S1 and the new Fig. S18 presented below for a schematic diagram). Thus, both geometric-wave-based and connectome-neural mass-based models account for white-matter connectivity, albeit in different ways; the geometric view approximates connectivity using a homogeneous distance-dependent kernel while also accounting for geometry, whereas the connectome-based view relies on empirical estimates of complex inter-regional connectivity without directly incorporating geometry. We have made this point on Lines 439–441 of the original manuscript (Lines 452–453 of the revised manuscript).

However, we believe that the role of white-matter connectivity in the wave model may not have been clear in our original exposition because we presented the wave equation in differential form ((Eq. (3) in the main text), where the role of white matter connectivity is somewhat obscured. This equation can also be expressed in integral form, where the exponential-distance connectivity dependence is more apparent. The two forms have been shown to be equivalent (Nunez, 1974; Jirsa and Haken, 1996; Robinson et al., 1997), which is a feature of wave equations more generally (Arfken, 1985).

We have added the following text to Results and Supplementary Material-S12.1 to clarify the above points:

Main Text Lines 320–322: Under this model, activity propagates between points on the neocortex through their white-matter connectivity, with a strength that decays approximately exponentially with distance (Supplementary Material-S1 and S12.1).

Supplementary Material Lines 534–563: Note that the propagation of activity between points is governed by their white-matter connectivity, with strength that decays approximately exponentially with distance. This distance-dependence is more apparent when Eq. (S13) is converted into its equivalent integral form as follows. Consider two points on the neocortex at locations \$\mathbf{r}'\$ and \$\mathbf{r}\$ connected by a white-matter tract (see visual schematic in Fig. S18). The activity \$\phi(\mathbf{r}, t)\$ at location \$\mathbf{r}\$ and time \$t\$ can be considered as a spatiotemporal convolution of the source \$Q(\mathbf{r}', t')\$ at location \$\mathbf{r}'\$ and time \$t'\$ and the white-matter-based connectivity kernel \$W(\mathbf{r}, t; \mathbf{r}', t')\$; i.e.,

$$\phi(\mathbf{r}, t) = \int W(\mathbf{r}, t; \mathbf{r}', t') Q(\mathbf{r}', t') d^2\mathbf{r}' dt' . \quad (S14)$$

In the isotropic case, W depends only on the spatial separation between points and the time difference such that $W(\mathbf{r}, t; \mathbf{r}', t') := W(\mathbf{r} - \mathbf{r}', t - t')$. The kernel W is also known as the Green's function, which is the activity ϕ due to a point source. Using the Green's function method⁷⁰, previous work has shown that an appropriate expression for W , such that it becomes a solution of the damped wave equation in Eq. (S13) with an intense point source $Q(\mathbf{r}, t; \mathbf{r}', t') = \delta(\mathbf{r} - \mathbf{r}')\delta(t - t')$ and for $|\mathbf{r} - \mathbf{r}'| \leq \gamma_s r_s(t - t')$, is^{4,6,71,72}

$$W(\mathbf{r} - \mathbf{r}', t - t') = \frac{\gamma_s}{r_s} \frac{\exp\left(\frac{-|\mathbf{r} - \mathbf{r}'|}{r_s}\right)}{\sqrt{\gamma_s^2 r_s^2 (t - t')^2 - |\mathbf{r} - \mathbf{r}'|^2}} \Theta[\gamma_s r_s (t - t') - |\mathbf{r} - \mathbf{r}'|], \quad (\text{S15})$$

where $|\mathbf{r} - \mathbf{r}'|$ is the white-matter tract distance between points, Θ is the Heaviside step function, and γ_s and r_s are the damping rate and spatial length scale, respectively, as defined in Eq. (S13). See⁶ for a detailed derivation of Eq. (S15). Equation (S15) thus demonstrates how the wave dynamics defined by Eq. (S13) can be directly related to an underlying isotropic anatomical connectivity that decays exponentially with distance. In particular, solving the damped wave equation in differential form in Eq. (S13) is equivalent to solving the integral equation,

$$\phi(\mathbf{r}, t) = \frac{\gamma_s}{r_s} \int \frac{\exp\left(\frac{-|\mathbf{r} - \mathbf{r}'|}{r_s}\right)}{\sqrt{\gamma_s^2 r_s^2 (t - t')^2 - |\mathbf{r} - \mathbf{r}'|^2}} \Theta[\gamma_s r_s (t - t') - |\mathbf{r} - \mathbf{r}'|] Q(\mathbf{r}', t') d^2 \mathbf{r}' dt', \quad (\text{S16})$$

with the latter explicitly showing the spatiotemporal effect invoked by white-matter connectivity of a characteristic range of r_s (~ 84 mm⁶).

New Fig. S18:

Fig. S18. Schematic of signal propagation in neural field theory. Two points on the cortex at locations \mathbf{r}' and \mathbf{r} are connected by a white-matter tract (red curve). For an isotropic medium, the activity $\phi(\mathbf{r}, t)$ is a convolution of the source $Q(\mathbf{r}', t')$ and a connectivity kernel $W(\mathbf{r}, t; \mathbf{r}', t')$ imposed by the white-matter connection, which depends only on the spatial separation, $\mathbf{r} - \mathbf{r}'$, and the time separation, $t - t'$.

Comment 2.1: Geometrically-based eigenmodes explain activation patterns and functional connectivity better than connectomic-based eigenmodes.

The authors show that geometrically-based eigenmodes can better reproduce functional connectivity from resting state fMRI and unthresholded activation maps from a variety of tasks. Neural field theory and connectome-based harmonics (using the term loosely) are well-established, but this paper is the first attempt, to my knowledge, to directly compare the explanatory power of the two models. However, the differences are not large, especially when the connectome has a higher density.

The most plausible explanation is that both processes contribute to the organization of brain activity. Certainly it is known that long-distance white matter pathways exist to facilitate information transfer between physically distant areas, but at the same time, connections within grey matter also allow communication between adjacent areas. The combination of these processes may account for the lack of correspondence between eigenmodes and FCGs in the cortex, as long distance connections may be necessary to account for some of the structure that is observed. In support of this idea, many brain network models based on the structural connectome capture the sensory vs. association structure that lies at either end of the primary FCG.

Response 2.1: The Reviewer makes some important points and we take this opportunity to clarify any misunderstandings. First, we clarify that Fig. S9 shows that the difference in the reconstruction accuracies of geometric and connectome eigenmodes generally become *larger* as the connectome becomes denser, and when using a higher number of modes, in favor of the geometric eigenmodes.

Second, as discussed in Response 2.0, activity in NFT also propagates via white-matter connections. However, in the approximation used here, the spatial distribution of connections from each point on the cortex falls off approximately exponentially with distance, with parameters that are the same at every point on the cortex. Nonuniformities may perturb the results slightly, but our findings demonstrate that this simple assumption is an excellent and parsimonious approximation. Hence, our work does not discount the importance of long-distance white-matter connectivity in brain function, which has also been found to be dominated by an exponential drop-off based on dMRI data (Henderson and Robinson, 2011; Roberts et al., 2016). Our work suggests that an EDR-like connectivity can already explain several aspects of macroscopic activity measurable by fMRI. We nonetheless agree that our analysis raises important and open questions about the functional role of long-range connectivity beyond the EDR.

We have added the following text to the Discussion section to discuss these points:

Main Text Lines 457–465: Our findings thus counter traditional views that emphasize intricate patterns of anatomical connections as the primary driver of coordinated dynamics^{33,63,64}. Indeed, recent estimates indicate that long-range cortical connections are relatively rare⁶⁵; they may therefore represent a relatively minor perturbation of the dominant effect imposed by EDR-like connectivity. Nonetheless, the topological centrality, metabolic cost, and tight genetic control of such connections^{66–68} suggest that they provide important functional and evolutionary advantages beyond wave-like dynamics⁶⁹. The limited resolution and sensitivity to preprocessing pipelines^{70,71} of dMRI and fMRI data complicate attempts to uncover the role of long-range connections, but high-quality animal tract-tracing and electrophysiological data may be helpful in this regard.

Comment 2.2: *Traveling waves explain dynamics.*

The authors show that traveling waves can explain average and dynamic functional connectivity at a level similar to that obtained with connectome-based models. I found this section to be weaker than the rest of the paper, because no traveling waves were actually examined. As the authors state, recent work by Raut et al. and Bolt et al. describes the presence of traveling waves in fMRI data, and it is unclear whether traveling waves based on the geometric eigenmodes can recapitulate their features. The authors provide a toy example based on the timing of activity in different areas, but since the timing differences are in milliseconds, it is difficult to relate it directly to fMRI, which is the focus of the rest of the paper. The comparisons to average activity maps, average functional connectivity, and even dynamic functional connectivity collapse the relationships across time. I would be very interested to see if the eigenmode model can capture different

dynamic aspects of the data. Different brain network models based on the structural connectome are mostly successful in reproducing time-averaged metrics, but differ in their ability to recreate dynamic features 1,2

The reports of ‘waves’ of activity in the brain are not actually new—they have been seen for years and in multiple modalities^{3–6}, as have other time-lagged phenomena⁷. These features can provide a more rigorous test for models of the brain’s intrinsic organization than time-averaged features like functional connectivity.

Response 2.2: The Reviewer makes an important point. We originally evaluated model performance with respect to both static (edge and node FC) and dynamic (FCD) FC-based properties of resting-state fMRI data, as these have been widely used for evaluating neural mass models in past work (Demirtaş et al., 2019; Deco et al., 2021; Aquino et al., 2022) and are easily interpretable. Our results show that the wave model captures the statistical properties of static node-averaged FC and FCD better than the neural mass model, whereas performance for edge-level FC is comparable between the two.

To address the Reviewer’s concerns, we have now added a new analysis that examines the spatiotemporal properties of lag threads, as described by Mitra et al. (2014, 2015) and cited by the Reviewer, which capture temporal properties of propagated activity in the brain. In particular, we calculated the optimal pairwise time delays (lags) between the BOLD-fMRI time courses of brain regions using empirical data, simulated data of the wave model, and simulated data of the mass model, resulting in three time-delay matrices (new Fig. S19A presented below). From the time-delay matrices, we calculated three lag projections (Mitra et al., 2014, 2015): (i) mean lag of a brain region with respect to all brain regions; (ii) first principal component (PC) of the time-delay matrix; and (iii) second PC of the time-delay matrix. The first type of lag projection measures the average ordering of brain regions, as used in several studies (Mitra et al., 2014; Raut et al., 2021; Bolt et al., 2022). However, Mitra et al. (2015) highlighted that taking the mean lag may miss several fundamental lag patterns underlying the data not captured by the mean. Hence, they suggested using PCA to recover the complete time-delay structure in resting-state fMRI data; here, we used the first two dominant PCs, which account for 74% of the variance in the empirical data. Importantly, we did not reoptimize the parameters of the wave and mass models for the above analysis. Instead we used the previously determined optimal parameters (Supplementary Material-S12.5) to ensure generalizability.

The results in the new Figs. S19B to S19D presented below show that, irrespective of the type of lag projection used, the wave model can capture the time-lagged structure of the empirical data better than the mass model. This further supports the hypothesis that wave dynamics can provide a unifying account of the properties of resting-state fMRI data, especially given that these new results naturally emerged without reoptimization of the parameters of both models. Once again, we emphasize the parsimony of the wave model in accounting for these diverse functional properties relative to the neural mass model (1 free and 1 fixed parameters for wave model vs 4 free and 14 fixed parameters for mass model).

We have added new text to Supplementary Material-S12.6 to explain the method for measuring time-lagged properties of fMRI data. We have also added new text to the Results and Discussion sections to discuss the results from this new analysis.

Supplementary Material Lines 811–858: In addition to the FC-based metrics used for model fitting and evaluation of model performance (Section S12.4), we investigated whether the wave and mass models can also capture the temporal properties of propagated activity. In particular, we analyzed the lag structure (or lag threads) of resting-state BOLD-fMRI time courses, as proposed by ^{95,96}. We briefly discuss the algorithm for calculating the lag structure of both empirical and simulated fMRI data below and refer the readers to previous articles ^{95,96} for further details. Note that there are several other ways of characterizing the

spatiotemporal properties of resting-state activity^{97–99}, but the currently chosen method is sufficient for our purposes.

The algorithm starts by calculating the lagged cross-covariance function of the time series between brain regions. Assuming that BOLD-fMRI time series are aperiodic¹⁰⁰, the time lag (or delay) between regions where the cross-covariance function exhibits an extremum (typically between 0 to 2 s) was obtained⁹⁵. This resulted in an anti-symmetric time-delay matrix TD , with elements τ_{ij} corresponding to the time delay between regions i and j , with column i representing the lag map of the system with reference to region i , and $\tau_{ij} = -\tau_{ji}$. Therefore, $\tau_{ij} > 0$ means that region j lags behind region i . The underlying lag structure was quantified in two ways: (i) taking the mean time lag of each region (mean of each column of TD); and (ii) applying a PCA on TD_{z_i} , which is TD with each column being zero-meaned. The first method obtains the average temporal ordering of brain regions, assuming that a single lag process governs the brain, which has been used in several past studies^{95,99,101}. However, for systems like the brain with multiple lag processes, the mean time lag cannot capture all fundamental lag patterns, which can be recovered via PCA⁹⁶. Here, we used the first two dominant PCs, explaining 74% of the variance of the empirical data. Thus, we calculated three lag projections: (i) mean lag; (ii) first PC (PC1 lag); and (iii) second PC (PC2 lag).

The TD matrix was calculated on empirical and simulated resting-state BOLD-fMRI time courses parcellated using the HCP-MMP1 parcellation (Section S5). For the empirical data, a TD matrix was calculated for each of the 255 HCP individuals (Section S4.1 and S4.3), and then an average TD matrix was calculated across individuals. For the simulated data, we generated time series for 255 trials (to match the 255 HCP individuals of the empirical data) of the wave and mass models using their respective original optimized parameters (Section S12.5). A TD matrix was calculated for each trial. Then, an average TD matrix was calculated across trials. The average TD matrices are shown in Fig. S19A. Finally, the three lag projections were obtained from the empirical and simulated average TD matrices (Figs. S19B to S19D).

The results in Fig. S19B show that the mean lag pattern of the wave model is significantly correlated with the empirical pattern, whereas the mass model's mean lag pattern is not. For PC1 lag, the performance of the wave model slightly decreased but was still superior to the mass model (Fig. S19C). Correlations between model and empirical patterns for PC2 lag (Fig. S19D) were not significant, but the correlation was still higher for the wave model. These results show that, regardless of the method for calculating lag projections, the wave model captures the time-lagged properties of empirical fMRI data better than the mass model. This further highlights that wave dynamics can provide an accurate and physically mechanistic account of macroscale, resting-state dynamics, consistent with previous studies^{102–104}. We also emphasize that the performance of these models in capturing lag structure is likely to be a conservative estimate, as the models rely on a simple and spatially uniform hemodynamic forward model that does not account for regional variations in neurovascular coupling^{86,105,106}. Such variations are likely to strongly affect the empirically observed lags.

Main Text Lines 336–337: Additionally, the wave model can better capture time-lagged properties^{46,47,51} of empirical resting-state activity compared to the mass model (Supplementary Material-S12.6 and Fig. S19).

Main Text Lines 499–501: This finding is consistent with experimental observations of wave dynamics in both human and animal fMRI data^{79,80}.

New Fig. S19:

Fig. S19. Comparison of the wave and mass models in capturing time-lagged properties of fMRI data. (A) Time-delay matrices from empirical data, simulated data using the wave model, and simulated data using the mass model of the left hemisphere. Negative–zero–positive values are colored as blue–white–red. (B) Mean lags from the matrices in panel A (mean of each column) projected on the cortical surface. Negative–zero–positive values are colored as blue–white–red. The scatter plots show the relationship of mean lags from empirical data and simulated data from the two models. The red line represents a linear fit with Pearson correlation coefficient r and spin-test p -value p_{spin} from 10,000 permutations. (C) Similar to panel B but on the first principal component (PC1) of the matrices in panel A. The number above the surfaces (var) corresponds to the variance explained by the PC. (D) Similar to panel B but on the second PC (PC2) of the matrices in panel A.

Comment 2.3: *Eigenmodes explain FCGs in subcortical structures.*

One of the strengths of the study is that the authors examined areas outside of the cortex, and found that geometric constraints could explain the functional connectivity gradients previously reported for these areas. These results are interesting and might be considered in the context of prior work that reported that large, repeated spatiotemporal patterns (in many ways comparable to the traveling waves in this manuscript) sweep across these FCGs⁸. The existence of a common mechanism across multiple brain areas is an elegant aspect of the geometrically-based model.

Response 2.3: Thank you for your interest in this finding and for drawing attention to this work. We have added the following text to the Discussion section to relate our results to other works on wave dynamics in non-neocortical regions:

Main Text Lines 468–470: ..., suggesting that the functional organization of regions outside the neocortex is also dominated by local anatomical connectivity and wave dynamics, as found in recent experiments^{46,72,73}.

Comment 2.4: *One of the aspects of this work that requires clarification is whether global signal regression was performed for any of the studies. The global signal itself is a spatiotemporal pattern^{9,10} that is linked to neuromodulatory input and arousal^{11,12}. If global signal regression is not performed, then changes in vigilance or arousal at the onset of a task might be included in the resulting activation maps. The global signal is not necessarily the only large-scale signal related to neuromodulation or arousal, and so even if GSR was performed, the long wave aspect of activity captured in the unthresholded maps could reflect these types of changes. We know that certain areas of the brain respond strongly and specifically to particular stimuli (e.g., whisker barrel in rodents) so it may be that the classical approach of thresholding and the proposed examination of whole brain maps will turn out to provide complementary views of the brain. On the other hand, activation of nearly the entire brain has been observed during task performance¹³, and it may be that the unthresholded maps are capturing sub-threshold task-related activity rather than other ongoing processes.*

Response 2.4: We used preprocessed data of task-activation maps (computed via FSL's FEAT analysis) and resting-state time series (ICA-FIXed) directly provided by HCP, without performing any additional preprocessing steps such as global signal removal (GSR). Nonetheless, the first eigenmode is a constant, global mode (see Fig. 1A and Fig. S1 of the manuscript), which intrinsically captures global deviations in the data in a way that is approximately similar to GSR. The difference in our approach is that we include the global mode in all our analyses and do not regress it out, allowing us to capture properties of the fMRI completely. Hence, modes after the first one capture functionally relevant processes beyond global activity.

We have added new text to Supplementary Material-S4.1 to briefly mention the above points:

Supplementary Material Lines 153–156: We did not perform any additional preprocessing steps, such as global signal removal (GSR), because the first eigenmode (considered as the global, constant mode) already explicitly captures global deviations in the data, allowing the other modes to capture functionally relevant non-global activity.

Comment 2.5: *Overall, this was a well-conducted, carefully-designed study. The authors' careful examination of choice of parcellation and threshold, along with the use of different datasets, shows their attention to rigor. While I feel that the emphasis on the geometric eigenmode model's (slightly) better performance in predicting activity patterns and functional connectivity is a bit overstated, it is a welcome counterbalance to the innumerable studies that simply assume a connectome-based model during their analysis and interpretation.*

Response 2.5: We thank the Reviewer for their comments and appreciation of our work.

Comment 2.6: *A few minor comments for the authors' consideration:*

Given the dependence on geometry and the curvature of the cortex, I would expect the eigenmodel approach to rely heavily on individual geometry. It would be interesting to see if differences in individual eigenmodes explain differences in activation or functional networks across subjects.

Response 2.6: This is an interesting point and something we are currently investigating as part of future work. The results in Fig. S4 show that the performance of geometric eigenmodes derived from individual-specific surface is, on average, similar to that of geometric eigenmodes derived from a template surface. This suggests that the use of a template surface is sufficient for our current purposes. However, there are some minute differences in the results in Fig. S4 between template and individual-specific eigenmodes. This is better seen in the new Fig. S5, presented below, where we plot the reconstruction accuracy for the data of each individual using 200 template-derived and individual-specific eigenmodes. For most of the participants, individual-specific eigenmodes perform slightly better than template-derived eigenmodes, especially in explaining task-activation maps. This could be because task activations driven by structured stimuli are highly individualized, which could be better captured by individual-specific eigenmodes and cortical geometry. However, the new Fig. S6 presented below shows that the performance of template-derived and individual-specific eigenmodes converges at very high-frequency modes, suggesting that many of the nuances of individual-specific geometry are encoded within the first 200 modes. This result is consistent with our own recent study showing that the most reliable individual differences in cortical geometry peak at around mode 140 (Chen et al., 2022). Remarkably, 200 modes covers a very small fraction of the maximum possible 32,492 modes (number of surface vertices = maximum number of modes).

It would certainly be interesting to understand whether certain individual-specific eigenmodes drive these activation differences, but it is not straightforward to address this question. Individual differences in the geometry of individual surface meshes can cause a divergence in the ordering and spatial patterning of individual eigenmodes, especially at short spatial wavelengths (Supplementary Material-S2; Henderson et al., 2022; Chen et al., 2022), which makes it difficult to compare eigenmodes across individual participants. We are currently working on the best approach to solve this problem as a future extension of the current work.

We have added the following text to Results and Supplementary Material-S2 to discuss the above points:

Main Text Lines 160–163: However, there are some participants where individual-specific eigenmodes perform slightly better, especially in reconstructing task-activation maps (Fig. S5), but their accuracies eventually converge with template-derived eigenmodes at short wavelengths (Fig. S6).

Supplementary Material Lines 91–98: Nonetheless, to highlight certain nuances of individual geometry, we show in Fig. S5 that 200 individual-specific eigenmodes perform slightly better than template-derived eigenmodes in some individuals, especially in reconstructing task-activation maps but not in reconstructing resting-state activity. However, Fig. S6 shows that the results for individual-specific and template-derived eigenmodes converge at very short wavelengths (~500th mode). Thus, the effects of individual differences in cortical geometry on brain function are captured by the first 200 modes, consistent with recent work³⁹, which corresponds to a very small fraction of the maximum possible number of eigenmodes (~0.6%).

New Fig. S5:

Fig. S5. Individual-based reconstruction accuracy of 7 key HCP task-contrast maps and resting-state FC using 200 template-derived and individual-specific geometric eigenmodes. See Section S4.2 and Table S2 for details about the contrast maps. wm = working memory. Template eigenmodes were derived from a template surface, while individual-specific eigenmodes were derived from individual-specific surfaces. Each point corresponds to an individual. The dotted lines represent the template accuracy = individual-specific accuracy lines. Points above the dotted lines mean that template accuracy > individual-specific accuracy. The insets show the histogram of the difference between the reconstruction accuracy achieved by template eigenmodes and individual-specific eigenmodes (i.e., template minus individual-specific).

New Fig. S6:

Fig. S6. Reconstruction accuracy of 7 key HCP task-contrast maps and resting-state FC using 200 to 500 template-derived and individual-specific geometric eigenmodes. See Section S4.2 and Table S2 for details about the contrast maps. wm = working memory. The solid lines represent results achieved by template eigenmodes derived from a template surface (Fig. 1A). The dashed lines represent results achieved by

individual-specific eigenmodes derived from individual-specific surfaces. The insets show the difference between the two results (i.e., template minus individual-specific).

Comment 2.7: The motivation for the very narrow frequency band for FCD (0.04-0.07 Hz) was unclear. There is no F label in Figure 1.

Response 2.7: The choice of the band was based on the work of Deco et al. (2017) using the same FCD metric, which was motivated by past work showing its functional relevance (Glerean et al., 2012; Pang and Robinson, 2019).

We have fixed the missing label in Fig. 1. We have also added the following text to Supplementary Material-S12.4 to briefly clarify the choice for the FCD's frequency band.

Supplementary Material Lines 752–753: ...; this band was based on⁹⁰ and was motivated by its functional relevance to the brain^{91,92}.

Comment 2.8: References

1. Cabral J, Kringelbach ML, Deco G. Functional connectivity dynamically evolves on multiple time-scales over a static structural connectome: Models and mechanisms. *NeuroImage*. 2017;160:84-96. Doi:10.1016/j.neuroimage.2017.03.045
2. Kashyap A, Keilholz S. Dynamic properties of simulated brain network models and empirical resting-state data. *Netw Neurosci*. 2019;3(2):405-426. Doi:10.1162/netn_a_00070
3. Majeed W, Magnuson M, Hasenkamp W, et al. Spatiotemporal dynamics of low frequency BOLD fluctuations in rats and humans. *NeuroImage*. 2011;54(2):1140-1150. Doi:10.1016/j.neuroimage.2010.08.030
4. Matsui T, Murakami T, Ohki K. Transient neuronal coactivations embedded in globally propagating waves underlie resting-state functional connectivity. *Proc Natl Acad Sci U S A*. 2016;113(23):1521299113-. Doi:10.1073/pnas.1521299113
5. Chan AW, Mohajerani MH, LeDue JM, Wang YT, Murphy TH. Mesoscale infraslow spontaneous membrane potential fluctuations recapitulate high-frequency activity cortical motifs. *Nat Commun*. 2015;6:7738. Doi:10.1038/ncomms8738
6. Tong Y, Hocke LM, Fan X, Janes AC, Frederick B deB. Can apparent resting state connectivity arise from systemic fluctuations? *Front Hum Neurosci*. 2015;9. Doi:10.3389/fnhum.2015.00285
7. Mitra A, Snyder AZ, Blazey T, Raichle ME. Lag threads organize the brain's intrinsic activity. *Proc Natl Acad Sci U S A*. Published online March 30, 2015:1503960112-. Doi:10.1073/pnas.1503960112
8. Yousefi B, Keilholz S. Propagating patterns of intrinsic activity along macroscale gradients coordinate functional connections across the whole brain. *NeuroImage*. 2021;231:117827. Doi:10.1016/j.neuroimage.2021.117827
9. Bolt T, Nomi JS, Bzdok D, et al. A parsimonious description of global functional brain organization in three spatiotemporal patterns. *Nat Neurosci*. 2022;25(8):1093-1103. Doi:10.1038/s41593-022-01118-1
10. Scholvinck ML, Maier A, Ye FQ, et al. Neural basis of global resting-state fMRI activity. *Proc Natl Acad Sci U S A*. 2010;107(22):10238-10243. Doi:10.1073/pnas.0913110107
11. Turchi J, Chang C, Ye FQ, et al. The Basal Forebrain Regulates Global Resting-State fMRI Fluctuations. *Neuron*. 2018;97(4):940-952.e4. doi:10.1016/j.neuron.2018.01.032
12. Wong CW, Olafsson V, Tal O, Liu TT. The amplitude of the resting-state fMRI global signal is related to EEG vigilance measures. *NeuroImage*. 2013;83:983-990. Doi:10.1016/j.neuroimage.2013.07.057
13. Gonzalez-Castillo J, Saad ZS, Handwerker DA, Inati SJ, Brenowitz N, Bandettini PA. Whole-brain, time-

locked activation with simple tasks revealed using massive averaging and model-free analysis. Proc Natl Acad Sci U A. 2012;109(14):5487-5492. Doi:10.1073/pnas.1121049109

Response 2.8: We thank the Reviewer for these references and have added most of them in the revised manuscript.

Response to Reviewer 3

Comment 3.1: Summary of the key results

This is a truly fascinating paper, bringing ideas highly influential in the natural sciences (mainly physics) to the domain of systems neuroscience. The authors adopted a neural field perspective on large scale brain dynamics, with the geometry of the cortical sheet imposing boundary conditions that determine a basis set of normal modes. From these normal modes, different aspects of brain activity (e.g. spontaneous, task-evoked) can be derived, crucially with better accuracy than when adopting a similar decomposition based on large-scale connectivity. Perhaps the most surprising key result is that a neural field assuming local connectivity provides a better description of several brain phenomena than a similar description based on the structural connectome.

The specific key results of this manuscript are the following:

- *A discrete set of normal modes can be obtained when adopting brain geometry as a boundary condition of a suitable partial differential equation*
- *This description supersedes a similar one based on the (long range) connectome*
- *Brain activations correspond to low frequency modes, which implies a global perturbation as opposed to a very localized perturbation*
- *Traveling waves can be used to describe some of the aforementioned phenomena*
- *Structures outside the cortex are functionally characterized by the geometric normal modes*

Response 3.1: We thank the Reviewer for their careful evaluation and appreciation of our work.

Comment 3.2: Originality and significance

This approach is certainly novel. The closest work is that of connectome eigenmodes, but besides some mathematical superficial similarities (e.g. the use of eigenmode decomposition), the interpretation of the findings and their scope is very different.

Response 3.2: We would also like to add to the Reviewer's comment that the geometric eigenmode approach provides a more compact representation of activity. Moreover, it links the present work to the wide body of verified NFT predictions for EEG, evoked-response, and other experiments over the last decades (see for example Robinson et al. (1997, 2001, 2005, 2016)).

Comment 3.3a: Data & methodology

I have some concerns and commentaries about the methods, and also about the interpretation of the results. I can summarize these in the following key points:

- *The exponential decay rule is not valid at the large scale. Locally, both DTI and fMRI FC show an exponential decay with distance from a given seed. However, we also know that this decay is bound to stop after a certain distance, and that connectivity must then start to increase. This is due to the strong homotopic (interhemispheric) connectivity of the brain: even if a region is weakly connected to its neighboring regions within a certain hemisphere, it is most likely strongly connected with its counterpart at the other brain*

hemisphere. To which extent does this represent a limitation for the proposed methodology, and how can this potential limitation be overcome?

Response 3.3a: The Reviewer raises an important point. We clarify that both intrahemispheric and interhemispheric connectivity strengths have been empirically found to follow the exponential-distance rule (EDR). This has been demonstrated for connectomes mapped at the regional (i.e., macro) scale with diffusion MRI (see for example Fig. 1C of Rosen and Halgren (2022) or Fig. 1D of Roberts et al. (2016)) and tract-tracing in rodents and non-human primates (Ercsey-Ravasz et al., 2013; Song et al., 2014; Horvát et al., 2016; Gămănuț et al., 2018). The exponential decay rule is thus applicable across multiple spatial scales in the brain, to a first approximation.

However, we agree that some connections, particularly homotopic inter-hemispheric connections, do not neatly conform to the dominant pattern established by the EDR. Although, such connections are numerically small in number ($\sim N$ homotopic compared to $\sim N^2$ heterotopic) and thus only slightly perturb the EDR curve (Roberts et al., 2016; Rosen and Halgren, 2022). In addition, Robinson et al. (2016) have shown that inter-hemispheric connections have a negligible impact on the spatial structure of geometric eigenmodes. Specifically, they showed that bihemispheric eigenmodes derived from the whole brain will simply be combinations of the eigenmodes derived from individual hemispheres, with the combinations either symmetric (i.e., mirror-symmetry across the sagittal midplane) or asymmetric (i.e., hemispheres having the same spatial structure but with flipped signs).

We have added the following text to Supplementary Material-S2 to briefly discuss bihemispheric eigenmodes:

Supplementary Material Lines 67–71: All our analyses were focused on unihemispheric eigenmodes, but our approach can easily be extended to the whole brain because bihemispheric eigenmodes can be represented as symmetric or antisymmetric combinations of the eigenmodes derived from each hemisphere²³; symmetric corresponds to mirror-symmetry across the sagittal midplane and asymmetric corresponds to hemispheres having the same spatial structure but with flipped signs.

Comment 3.3b: *-All reasonable functions can be written as a series of the eigenmodes with appropriate coefficients. This is known since the times of Fourier, when he obtained such decomposition in terms of sines and/or cosines. More complicated geometries will change the precise mathematical form of the eigenmodes, but the result will still be valid. In this sense, the results could be deemed trivial by some (personally I think that even if a result is mathematically trivial, it could be highly relevant within a certain neurological context, as in this particular case). However, I feel that because of this, the comparison between the functional and connectome eigenmodes is not entirely fair. Any function with the cortex as its domain can be written as a suitable expansion of the corresponding functional eigenmodes (by construction), but the connectome eigenmodes do not span the same space (they were obtained in a different problem with different boundary conditions) and hence it is not surprising (at least in my opinion) that they are less accurate in this context compared to the functional eigenmodes.*

Response 3.3b: Thank you for raising this point. We first wish to clarify terms. We avoid the use of ‘functional eigenmodes’ in the present context and prefer the term ‘geometric eigenmodes’ to emphasize that the eigenmodes were derived from geometry, and not from the functional data itself, as done in some recent studies (Glomb et al., 2021).

With regards to the comment on Fourier decomposition, we have now added an additional analysis to compare the performance of the geometric eigenmodes with simple Fourier basis sets in decomposing fMRI activity data. In particular, we constructed six different spatial Fourier basis sets using combinations of sines and/or cosines in 3D Euclidean space (see new Supplementary Material-S9 or the revisions in the text shown below for details). The Fourier basis sets are shown in the new Fig. S14A, presented below. We then used them to reconstruct task-activation maps and resting-state data and compared their reconstruction accuracies with the geometric eigenmodes (new Fig. S14B). The new Fig. S14B shows that geometric eigenmodes significantly outperform the Fourier basis sets, providing evidence that the accurate representation provided by geometric eigenmodes is not a trivial result of any basis set expansion.

We also emphasize that theoretically, as evident in the above new results, Fourier decompositions are poorly suited to non-regularly shaped objects such as the cortical surface. Moreover, Fourier basis sets do not satisfy the intrinsic boundary conditions of the cortical surface (Do Carmo, 1976). For non-regularly shaped objects, our approach using the Laplace-Beltrami operator (LBO) for deriving the eigenmodes is more appropriate and is considered to be a generalization of the Fourier decomposition to non-regular (Riemannian) manifolds such as the cortex (Klingenberg, 1995; Reuter, 2006). We also clarify that the goal of the work is not to find an optimal basis set for decomposing brain maps; instead, we aim to compare the two most physiologically plausible and anatomically grounded basis sets proposed in the literature thus far—geometric and connectome eigenmodes—to understand the most fundamental structural constraints on brain dynamics.

We have added the following text to Supplementary Material-S9 and Results and Discussion sections to discuss the above new analysis and considerations:

Supplementary Material Lines 370–437: To further confirm that the performance of geometric eigenmodes is not trivially driven by how the mathematics of any basis set expansion, we compared the geometric eigenmodes to six simple spatial Fourier basis sets based on combinations of sines and/or cosines. The Fourier basis sets were constructed using the following real-valued functions:

$$F_1 := \cos\left(\frac{2\pi(j_x - 1)x}{L_x} + \frac{2\pi(j_y - 1)y}{L_y} + \frac{2\pi(j_z - 1)z}{L_z}\right), \quad (S8)$$

$$F_2 := C_1 \cos\left(\frac{2\pi(j_x - 1)x}{L_x}\right) + C_2 \cos\left(\frac{2\pi(j_y - 1)y}{L_y}\right) + C_3 \cos\left(\frac{2\pi(j_z - 1)z}{L_z}\right), \quad (S9)$$

$$F_3 := C_1 \cos\left(\frac{2\pi(j_x - 1)x}{L_x} + \frac{2\pi(j_y - 1)y}{L_y} + \frac{2\pi(j_z - 1)z}{L_z}\right) + C_2 \sin\left(\frac{2\pi(j_x - 1)x}{L_x} + \frac{2\pi(j_y - 1)y}{L_y} + \frac{2\pi(j_z - 1)z}{L_z}\right), \quad (S10)$$

where j_x, j_y, j_z are integer constants, x, y, z are the spatial positions of each point on the spherical surface mesh representation of the cortex, L_x, L_y, L_z are the periods in each direction, and C_1, C_2, C_3 are fitting constants. For each function, we constructed a regular and an irregular version, resulting in six basis sets. The regular version corresponds to when $j_x = j_y = j_z = j$ such that the spatial wavelengths of mode j in the x -, y -, and z -directions are regularly spaced and increase by $\frac{2\pi}{L_x}, \frac{2\pi}{L_y}, \frac{2\pi}{L_z}$ as the mode number increases, which is a standard implementation in Fourier analysis. The irregular version corresponds to when j_x, j_y, j_z are integer combinations and not necessarily equal; hence, the spatial wavelengths in the x -, y -, and z -directions are irregularly spaced. We implemented this version because it affords the Fourier basis sets greater freedom and thus the best possible chance of performing well. To associate a (j_x, j_y, j_z) combination to a single mode

j , we arranged the combinations in order of increasing $j_x + j_y + j_z$. For example, the (j_x, j_y, j_z) combination for the first ten modes follows the set $\{(1,1,1), (1,1,2), (1,2,1), (2,1,1), (1,2,2), (2,1,2), (2,2,1), (1,1,3), (1,3,1), (3,1,1)\}$. These two versions allow us to explore how the spatial wavelengths can affect the decompositions of the Fourier basis sets, given that the spatial wavelengths of the geometric eigenmodes are irregularly spaced (see Table S1).

Next, we defined the periods to be $L_i := [\max(i) - \min(i)]$ for $i = x, y, z$, to have a heuristic that respects the shape of the brain. In addition, the -1 in $(j_i - 1)$ ensures that the first mode is constant to make it comparable to the first geometric mode. The forms in Eqs. (S9) and (S10) have more degrees of freedom, with the fitting constants, C_1, C_2, C_3 , estimated separately, allowing us to weight the cosine and/or sine functions differently. Therefore, mode j now has multiple amplitudes to be estimated during the decomposition in Eq. (S4) instead of just one. These added degrees of freedom result in an increased model complexity relative to geometric eigenmodes; more specifically, the Fourier basis sets formed by Eqs. (S9) and (S10) involve estimating 2 and 3 coefficients per mode, respectively, whereas geometric eigenmodes require fitting just one coefficient per mode.

Although other forms and complicated choices can be made to construct a Fourier basis set, the above choices are well motivated by their simplicity, yielding unique, real-valued spatial modes, and cover the key implementation choices one could make. Figure S14A shows the spatial profiles of the resulting modes with unit coefficients, which we used to reconstruct task-activation maps and resting-state data (similar to Fig. 1D). Figure S14B shows that geometric eigenmodes significantly outperform the Fourier basis sets in reconstructing both task-evoked and resting-state data, further emphasizing that the accurate representation provided by geometric eigenmodes is not a trivial result of any basis set expansion. Moreover, this conclusion remains regardless of how the spatial wavelengths of the modes in the x-, y-, and z-directions are defined. To reiterate, even though the Fourier basis sets based on Eqs. (S9) and (S10) have more degrees of freedom to fit the data when compared to geometric eigenmodes, geometric eigenmodes still show superior performance, underscoring their parsimony in accounting for brain dynamics.

We present the above analysis of Fourier basis sets for completeness and to demonstrate that such functions cannot accommodate the boundary conditions of non-regularly shaped objects, such as the cortical surface (e.g., Dirichlet or Neumann conditions for the medial wall)⁶¹. This is because Fourier basis sets can only be properly constructed for objects with regular shapes (e.g., rectangle) and are more suitable to be used for analyzing functions defined on such shapes (e.g., 2D images). Hence, for the cortical surface defined on a Riemannian manifold, using a Fourier basis set is generally not well motivated and using the eigenmodes of the Laplace-Beltrami operator is more appropriate^{62,63}. In fact, mathematically speaking, LBO eigenmode analysis on the Riemannian manifold is considered a generalization of Fourier analysis⁶⁴. The same applies to discrete networks, such as a connectome; for this reason, the eigendecomposition of the graph Laplacian (as done in the work) is commonly used in the field of spectral graph theory as an alternative to a classical Fourier transform⁶⁵. Thus, whilst it is possible to decompose spatial maps of brain activity using Fourier basis sets, they are highly inefficient and poorly suited to the current problem. We therefore do not advocate their use. We also note that Fourier basis sets offer no insights into the generative processes underlying brain activity. Our primary focus is to compare physiologically principled and anatomically constrained basis sets (i.e., geometric and connectome eigenmodes) to uncover the critical constraints on brain dynamics, and not to identify statistically optimal basis sets.

Main Text Lines 239–242: We additionally find that geometric eigenmodes show stronger out-of-sample generalization than principal components of the functional data itself (calculated via principal component

analysis (PCA); Supplementary Material-S8 and Figs. S12 and S13) and better performance than Fourier spatial basis sets (Supplementary Material-S9 and Fig. S14).

Main Text Lines 445–449: The extensive comparisons of geometric eigenmodes with other anatomical (connectome and EDR eigenmodes) and mathematical (PCA and Fourier) basis sets show that their superior performance in capturing macroscale neocortical activity is not trivially driven by generic mathematical properties of basis set expansions. Rather, this result indicates that geometry represents a fundamental anatomical constraint on dynamics.

New Fig. S14:

Fig. S14. Comparison of geometric eigenmodes and Fourier basis sets. (A) Spatial maps of modes 1, 2, 3, 4, 10, 50, and 100 of six different Fourier basis sets with unit coefficients. The terms reg and irreg mean that the spatial wavelengths of the modes in the x-, y-, and z-directions are spaced in regular and irregular increments, respectively. See Supplementary Material-S9 for details. (B) Reconstruction accuracy of 7 key HCP task-contrast maps and resting-state FC. See Section S4.2 and Table S2 for details about the contrast maps. wm = working memory.

Finally, in relation to the comment regarding the space spanned by the modes, we clarify that the connectome eigenmodes were derived using a high-resolution, vertex-wise connectome (see Supplementary Material-S6) with 32,492 vertices per hemisphere. Hence, the resulting modes span a space with 32,492 dimensions (number of available modes), which is exactly the same as geometric eigenmodes. The difference

between the two is in the way in which connectivity between vertices is defined, as outlined in Supplementary Material-S2 and S3. However, it is the number of vertices, and not their connectivity, that defines the dimensionality of the eigenmodes. Since the number of vertices is equivalent between the two approaches, geometric and connectome eigenmodes are compared on equal footing. For highly discretized connectome eigenmodes (parcel level), the dimensions are different and much smaller; hence, we rescaled the x-axis of Fig. S11 to show the % of available modes covered by 1 to 200 modes, enabling proper comparisons with the geometric eigenmodes. Together, these results emphasize the outstanding accuracy and parsimony that geometric eigenmodes provide in describing macroscale activity.

We have added the following text to Supplementary Material-S6 to clarify this point:

Supplementary Material Lines 254–256: Note also that the use of a high-resolution, vertex-level connectome results in connectome eigenmodes spanning a space with dimensions (number of modes) equal to the number of vertices, allowing fair comparison with geometric eigenmodes.

Comment 3.3c: - *I am also wondering about the utility of this approach beyond determining the locality of certain patterns of spontaneous or evoked brain activity. Different tasks and/or brain states result in clearly different patterns of brain activation without changes to brain geometry, therefore the same set of basis functions should be useful to describe them. Based on my understanding of this article, the relevant information is encoded in the coefficients of this expansion, which in turn signal the relative "energy" (wavelengths) needed for this expansion to be accurate. This translates to considerations about locality of the patterns (i.e. is only the first coefficients are needed, this implies low spatial frequencies). While certainly interesting, I also find that this information could be obtained without resorting to such complicated methods, simply focusing on the spatial smoothness of the patterns. Regarding this point, the fMRI volumetric data can be subject to a standard 3D Fourier decomposition, and I wonder whether the proposed method gives information that is not available from this representation.*

Response 3.3c: The Reviewer is correct that different patterns of brain activations can be viewed as excitations of the different modes, with some dominating others, as measured by the amplitude of the expansion coefficients. Our results also show that different tasks are associated with different profiles of coefficient magnitudes across modes (i.e., different modal power spectra), which aligns with the Reviewer's intuition (see Figs. 3 and S16). Whilst the eigenmode reconstruction approach has parallels to how smoothing works (e.g., keeping the low-order modes will remove the high spatial-frequency content of the data, including noise), its utility and our work more generally is not limited to a simple smoothing kernel. Geometric eigenmodes have a deeper meaning, with strong links to the generative process governing brain dynamics, as demonstrated by a large body of theoretical and empirical work through the lens of neural field theory (NFT) that has been conducted over the past few decades (see for example Robinson et al. (1997, 2001, 2005, 2016)). This is clearly demonstrated in Fig. 4 of the manuscript, where we show that the link between geometry and function is attributable to a dominant role for traveling waves in driving brain function, as predicted by NFT. This is certainly impossible to achieve by, for example, using a Gaussian function or low-pass Fourier filter to smooth the data. In addition, filtering imposes a single scale representation of the data determined by the kernel size, whereas the geometric eigenmodes provide a physically principled multiscale representation. Hence, the eigenmode approach has both practical and biologically meaningful advantages.

We have added the following text to Supplementary Material-S10 to emphasize this point:

Supplementary Material Lines 499–504: Moreover, whilst spatially extended patterns of task activations can also potentially be recovered using simple smoothing functions and avoiding statistical thresholding, the geometric eigenmode approach provides much deeper understanding of the mechanisms underlying the activations because the modes can be directly linked to rigorous biophysical models of brain dynamics as established by NFT (see for example ^{6,7,14,23}). They also offer a natural multiscale characterization of the data.

Finally, we note that a 3D Fourier decomposition of an entire MRI image volume is not appropriate in the present context as the results will be sensitive to signal variations in white matter, grey matter, cerebrospinal fluid, non-brain tissue, the space outside the brain, and the boundaries between these different objects. We are only interested in understanding spatial patterns defined on the grey matter, so an appropriate decomposition of spatial patterns of activity expressed on the cortical manifold is required. As outlined in Response 3.3b, the LBO eigenmode analysis is the appropriate choice, which is considered to be a generalization of the Fourier decomposition to this domain.

Comment 3.4: *Appropriate use of statistics and treatment of uncertainties*

I do not have any concerns regarding the use of statistics in this manuscript.

Response 3.4: No comment required.

Comment 3.5: *Conclusions: robustness, validity, reliability*

The conclusions are generally valid and robust, except when they depend on some of the concerns I raised above. For instance, the authors dismiss the utility of long-range connectomics to describe whole-brain activity patterns in contrast to their own approach because of the comparison between functional and connectome eigenmodes. Yet as I said, I do not think this is an entirely fair comparison, since the functional eigenmodes are guaranteed to allow the accurate reconstruction of any function on the cortical mesh, by construction, while this is not true of the connectome decomposition.

Response 3.5: We hope that Response 3.3b has clarified any misunderstandings. To reiterate, both geometric and connectome eigenmodes were derived using a vertex-level representation, spanning 32,492 dimensions (number of vertices on the cortical surface). They thus have equivalent dimensionality and span the same space, allowing fair comparison.

We have added the following text to Supplementary Material-S6 to clarify this point:

Supplementary Material Lines 254–256: Note also that the use of a high-resolution, vertex-level connectome results in connectome eigenmodes spanning a space with dimensions (number of modes) equal to the number of vertices, allowing fair comparison with geometric eigenmodes.

We also emphasize that the superiority of geometric eigenmodes should not be taken as a dismissal of the utility of long-range connections. In neural field theory (NFT), activity of points on the cortex propagates along the white-matter tracts with a connectivity that falls off approximately exponentially with distance as is observed experimentally (Nunez, 1974) (see Introduction and Supplementary Material-S1 and the new Fig.

S18 presented below for a schematic diagram). This point may have been obscured because our original manuscript wrote the NFT's wave equation in differential form (Eq. (3) of the main text) but the distance-dependence is clearly seen when the equation is expressed in its equivalent integral form (Eq. (S16) of the revised Supplementary Material) (Nunez, 1974; Jirsa and Haken, 1996; Robinson et al., 1997). Thus, the geometric eigenmodes do account for long-range connectivity; they just assume that this connectivity can be approximated using a simple, isotropic, EDR-like rule. Our findings indicate that this approximation, when combined with a model of cortical geometry, is better for understanding function when compared with connectome eigenmodes, which capture the full array of topologically complex short- and long-range connections but ignore the physical embedding and geometry of the brain.

We have added the following text to Results and Supplementary Material-S12.1 to clarify the above points:

Main Text Lines 320–322: Under this model, activity propagates between points on the neocortex through their white-matter connectivity, with a strength that decays approximately exponentially with distance (Supplementary Material-S1 and S12.1).

Supplementary Material Lines 534–563: Note that the propagation of activity between points is governed by their white-matter connectivity, with strength that decays approximately exponentially with distance. This distance-dependence is more apparent when Eq. (S13) is converted into its equivalent integral form as follows. Consider two points on the neocortex at locations \mathbf{r}' and \mathbf{r} connected by a white-matter tract (see visual schematic in Fig. S18). The activity $\phi(\mathbf{r}, t)$ at location \mathbf{r} and time t can be considered as a spatiotemporal convolution of the source $Q(\mathbf{r}', t')$ at location \mathbf{r}' and time t' and the white-matter-based connectivity kernel $W(\mathbf{r}, t; \mathbf{r}', t')$; i.e.,

$$\phi(\mathbf{r}, t) = \int W(\mathbf{r}, t; \mathbf{r}', t') Q(\mathbf{r}', t') d^2 \mathbf{r}' dt'. \quad (\text{S14})$$

In the isotropic case, W depends only on the spatial separation between points and the time difference such that $W(\mathbf{r}, t; \mathbf{r}', t') := W(\mathbf{r} - \mathbf{r}', t - t')$. The kernel W is also known as the Green's function, which is the activity ϕ due to a point source. Using the Green's function method⁷⁰, previous work has shown that an appropriate expression for W , such that it becomes a solution of the damped wave equation in Eq. (S13) with an intense point source $Q(\mathbf{r}, t; \mathbf{r}', t') = \delta(\mathbf{r} - \mathbf{r}')\delta(t - t')$ and for $|\mathbf{r} - \mathbf{r}'| \leq \gamma_s r_s (t - t')$, is^{4,6,71,72}

$$W(\mathbf{r} - \mathbf{r}', t - t') = \frac{\gamma_s}{r_s} \frac{\exp\left(\frac{-|\mathbf{r} - \mathbf{r}'|}{r_s}\right)}{\sqrt{\gamma_s^2 r_s^2 (t - t')^2 - |\mathbf{r} - \mathbf{r}'|^2}} \Theta[\gamma_s r_s (t - t') - |\mathbf{r} - \mathbf{r}'|], \quad (\text{S15})$$

where $|\mathbf{r} - \mathbf{r}'|$ is the white-matter tract distance between points, Θ is the Heaviside step function, and γ_s and r_s are the damping rate and spatial length scale, respectively, as defined in Eq. (S13). See⁶ for a detailed derivation of Eq. (S15). Equation (S15) thus demonstrates how the wave dynamics defined by Eq. (S13) can be directly related to an underlying isotropic anatomical connectivity that decays exponentially with distance. In particular, solving the damped wave equation in differential form in Eq. (S13) is equivalent to solving the integral equation,

$$\phi(\mathbf{r}, t) = \frac{\gamma_s}{r_s} \int \frac{\exp\left(\frac{-|\mathbf{r} - \mathbf{r}'|}{r_s}\right)}{\sqrt{\gamma_s^2 r_s^2 (t - t')^2 - |\mathbf{r} - \mathbf{r}'|^2}} \Theta[\gamma_s r_s (t - t') - |\mathbf{r} - \mathbf{r}'|] Q(\mathbf{r}', t') d^2 \mathbf{r}' dt', \quad (\text{S16})$$

with the latter explicitly showing the spatiotemporal effect invoked by white-matter connectivity of a characteristic range of r_s ($\sim 84 \text{ mm}^6$).

New Fig. S18:

Fig. S18. Schematic of signal propagation in neural field theory. Two points on the cortex at locations \mathbf{r}' and \mathbf{r} are connected by a white-matter tract (red curve). For an isotropic medium, the activity $\phi(\mathbf{r}, t)$ is a convolution of the source $Q(\mathbf{r}', t')$ and a connectivity kernel $W(\mathbf{r}, t; \mathbf{r}', t')$ imposed by the white-matter connection, which depends only on the spatial separation, $\mathbf{r} - \mathbf{r}'$, and the time separation, $t - t'$.

Comment 3.6a: *Suggested improvements*

I suggest the following new analysis/changes to improve the manuscript:

- Explore the consequences of including long range homotopic connections in the model, or provide an argument (most likely based on considerations about symmetry?) showing that this is not a concern for this approach.

Response 3.6a: As discussed in Response 3.3a, we reiterate that Robinson et al. (2016) showed that inter-hemispheric connections have a negligible impact on the geometric eigenmodes such that bihemispheric eigenmodes of the whole brain can be represented as symmetric or asymmetric combinations of the eigenmodes derived from individual hemispheres. Hence, this does not affect the current work's approach.

We have added the following text to Supplementary Material-S2 to briefly discuss this point:

Supplementary Material Lines 67–71: All our analyses were focused on unihemispheric eigenmodes, but our approach can easily be extended to the whole brain because bihemispheric eigenmodes can be represented as symmetric or antisymmetric combinations of the eigenmodes derived from each hemisphere²³; symmetric corresponds to mirror-symmetry across the sagittal midplane and asymmetric corresponds to hemispheres having the same spatial structure but with flipped signs.

Comment 3.6b: - Check whether the proposed method is capable of providing more information than a simple 3D Fourier transform of the, e.g., activation patterns, without any consideration to boundary conditions and/or the geometry of the cortical sheet.

Response 3.6b: As discussed in Response 3.3c, if one simply treated the cortex as a subset of a 3D cube and Fourier transformed in that larger volume, the sharp edges imposed by the cortical surface and other tissue types would lead to a representation dominated by high-order modes. An analog would be to try to represent the modes of a circular drumhead by rectangular Fourier coefficients—an expansion that is mathematically possible but which would be highly inefficient and unsuited to the problem.

Moreover, as discussed in Response 3.3b, we have shown in our new analysis in Fig. S14 (reproduced below for convenience) that if one constructs simple Fourier basis sets to describe task-evoked and resting-state fMRI data, they will perform poorly in comparison with geometric eigenmodes, proving that they are not suitable for our purposes. Please see Response 3.3b and the associated revisions to the text presented there for details. Further details are also presented in Response 1.1.

New Fig. S14:

Fig. S14. Comparison of geometric eigenmodes and Fourier basis sets. (A) Spatial maps of modes 1, 2, 3, 4, 10, 50, and 100 of six different Fourier basis sets with unit coefficients. The terms reg and irreg mean that the spatial wavelengths of the modes in the x-, y-, and z-directions are spaced in regular and irregular increments, respectively. See Supplementary Material-S9 for details. (B) Reconstruction accuracy of 7 key HCP task-contrast maps and resting-state FC. See Section S4.2 and Table S2 for details about the contrast maps. wm = working memory.

Comment 3.6c: - Tone down the comparison between functional and connectome eigenmodes, or provide a counter-argument to show that my concern of a biased comparison is not valid.

Response 3.6c: As per Responses 3.3b and 3.5, we reiterate that geometric and connectome eigenmodes span the same space because they are derived from the same high-resolution mesh of the cortical surface

(i.e., they span the same space with number of dimensions equal to the number of vertices). Hence, their comparison is fair.

We have added the following text to Supplementary Material-S6 to clarify this point:

Supplementary Material Lines 254–256: Note also that the use of a high-resolution, vertex-level connectome results in connectome eigenmodes spanning a space with dimensions (number of modes) equal to the number of vertices, allowing fair comparison with geometric eigenmodes.

Comment 3.7: *References*

No comments in this regard.

Response 3.7: No comment required.

Comment 3.8: *Clarity and context: lucidity of abstract/summary, appropriateness of abstract, introduction and conclusions*

The manuscript is clearly written and will be of interest to an ample and general audience. Figures are beautiful and informative.

Response 3.8: We thank the Reviewer for their positive appraisal of our work.

Comment 3.9: *Final comment*

I think this is a thought-provoking and original article, and that it is suitable for publication in Nature. My concerns and comments should not be interpreted as criticism to prevent the publication of this manuscript, but as a natural part of a reviewing process aimed to improve the published work, the robustness of its results, and the scope of its conclusions. I will happily endorse the publication of a revised version of this manuscript after these concerns are adequately addressed.

Response 3.9: We hope that the discussions in the above Responses, the new analysis in Fig. S14, the new schematic diagram in Fig. S18, and the revisions made to the manuscript sufficiently address the concerns of the Reviewer.

Response to Reviewer 4

Comment 4.1: *This is a very well-written manuscript focusing on the study of brain physiological organization principles, in particular investigating the role of pure geometry -as opposed to neural bundle architecture- in shaping dynamics of brain activity. To do so, brain data from Magnetic Resonance Imaging (MRI) taken from existent online databases of healthy subjects are employed. The key findings reported by the authors are twofold: 1) Brain activity could be equally well or better explained by modes based on brain geometry, than by complex white matter architecture of neural connections; 2) Mechanisms driving brain activity propagation follow simple wave-like dynamics, rather than complex neural mass models that were employed so far.*

The exploration proposed is novel and the topic approached -mechanisms of brain organization- is in principle of high relevance for a broad audience.

The overall manuscript quality appears excellent: clearly written and very well presented (high-quality graphical elements efficiently summarizing results and analyses); the analyses performed are sound; statistical analysis is appropriate; previous work was correctly referenced.

The study encompasses a very large body of analyses, which are per se very interesting for the field, to show the relevance of geometric constrains in brain functioning. However, I have some concerns about the very strong claims the authors report in favor of a geometry-centric view that completely discard the role of white matter architecture in brain neural dynamics. I think therefore that these claims should be reduced, in light of the following remarks. I also think this would not affect the validity, nor the interest of the experiments performed.

Response 4.1: We thank the Reviewer for this positive appraisal of our work. We take this opportunity to clarify that the use of geometric eigenmodes within the context of neural field theory (NFT) does indeed account for white-matter connectivity, such that the activity of points on the cortex propagates along the white-matter tracts (see the new Fig. S18 presented below for a schematic diagram). In this sense, it does not discard the role of white-matter architecture; it simply makes an approximation that the connectivity between different points on the cortical sheet is isotropic and decays roughly exponentially with distance. Empirical data support this approximation (Robinson et al., 1997; Braitenberg and Schüz, 2018; Wang et al., 2016; Roberts et al., 2017). The isotropic assumption allows for the brain's geometric properties (e.g., boundary conditions and convolutions) to determine the spatial nature of the geometric eigenmodes. Hence, we do not discount the importance of long-distance connectivity in brain function, but our work suggests that an EDR-like connectivity suffices to yield eigenmodes that provide compact representations of macroscopic activity measurable by fMRI. This is why we also evaluated the representation of synthetic EDR connectome eigenmodes, which showed comparable performance to the geometric eigenmodes.

We believe that this point may have been obscured in our initial exposition, which presented NFT's wave equation in differential form. The distance-dependence of the activity at any given point is more clearly seen when the equation is expressed in its equivalent integral form (Eq. (S16) of the revised Supplementary Material) (Nunez, 1974; Jirsa and Haken, 1996; Robinson et al., 1997).

We have added the following text to Results and Supplementary Material-S12.1 to clarify the above points:

Main Text Lines 320–322: Under this model, activity propagates between points on the neocortex through their white-matter connectivity, with a strength that decays approximately exponentially with distance (Supplementary Material-S1 and S12.1).

Supplementary Material Lines 534–563: Note that the propagation of activity between points is governed by their white-matter connectivity, with strength that decays approximately exponentially with distance. This distance-dependence is more apparent when Eq. (S13) is converted into its equivalent integral form as follows. Consider two points on the neocortex at locations \mathbf{r}' and \mathbf{r} connected by a white-matter tract (see visual schematic in Fig. S18). The activity $\phi(\mathbf{r}, t)$ at location \mathbf{r} and time t can be considered as a spatiotemporal convolution of the source $Q(\mathbf{r}', t')$ at location \mathbf{r}' and time t' and the white-matter-based connectivity kernel $W(\mathbf{r}, t; \mathbf{r}', t')$; i.e.,

$$\phi(\mathbf{r}, t) = \int W(\mathbf{r}, t; \mathbf{r}', t') Q(\mathbf{r}', t') d^2 \mathbf{r}' dt'. \quad (\text{S14})$$

In the isotropic case, W depends only on the spatial separation between points and the time difference such that $W(\mathbf{r}, t; \mathbf{r}', t') := W(\mathbf{r} - \mathbf{r}', t - t')$. The kernel W is also known as the Green's function, which is the activity ϕ due to a point source. Using the Green's function method⁷⁰, previous work has shown that an appropriate expression for W , such that it becomes a solution of the damped wave equation in Eq. (S13) with an intense point source $Q(\mathbf{r}, t; \mathbf{r}', t') = \delta(\mathbf{r} - \mathbf{r}')\delta(t - t')$ and for $|\mathbf{r} - \mathbf{r}'| \leq \gamma_s r_s(t - t')$, is^{4,6,71,72}

$$W(\mathbf{r} - \mathbf{r}', t - t') = \frac{\gamma_s}{r_s} \frac{\exp\left(\frac{-|\mathbf{r} - \mathbf{r}'|}{r_s}\right)}{\sqrt{\gamma_s^2 r_s^2 (t - t')^2 - |\mathbf{r} - \mathbf{r}'|^2}} \Theta[\gamma_s r_s (t - t') - |\mathbf{r} - \mathbf{r}'|], \quad (\text{S15})$$

where $|\mathbf{r} - \mathbf{r}'|$ is the white-matter tract distance between points, Θ is the Heaviside step function, and γ_s and r_s are the damping rate and spatial length scale, respectively, as defined in Eq. (S13). See⁶ for a detailed derivation of Eq. (S15). Equation (S15) thus demonstrates how the wave dynamics defined by Eq. (S13) can be directly related to an underlying isotropic anatomical connectivity that decays exponentially with distance. In particular, solving the damped wave equation in differential form in Eq. (S13) is equivalent to solving the integral equation,

$$\phi(\mathbf{r}, t) = \frac{\gamma_s}{r_s} \int \frac{\exp\left(\frac{-|\mathbf{r} - \mathbf{r}'|}{r_s}\right)}{\sqrt{\gamma_s^2 r_s^2 (t - t')^2 - |\mathbf{r} - \mathbf{r}'|^2}} \Theta[\gamma_s r_s (t - t') - |\mathbf{r} - \mathbf{r}'|] Q(\mathbf{r}', t') d^2 \mathbf{r}' dt', \quad (\text{S16})$$

with the latter explicitly showing the spatiotemporal effect invoked by white-matter connectivity of a characteristic range of r_s (~ 84 mm⁶).

New Fig. S18:

Fig. S18. Schematic of signal propagation in neural field theory. Two points on the cortex at locations \mathbf{r}' and \mathbf{r} are connected by a white-matter tract (red curve). For an isotropic medium, the activity $\phi(\mathbf{r}, t)$ is a convolution of the source $Q(\mathbf{r}', t')$ and a connectivity kernel $W(\mathbf{r}, t; \mathbf{r}', t')$ imposed by the white-matter connection, which depends only on the spatial separation, $\mathbf{r} - \mathbf{r}'$, and the time separation, $t - t'$.

Comment 4.2: *Since we know that neural transmission happens through white matter intricate connections, here the interest should lie more in investigating which *kind* of information present in the fMRI signal can be justified by brain geometry, and which one cannot. This would go beyond the sole comparison of reconstruction accuracies (*amount* of information), which appear anyway fairly similar in the case of geometric and connectome eigenmodes (and depending a lot on how the connectome is constructed). In Fig. 2C and D the authors compare the reconstruction accuracy between connectome and geometric eigenmodes, claiming it higher for the geometric case. However, they do not discuss how initial modes from structural connectomes -the ones with more energy- appear to perform better than geometric ones (e.g. up to the first 15/20). This goes well with the hypothesis that SC modes would, as expected, capture large-scale distributed networks, “function-wise” important, better than geometric ones present with the same energy in the functional data.*

Further exploration of this matter would help in interpreting which portion of functional activity measured via fMRI can be explained well by geometry and which one cannot, since more dependent on the white matter architecture.

Response 4.2: The Reviewer raises an interesting point. It is true that connectome eigenmodes seem to perform better than geometric eigenmodes in reconstructing some task-activation maps, but not resting-state FC, using lower-order modes (<10). This could allude to the potential advantage that long-range connections provide for shaping large-scale patterns of evoked activity. However, in absolute terms, the accuracies achieved by reconstructions comprising <10 connectome modes remain relatively low (average $r = 0.42$ across the different tasks) compared to using, for example, at least 100 modes (average $r = 0.71$), so it is difficult to confidently claim that this represents some differential effect of long-range connections.

We have revised the caption of Fig. 2 to briefly cover the above point:

Main Text Lines 227–230: Note that while there seems to be a performance advantage for connectome eigenmodes for reconstructions incorporating <10 modes relative to geometric eigenmodes, the reconstruction accuracy is generally low (average \$r = 0.42\$ across the different tasks) compared to the accuracy for 100 modes (average \$r = 0.71\$ ).

Caution is also warranted in drawing a sharp distinction between geometric eigenmodes and “long-range connectivity”. As outlined in Response 4.1, geometric eigenmodes incorporate long-range connectivity that is explained by an EDR-like rule. The primary distinction is between axonal connectivity that can be accounted for by a simple EDR-like rule and more topologically complex short- and long-range connections, with the latter empirically captured by the connectome eigenmodes. Our results suggest that the EDR-like approximation is sufficient to explain a diverse range of brain dynamics.

We have made the following revisions to the manuscript to ensure our terminology is precise:

Main Text Lines 91–92: ... a combination of topologically complex short- and long-range connectivity ...

Main Text Lines 454–455: ... topologically complex connections that exist beyond a simple EDR ...

Supplementary Material Line 240: ... complex short- and long-range connections ...

We have also added the following text to Supplementary Material-S12.1 to more clearly explain this distinction:

Supplementary Material Lines 618–621: Hence, the use of geometric eigenmodes implicitly incorporates an EDR-like connectivity, which also includes long-range connections, but does not directly account for topologically complex connections not conforming to a simple exponential rule.

We agree that our findings raise important questions about the functional role and significance of topologically complex long-range connections that are not explained by EDR-like connectivity. We are currently working on approaches to address this question, but it is outside the present scope.

We have added the following text to the Discussion section to acknowledge this as a future extension of our work:

Main Text Lines 457–465: Our findings thus counter traditional views that emphasize intricate patterns of anatomical connections as the primary driver of coordinated dynamics^{33,63,64}. Indeed, recent estimates indicate that long-range cortical connections are relatively rare⁶⁵; they may therefore represent a relatively minor perturbation of the dominant effect imposed by EDR-like connectivity. Nonetheless, the topological centrality, metabolic cost, and tight genetic control of such connections^{66–68} suggest that they provide important functional and evolutionary advantages beyond wave-like dynamics⁶⁹. The limited resolution and sensitivity to preprocessing pipelines^{70,71} of dMRI and fMRI data complicate attempts to uncover the role of long-range connections, but high-quality animal tract-tracing and electrophysiological data may be helpful in this regard.

Comment 4.3: *In Fig. 4C the authors show the propagation of activity with the wave model after stimulation of the visual cortex, bringing it as an evidence to claim that stimulus-evoked cortical activity would be solely driven by geometry and not by “complex patterns of connectivity”. Since we know from previous literature that white matter anatomy in the visual cortex indeed accounts for directionality and propagation of neural activation, it would seem more fair to say that geometry is enough to explain (a big portion of) what we record at the level of fMRI signals, rather than it is enough to justify thoroughly neural activation processes in the brain. In this regard, it would be interesting to see how the wave model performs in explaining more complex high-level cognitive processes involving widely distributed networks, which supposedly would require long-range neural connections.*

Response 4.3: We agree with the Reviewer that our results only show that, at fMRI macroscale level, geometry seems to drive a significant portion of observed phenomena, such as the dorsal-ventral stream separation of activity; however, other factors also likely contribute, such as complex connectivity. We have toned down our language in regard to this point to ensure that we do not discredit the huge body of work on the importance of the intricate microscale visual cortex connectivity involved in structures such as ocular dominance and orientation preference columns.

We have revised the following text in the Results and Discussion sections accordingly:

Main Text Lines 374–377: Remarkably, this result indicates that geometric constraints on travelling waves of evoked activity are sufficient for the segregation of the dorsal and ventral processing streams, which have traditionally been thought to be mainly driven by complex patterns of layer-specific connectivity ^{52,54,55}.

Main Text Lines 508–514: These canonical properties of hierarchical visual processing have been extensively studied for decades and are classically thought to be driven by complex patterns of layer-specific inter-regional connectivity ^{52,54,55}, but our analysis shows that waves traveling through the cortical geometry are sufficient for the emergence of segregated, hierarchical processing streams. In other words, our findings cannot rule out a role for complex inter-regional connectivity, but they do indicate that such connectivity is not necessary for the emergence of these macroscale dynamics.

We agree that it would be interesting to investigate whether the model can explain ‘complex high-level cognitive processes’, but this is difficult to simulate. We chose the visual system because the location of the input is well defined (i.e., V1), as is the expected sequence of evoked activity. For more complex stimuli, the locations and nature of the inputs are not as clear, and the patterns of evoked activity are less well understood. However, our analysis of the wave model’s ability to capture patterns of spontaneous resting-state activity indicates that it can indeed account for complex spatiotemporal patterns of activity, given that the spatiotemporal structure of spontaneous BOLD signals is similar to that evoked by tasks (Smith et al., 2009). We have also included an additional analysis to investigate whether the wave model can also explain more complex time-lagged properties of resting-state data. In particular, we measured lag threads in resting-state fMRI data previously proposed by Mitra et al. (2014, 2015), which capture the temporal properties of propagated activity across the whole brain. See Supplementary Material-S12.6 or the revisions in the text shown below for details about this method. The new results presented in Fig. S19 below show that the wave model can capture the time-lagged structure of the empirical data better than the mass model. This further supports that wave dynamics can provide a unifying account of complex properties of resting-state fMRI data. Further details of this analysis are also presented in Response 2.2.

We have added new text to Supplementary Material-S12.6 to explain the method for measuring time-lagged properties of fMRI data. We have also added new text to the Results and Discussion sections to discuss the results from this new analysis.

Supplementary Material Lines 811–858: In addition to the FC-based metrics used for model fitting and evaluation of model performance (Section S12.4), we investigated whether the wave and mass models can also capture the temporal properties of propagated activity. In particular, we analyzed the lag structure (or lag threads) of resting-state BOLD-fMRI time courses, as proposed by ^{95,96}. We briefly discuss the algorithm for calculating the lag structure of both empirical and simulated fMRI data below and refer the readers to previous articles ^{95,96} for further details. Note that there are several other ways of characterizing the spatiotemporal properties of resting-state activity ^{97–99}, but the currently chosen method is sufficient for our purposes.

The algorithm starts by calculating the lagged cross-covariance function of the time series between brain regions. Assuming that BOLD-fMRI time series are aperiodic ¹⁰⁰, the time lag (or delay) between regions where the cross-covariance function exhibits an extremum (typically between 0 to 2 s) was obtained ⁹⁵. This resulted in an anti-symmetric time-delay matrix TD , with elements τ_{ij} corresponding to the time delay between regions i and j , with column i representing the lag map of the system with reference to region i , and $\tau_{ij} = -\tau_{ji}$. Therefore, $\tau_{ij} > 0$ means that region j lags behind region i . The underlying lag structure was quantified in two ways: (i) taking the mean time lag of each region (mean of each column of TD); and (ii)

applying a PCA on TD_z , which is TD with each column being zero-meaned. The first method obtains the average temporal ordering of brain regions, assuming that a single lag process governs the brain, which has been used in several past studies^{95,99,101}. However, for systems like the brain with multiple lag processes, the mean time lag cannot capture all fundamental lag patterns, which can be recovered via PCA⁹⁶. Here, we used the first two dominant PCs, explaining 74% of the variance of the empirical data. Thus, we calculated three lag projections: (i) mean lag; (ii) first PC (PC1 lag); and (iii) second PC (PC2 lag).

The TD matrix was calculated on empirical and simulated resting-state BOLD-fMRI time courses parcellated using the HCP-MMP1 parcellation (Section S5). For the empirical data, a TD matrix was calculated for each of the 255 HCP individuals (Section S4.1 and S4.3), and then an average TD matrix was calculated across individuals. For the simulated data, we generated time series for 255 trials (to match the 255 HCP individuals of the empirical data) of the wave and mass models using their respective original optimized parameters (Section S12.5). A TD matrix was calculated for each trial. Then, an average TD matrix was calculated across trials. The average TD matrices are shown in Fig. S19A. Finally, the three lag projections were obtained from the empirical and simulated average TD matrices (Figs. S19B to S19D).

The results in Fig. S19B show that the mean lag pattern of the wave model is significantly correlated with the empirical pattern, whereas the mass model's mean lag pattern is not. For PC1 lag, the performance of the wave model slightly decreased but was still superior to the mass model (Fig. S19C). Correlations between model and empirical patterns for PC2 lag (Fig. S19D) were not significant, but the correlation was still higher for the wave model. These results show that, regardless of the method for calculating lag projections, the wave model captures the time-lagged properties of empirical fMRI data better than the mass model. This further highlights that wave dynamics can provide an accurate and physically mechanistic account of macroscale, resting-state dynamics, consistent with previous studies^{102–104}. We also emphasize that the performance of these models in capturing lag structure is likely to be a conservative estimate, as the models rely on a simple and spatially uniform hemodynamic forward model that does not account for regional variations in neurovascular coupling^{86,105,106}. Such variations are likely to strongly affect the empirically observed lags.

Main Text Lines 336–337: Additionally, the wave model can better capture time-lagged properties^{46,47,51} of empirical resting-state activity compared to the mass model (Supplementary Material-S12.6 and Fig. S19).

Main Text Lines 499–501: This finding is consistent with experimental observations of wave dynamics in both human and animal fMRI data^{79,80}.

New Fig. S19:

Fig. S19. Comparison of the wave and mass models in capturing time-lagged properties of fMRI data. (A) Time-delay matrices from empirical data, simulated data using the wave model, and simulated data using the mass model of the left hemisphere. Negative–zero–positive values are colored as blue–white–red. (B) Mean lags from the matrices in panel A (mean of each column) projected on the cortical surface. Negative–zero–positive values are colored as blue–white–red. The scatter plots show the relationship of mean lags from empirical data and simulated data from the two models. The red line represents a linear fit with Pearson correlation coefficient r and spin-test p -value p_{spin} from 10,000 permutations. (C) Similar to panel B but on the first principal component (PC1) of the matrices in panel A. The number above the surfaces (var) corresponds to the variance explained by the PC. (D) Similar to panel B but on the second PC (PC2) of the matrices in panel A.

Comment 4.4: The authors limit the exploration at $N=200$ eigenmodes throughout the study, but this is very few w.r.t the full decomposition. Low frequency eigenmodes are known to explain most of the energy in the signal, but it would be interesting to see what happens also at higher frequency modes, if they have a role in explaining individual features of functional activity, as it is known to happen for connectome modes, or not. In other words, the reconstruction with $N=200$ modes reaches around 80-90% accuracy depending on the task, but what is the remaining part of the signal, not explainable with geometric low frequency modes, therefore not directly inferable from geometry?

This would also help disentangling the part of information that is common to the population, possibly driven by the task and comprising low-frequency modes, and the one that is more peculiar to different individuals.

Response 4.4: The Reviewer raises an important point. To address this comment, and Comment 2.6, we performed two new additional analyses: (i) we compared reconstruction accuracy at the individual level using 200 template-derived and individual-specific eigenmodes to better understand the role that individual geometry plays (new Fig. S5 presented below); and (ii) we calculated reconstruction accuracy up to 500 modes to understand whether very high-frequency individual-specific modes can explain the remaining 10% of the data unexplained by the first 200 modes (new Fig. S6 presented below).

The new Fig. S5 shows that, for many participants, individual-specific eigenmodes performed slightly better than template-derived eigenmodes, especially when reconstructing task-activation maps but not resting-state FC. This result suggests that individual differences in cortical geometry play a role in shaping inter-individual variability in stimulus-evoked activity.

The new Fig. S6 shows that modes 200 to 500 only improve reconstruction accuracy slowly and minimally (maximum improvement in accuracy of <0.02). It is interesting to note though that the performance of template-derived and individual-specific eigenmodes seem to converge at the very high-frequency modes, suggesting that much of the nuances of individual-specific geometry are encoded within the first 200 modes, which have spatial wavelengths >29 mm. This result is consistent with our own recent study showing that the most reliable individual differences in cortical geometry peak at around mode 140 (Chen et al., 2022). Remarkably, 200 modes covers a very small fraction of the maximum possible 32,492 modes (number of surface vertices = maximum number of modes).

We have added the following text to Results and Supplementary Material-S2 to discuss the above points:

Main Text Lines 160–163: However, there are some participants where individual-specific eigenmodes perform slightly better, especially in reconstructing task-activation maps (Fig. S5), but their accuracies eventually converge with template-derived eigenmodes at short wavelengths (Fig. S6).

Supplementary Material Lines 91–98: Nonetheless, to highlight certain nuances of individual geometry, we show in Fig. S5 that 200 individual-specific eigenmodes perform slightly better than template-derived eigenmodes in some individuals, especially in reconstructing task-activation maps but not in reconstructing resting-state activity. However, Fig. S6 shows that the results for individual-specific and template-derived eigenmodes converge at very short wavelengths ($\sim 500^{\text{th}}$ mode). Thus, the effects of individual differences in cortical geometry on brain function are captured by the first 200 modes, consistent with recent work³⁹, which corresponds to a very small fraction of the maximum possible number of eigenmodes ($\sim 0.6\%$).

New Fig. S5:

Fig. S5. Individual-based reconstruction accuracy of 7 key HCP task-contrast maps and resting-state FC using 200 template-derived and individual-specific geometric eigenmodes. See Section S4.2 and Table S2 for details about the contrast maps. wm = working memory. Template eigenmodes were derived from a template surface, while individual-specific eigenmodes were derived from individual-specific surfaces. Each point corresponds to an individual. The dotted lines represent the template accuracy = individual-specific accuracy lines. Points above the dotted lines mean that template accuracy > individual-specific accuracy. The insets show the histogram of the difference between the reconstruction accuracy achieved by template eigenmodes and individual-specific eigenmodes (i.e., template minus individual-specific).

New Fig. S6:

Fig. S6. Reconstruction accuracy of 7 key HCP task-contrast maps and resting-state FC using 200 to 500 template-derived and individual-specific geometric eigenmodes. See Section S4.2 and Table S2 for details about the contrast maps. wm = working memory. The solid lines represent results achieved by template eigenmodes derived from a template surface (Fig. 1A). The dashed lines represent results achieved by

individual-specific eigenmodes derived from individual-specific surfaces. The insets show the difference between the two results (i.e., template minus individual-specific).

Comment 4.5: *About the FC reconstruction with the two models (Fig. 4), authors claim the wave model performs better than the neural mass model, but a clear benefit cannot be appreciated nor when looking visually at the matrix, nor when comparing correlation values (edge-wise correlation of 0.4). In addition to the metrics proposed to evaluate the models, it would be interesting to see if the FC main blocks are preserved in these reconstructions.*

Response 4.5: We agree that evaluation of edge-wise FC shows that the wave and mass models have comparable performance. We arrived at the conclusion that the wave model performs better by considering all the FC-based metrics assessed; i.e., static (edge and node FC) and dynamic (FCD) properties. For node-based FC, the correlations are 0.59 and 0.51 for the wave and mass models, respectively. For FCD, the KS statistic is 0.08 and 0.19 for the wave and mass models, respectively (lower KS values indicate better fits). More importantly, the wave model is much simpler than the mass model both in terms of the number of parameters and its construction. This emphasizes that the wave model is not just more accurate but also more parsimonious in accounting for properties of resting-state data compared to the mass model. We understand that there is a myriad of other metrics that can be used to compare the two models. However, we used the three FC-based metrics above as they have been widely used for evaluating neural mass models in past work (Demirtaş et al., 2019; Deco et al., 2021; Aquino et al., 2022); hence, our results are easily interpretable and contextualized with the extant literature.

As per Response 4.3, we have added a new analysis (new Fig. S19 reproduced below for convenience) to show that the wave model accounts for the structure of lagged interactions between brain regions, otherwise known as “lag threads”, better than the neural mass model. This is a stronger constraint on model performance than FC-based metrics. Please see Response 4.3 (and also Response 2.2) and the associated revisions to the text presented there for details. These results provide a more extensive and stronger support to the hypothesis that wave dynamics can provide a unifying account of the properties of resting-state fMRI data.

New Fig. S19:

Fig. S19. Comparison of the wave and mass models in capturing time-lagged properties of fMRI data. (A) Time-delay matrices from empirical data, simulated data using the wave model, and simulated data using the mass model of the left hemisphere. Negative–zero–positive values are colored as blue–white–red. (B) Mean lags from the matrices in panel A (mean of each column) projected on the cortical surface. Negative–zero–positive values are colored as blue–white–red. The scatter plots show the relationship of mean lags from empirical data and simulated data from the two models. The red line represents a linear fit with Pearson correlation coefficient r and spin-test p -value p_{spin} from 10,000 permutations. (C) Similar to panel B but on the first principal component (PC1) of the matrices in panel A. The number above the surfaces (var) corresponds to the variance explained by the PC. (D) Similar to panel B but on the second PC (PC2) of the matrices in panel A.

Comment 4.6: In Figure S7, authors assess the effect of thresholding the connectome at different densities, showing that this has an impact on the reconstruction accuracy measure. First, we can indeed expect methodological choices in the construction of the connectomes to affect the results, so the effect of these variables – such as the tractography algorithms used and the type of measure taken as connectivity metric, in this case the number of streamlines not normalized for regional volumes- should be considered. Second, the density analysis is not completely clear to me: fig. S7 shows that lower density connectomes lead to better reconstruction accuracy. How do the authors explain then that between the two connectomes considered in

the main analysis (Fig. 2), that should only differ in density, the one at higher density (1.55% vs 0.1%) performs better?

Response 4.6: We agree that several aspects can affect the quality of generated connectomes from diffusion MRI, including preprocessing steps, tractography pipelines, and thresholding (see for example a recent study from our group that explored this; Gajwani et al., *bioRxiv*, 2022). That study shows that parcellation type and threshold exerted some of the biggest effects on connectome architecture, so we evaluated the effects of these choices in our analysis to verify the robustness of our results and give the connectome eigenmodes the best possible chance.

Regarding the comment on streamline normalization, we used a high-resolution connectome with each surface mesh vertex treated as a node; normalization is therefore not required.

We have added the following text to Supplementary Material-S6 to briefly mention this point:

Supplementary Material Lines 224–225: Connection weights between vertices (considered as nodes) were estimated as the number of interconnecting streamlines without the need for normalization⁵⁵.

With respect to the density issue, we thank the Reviewer for raising this point, which led us to perform an additional analysis to more finely sample density thresholds from 0.1% to 5%. The results of this analysis, presented in the new Fig. S10 below, interestingly show that the reconstruction accuracy of task-activation maps and resting-state FC using 200 connectome eigenmodes does not follow a monotonic function with respect to density threshold. In fact, the accuracy increases from a density of 0.1%, then reaches an optimal value between 0.4% and 0.7%, before decreasing monotonically as the density is further increased. However, regardless of density threshold used, the performance of the connectome eigenmodes is always substantially lower than that of the geometric eigenmodes.

We have added the following text to Supplementary Material-S6 to discuss the results from this new analysis:

Supplementary Material Lines 269–276: Notably, finely sampling connection densities from 0.01% to 5% reveals a putatively optimal threshold between 0.4% and 0.7% that results in peak reconstruction accuracy for 200 connectome eigenmodes. This optimal threshold may represent a physiologically plausible trade-off between sensitivity and specificity in tract reconstruction with diffusion MRI. However, the reconstruction accuracy never surpasses the accuracy of the geometric eigenmodes. The sensitivity of the reconstruction accuracies obtained with connectome eigenmodes to the density threshold further underscores the complexity of this approach in comparison to geometric eigenmodes, which require no choices regarding a specific threshold.

New Fig. S10:

Fig. S10. Reconstruction accuracy of 7 key HCP task-contrast maps and resting-state FC using 200 connectome eigenmodes of varying connectome densities. See Section S4.2 and Table S2 for details about the contrast maps. wm = working memory. The solid line corresponds to the reconstruction accuracy achieved by 200 geometric eigenmodes. The dotted lines correspond to connectome densities of 0.1% and 1.5% used to generate the connectome and density-matched connectome eigenmodes in Fig. 2, respectively.

Reviewer Reports on the First Revision:

Referee #1 (Remarks to the Author):

This manuscript shows that a brain basis set based on surface topography of the brain represents an efficient basis set for modeling function.

The work is innovative and provides an interesting window on the relationship between structure and function.

The data and methodology are sound and the work uses appropriate statistical methodology.

My questions from the previous round have been addressed and I have no further concerns and recommend the work for publication in Nature.

Referee #2 (Remarks to the Author):

The authors have convincingly addressed the points raised in the prior review, and their work adds to our understanding of how systems-level activity is organized in the brain. Congratulations on a very nice piece of work!

Regards,
Shella Keilholz

Referee #3 (Remarks to the Author):

The authors have performed an impressive amount of work to address the concerns raised by me and the other reviewers. I still think this is a highly original and relevant work, and now I'm also convinced about the technical side of the manuscript. As I said in my original review, I now fully endorse the manuscript for publication in Nature.

-Enzo Tagliazucchi

Referee #4 (Remarks to the Author):

The authors made a considerable effort to answer all reviewers' concerns, including some additional analyses which go mainly in the Supplementary Material of the revised manuscript. In particular, what I think is more explicit in the new formulation is how white matter still plays a role in the proposed NFT model, even if this could be further underlined in the text.

I have an additional comment specific to response 4.2: I suggested that the importance of the connectome eigenmodes might be reflected mostly on the low frequency components, that have a superior performance than geometry in explaining brain function (as it's very clear from fig. 2D for the density matched connectome). I don't think that it's relevant, in this context, to comment on the obviously smaller reconstruction accuracy of the first 10 components, compared to 100, as this was not the point raised. The aim of my comment was rather to evaluate (and interpret, potentially in

the discussion) the importance of these low frequency patterns in the connectome case with respect to the geometric mode case.

Author Rebuttals to First Revision:

Reviewer Responses

We thank the Reviewers for their positive feedback. In what follows, we provide a point-by-point response (in regular front) to each comment (in *italics*), which we number for ease of reference. Where applicable, text edits in the revised manuscript are highlighted in blue.

Response to Reviewer 4

Comment 4.1: *The authors made a considerable effort to answer all reviewers' concerns, including some additional analyses which go mainly in the Supplementary Material of the revised manuscript. In particular, what I think is more explicit in the new formulation is how white matter still plays a role in the proposed NFT model, even if this could be further underlined in the text.*

I have an additional comment specific to response 4.2: I suggested that the importance of the connectome eigenmodes might be reflected mostly on the low frequency components, that have a superior performance than geometry in explaining brain function (as it's very clear from fig. 2D for the density matched connectome). I don't think that it's relevant, in this context, to comment on the obviously smaller reconstruction accuracy of the first 10 components, compared to 100, as this was not the point raised. The aim of my comment was rather to evaluate (and interpret, potentially in the discussion) the importance of these low frequency patterns in the connectome case with respect to the geometric mode case.

Response 4.1: As we have mentioned in our previous response to the Reviewer's comment, whilst low-order connectome eigenmodes (<10) seem to perform better than geometric eigenmodes in reconstructing task-activation maps, the absolute accuracies are low. It is important to explain this result as it provides a necessary context for interpreting the apparent performance advantage of connectome modes at low frequencies. To further address Reviewer 4's comment, we have added the following text to the Discussion of the revised manuscript:

Lines 378–384: Figure 2d alludes to these advantages, where low-order connectome eigenmodes perform slightly better in reconstructing task-activation maps at low frequencies. Low-order connectome eigenmodes comprise more complex spatial patterns than low-order geometric eigenmodes, which may provide them greater flexibility in capturing spatially complex patterns of task activations. However, this advantage for connectome eigenmodes only persists for the first 25–30 modes, where reconstruction accuracies are generally low (i.e., \$r < 0.50\$ ) and do not differ substantially from the EDR eigenmodes.